# Scalable Element-wise Finite-Time Optimization for Deep Neural Networks

## Abstract

Optimization algorithms are fundamental to deep neural network training, where exponential growth from millions to hundreds of billions of parameters has made training acceleration a critical necessity. While adaptive methods like Adam achieve remarkable success through element-wise learning rates, understanding their continuous-time counterparts can provide valuable theoretical insights into convergence guarantees beyond asymptotic rates. Recent advances in continuous-time optimization have introduced fixed-time stable methods that promise finite-time convergence independent of initial conditions. However, existing approaches like FxTS-GF suffer from dimensional coupling, where coordinate updates depend on global gradient norms, creating suboptimal scaling in high-dimensional problems typical of deep learning. To address this issue, we introduce an element-wise finite-time optimization framework that eliminates dimensional coupling through coordinate-independent dual-power dynamics. Furthermore, we extend the framework to momentum-enhanced variants for deep model training while preserving convergence properties through continuous-time analysis. Under mild assumptions, we establish rigorous finite-time and fixed-time convergence guarantees. Notably, our framework reveals that widely-used sign-based optimizers like SignSGD and Signum emerge as limiting cases, providing theoretical grounding for their empirical effectiveness. Experiments on CIFAR-10/100 and C4 language modeling demonstrate consistent improvements over existing methods.

## 1 Introduction

Optimization algorithms are the cornerstone of deep neural network training, determining both the feasibility and efficiency of learning in large-scale models. As neural networks have grown exponentially, training acceleration has evolved from a convenience to a critical necessity, where a single large language model(LLM) can require thousands of GPU-hours and cost millions of dollars to train. This computational reality has driven decades of intensive research into faster optimization methods. The journey began with stochastic gradient descent (SGD), the foundational algorithm that enabled neural network training, followed by momentum-based acceleration techniques like heavy-ball momentum (Polyak, 1964) and Nesterov acceleration (Nesterov, 1983) that significantly improved convergence rates.The development of adaptive methods marked a major breakthrough: AdaGrad (Duchi et al., 2011) introduced coordinate-wise learning rates, RMSprop improved upon this with exponential moving averages, while Adam (Kingma & Ba, 2014) combined both first and second moment estimation and AdamW (Loshchilov & Hutter, 2017) which decouples weight decay. More recent advances include second-order methods like Shampoo (Gupta et al., 2018) that use full preconditioning matrices, and Sophia (Liu et al., 2023) which efficiently approximates Hessian information for LLM training representing the current pinnacle of discrete optimization approaches.

However, the optimization methods described above are fundamentally designed from a discrete-time perspective, focusing on iterative parameter updates with step-by-step convergence analysis. While this discrete viewpoint has achieved remarkable practical success, it inherently limits the theoretical frameworks available for convergence analysis. In contrast, continuous-time optimization theory offers fundamentally different analytical tools through dynamical systems theory that can provide stronger convergence guarantees. Recent advances have introduced finite-time and fixed-time stability concepts (Bhat & Bernstein, 2000; Polyakov, 2011), where systems reach equilibrium exactly within bounded time horizons, with fixed-time variants providing bounds independent of

initial conditions. The pioneering work by Budhraja et al. (2022) first applied these concepts to optimization and deep learning model training, introducing fixed-time stable gradient flows (FxTS-GF), employing dynamics of the form:

$$\dot{\mathbf{w}}(t) = -c_1 \frac{\nabla \mathcal{L}(\mathbf{w})}{\|\nabla \mathcal{L}(\mathbf{w})\|^{\frac{p_1-2}{p_1-1}}} - c_2 \frac{\nabla \mathcal{L}(\mathbf{w})}{\|\nabla \mathcal{L}(\mathbf{w})\|^{\frac{p_2-2}{p_2-1}}}, \tag{1}$$

where $c_1, c_2 > 0$, $p_1 > 2$ and $p_2 \in (1, 2)$, which demonstrated superior performance than Adam in solving the Rosenbrock function and training shallow neural networks model using the MNIST dataset. While these methods provide elegant theoretical guarantees, the global norm $\|\nabla \mathcal{L}(\mathbf{w})\|$ creates dimensional coupling that becomes problematic in large-scale settings. In high-dimensional networks, this global normalization is dominated by the largest gradient components, diminishing updates for smaller but potentially crucial gradients. This limitation highlights a critical gap between control-theoretic optimization and practical deep learning requirements. In contrast, the remarkable success of Adam-family optimizers in deep learning stems from their element-wise adaptivity, where each parameter maintains its own learning rate based on local gradient statistics. This design philosophy naturally handles the heterogeneous optimization landscape and has proven essential for training neural networks, especially for transformer-based models (Zhang et al., 2024a).

Motivated by this insight, we ask: *Can we bring the theoretical rigor of finite/fixed-time optimization to large-scale deep learning by adopting element-wise design principles?*

We introduce an element-wise finite-time optimization framework specifically designed for the challenges of large-scale neural network training:

$$\dot{\mathbf{w}} = -c_1 \operatorname{sign}(\mathbf{g}) \odot |\mathbf{g}|^{p_2} - c_2 \operatorname{sign}(\mathbf{g}) \odot |\mathbf{g}|^{2-p_1} = -c_1 \frac{\mathbf{g}}{|\mathbf{g}|^{1-p_2}} - c_2 \frac{\mathbf{g}}{|\mathbf{g}|^{p_1-1}}, \tag{2}$$

where $\mathbf{g} = \nabla \mathcal{L}(\mathbf{w})$ is the gradient of the objective function $\mathcal{L}(\mathbf{w})$, $c_1, c_2 > 0$, $p_1 < 2$ and $p_2 \in (0, 1)$, $\odot$ denote the element-wise operations. This design eliminates dimensional coupling by making each coordinate evolve independently, preserving the element-wise adaptivity crucial for deep learning while maintaining rigorous convergence guarantees from control theory. Our framework reveals a remarkable theoretical bridge between control-theoretic optimization and practical deep learning methods. When parameters approach certain limits ($p_1 \to 2$, $p_2 \to 0$), our dynamics reduce to SignSGD (Bernstein et al., 2018). This connection suggests that successful sign-based optimizers used in practice are actually principled approximations of theoretically grounded finite/fixed-time gradient flows. Our contributions are as follows:

1. **Element-wise finite/fixed-time optimization framework**: A coordinate-independent reformulation of fixed-time gradient flows that eliminates dimensional coupling through element-wise dual-power dynamics, enabling scalable application to deep neural networks with rigorous convergence guarantees

2. **Momentum-enhanced variants with rigorous theoretical guarantees**: We extend our framework to incorporate exponential moving averages and Polyak momentum for stochastic training. We establish: (i) finite/fixed-time convergence for the continuous-time dynamics, and (ii) sublinear convergence rates for the discrete algorithms under mild assumptions.

3. **Theoretical unification of sign-based optimizers**: We show that widely used distributed training optimizers SignSGD and Signum emerge as limiting cases of our method, providing theoretical foundation for their empirical effectiveness in large-scale model training.

## 2 RELATED WORK

**Deep Learning Optimization:** The success of large-scale neural network training fundamentally relies on adaptive optimization methods that adjust learning rates based on gradient statistics. This paradigm began with AdaGrad (Duchi et al., 2011), which introduced coordinate-wise learning rates by accumulating squared gradients, and was further developed by the Adam family of optimizers (Kingma & Ba, 2014; Loshchilov & Hutter, 2017). Adam combines first and second moment estimation with exponential moving averages, while AdamW decouples weight decay from gradient-based updates, and AdaBelief (Zhuang et al., 2020) refines second-moment estimation to better capture gradient predictability, leading to their widespread adoption across various deep learning applications. Adam's widespread adoption across deep learning applications (Orvieto & Gower, 2025)

reflects its consistent performance advantages over SGD, particularly pronounced in transformer architectures. Recent theoretical work by Zhang et al. (Zhang et al., 2024a) explains this effectiveness by analyzing transformer Hessian structures, revealing significant coordinate heterogeneity where different parameters experience vastly different curvature properties. This heterogeneous structure necessitates coordinate-specific learning rates, providing theoretical justification for element-wise adaptive methods. Dong et al. (2025) further quantify this heterogeneous structure, establishing rigorous mathematical foundations for adaptive optimization's empirical success. The principle of heterogeneity-aware optimization has manifested in various forms across the field's development. Earlier work like LAMB (You et al., 2019) recognized that different network depths require distinct optimization strategies, extending adaptive principles to layer-wise normalization for effective large-batch training of transformers. More recently, Adam-mini (Zhang et al., 2024b) explicitly exploits the block-diagonal structure of neural network Hessians, assigning learning rates per dense sub-block while maintaining computational efficiency. These methods demonstrate that the key insight extends beyond simple element-wise adaptation to various forms of structured, heterogeneity-aware optimization. More recent advances continue to push the boundaries of adaptive optimization. Advanced preconditioning methods like Shampoo (Gupta et al., 2018) and SOAP (Vyas et al., 2024) implement block-diagonal preconditioning matrices, while ASGO (An et al., 2025) introduces adaptive structured gradient optimization. Second-order approaches like Sophia (Liu et al., 2023) efficiently approximate Hessian information specifically for large language model training. For distributed training of large models, sign-based methods like SignSGD, Signum (Bernstein et al., 2018), and Lion (Chen et al., 2023) maintain adaptive characteristics while providing communication efficiency.

**Finite-Time and Fixed-Time Optimization Theory:** The theory of finite-time stability has its roots in control systems, where Bhat & Bernstein (2000) established fundamental results for systems that reach equilibrium in finite time rather than asymptotically. Polyakov Polyakov (2011) extended this framework to fixed-time stability, providing uniform convergence bounds independent of initial conditions, which is particularly valuable for control applications with strict timing requirements. These theories are then extended to the optimization Garg & Panagou (2021), and distributed optimization Chen & Li (2018), with applications on multi-agent system and ummaned autonomus system Liu et al. (2022). In Nguyen et al. (2022), the fixed time convergence theory is further extended to systems with time-varying coefficients. These methods demonstrate extraordinary acceleration in small-scale continuous systems, whereas their application in the deep learning field remains largely unexplored. Recently, these theoretical advances have recently been applied to machine learning and deep learning problems. Budhraja et al. (2022) introduced fixed-time stable gradient flows (FxTS-GF) for convex optimization , demonstrating how control-theoretic concepts can provide stronger convergence guarantees than classical gradient descent.

## 3 PROBLEM FORMULATION AND THEORETICAL FRAMEWORK

Consider the unconstrained optimization problem:

$$\min_{\mathbf{w} \in \mathbb{R}^d} \mathcal{L}(\mathbf{w}) \tag{3}$$

where $\mathcal{L} : \mathbb{R}^d \to \mathbb{R}$ is continuously differentiable with global minimum $\mathbf{w}^*$ and optimal value $\mathcal{L}^* = \mathcal{L}(\mathbf{w}^*)$. Classical approaches to solving problem (3) include first-order and second-order methods. Gradient descent employs the update rule $\mathbf{w}_{k+1} = \mathbf{w}_k - \eta \nabla \mathcal{L}(\mathbf{w}_k)$ and offers simplicity but exhibits slow linear convergence rates under strong convexity. The discrete-time algorithms can be interpreted as dynamical system, while its continuous-time counterparts, derived by considering infinitesimal step sizes, take the form of differential equations, i.e.

$$\dot{\mathbf{w}}(t) = -\nabla \mathcal{L}(\mathbf{w}(t)). \tag{4}$$

Analyzing the continuous-time system can provide valuable theoretical insights, such as stability properties and convergence rates, offering a complementary perspective to discrete optimization analysis. Both classical discrete methods and their continuous-time counterparts provide only asymptotic convergence guarantees: the objective function approaches the optimum as time or iterations tend to infinity, but never reaches it exactly in finite time. Recent advances in optimization theory have introduced stronger convergence concepts that go beyond asymptotic guarantees. These developments draw from the stability theory of dynamical systems to provide finite-time and fixed-time convergence guarantees, with definitions given as Definition 1, 2.

**Definition 1** (Finite-Time Stability Bhat & Bernstein (2000)). *A dynamical system $\dot{x} = f(x)$ with equilibrium at $x = 0$ is finite-time stable if there exists a settling time function $T : \mathbb{R}^n \to \mathbb{R}_+$ such that for any initial condition $x(0) = x_0$, the solution satisfies $x(t) = 0$ for all $t \geq T(x_0)$.*

**Definition 2** (Fixed-Time Stability Polyakov (2011)). *A finite-time stable system is fixed-time stable if the settling time function $T(x_0)$ is globally bounded: $\sup_{x_0 \in \mathbb{R}^n} T(x_0) < \infty$.*

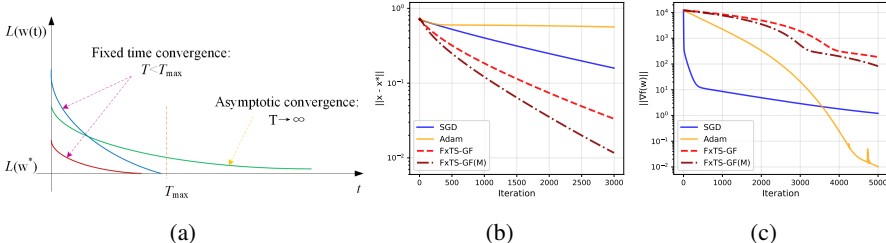

(a)         (b)         (c)

Figure 1: (a) Fixed time convergence vs asymptotic convergence, (b) Convergence result for minimizing Rosenbrock function with various optimization methods, where fixed time methods converge faster than SGD and Adam, (c) Convergence result for small-scale quadratic problem with various optimization methods, where Adam converge faster than fixed time methods.

The key advantage of fixed-time stability is that convergence time is independent of initial conditions, providing stronger guarantees than classical finite-time stability, as demonstrated in Figure 1a. Designing finite/fixed time gradient flow normally involves the following lemmas:

**Lemma 1** (Bhat & Bernstein (2000)). *Let $V(t)$ be absolutely continuous and satisfy $\dot{V}(t) \leq -\alpha V(t)^\gamma$ for $\alpha > 0$ and $\gamma \in (0, 1)$. Then $V(t)$ reaches zero in finite time $T^* \leq \frac{V(0)^{1-\gamma}}{\alpha(1-\gamma)}$.*

**Lemma 2** ( Polyakov (2011)). *Consider a Lyapunov function $V(\mathbf{w}) \geq 0$ with $V(\mathbf{w}) = 0$ if and only if $\mathbf{w} = \mathbf{w}^*$. If there exist constants $a, b > 0$, $0 < \alpha < 1 < \beta$ such that $\dot{V} \leq -aV^\alpha - bV^\beta$, then the system is fixed-time stable with settling time bounded by $T \leq \frac{1}{a(1-\alpha)} + \frac{1}{b(\beta-1)}$.*

Followed by Lemma 1, 2, different control laws are designed to achieve the finite time or fixed time convergence. However, existing finite/fixed-time optimization methods, including FxTS-GF(M) (shown in (1)), encounter fundamental scalability limitations due to dimensional coupling that severely impede their adoption in large-scale machine learning. To demonstrate this property, we adopt the set from (Budhraja et al., 2022) with Rosenbrock function $f(x_1, x_2) = (1 - x_1)^2 + 100(x_2 - x_1^2)^2$, and analyze the generic quadratic minimization problem, $f(x) = \frac{1}{2} w^T H w$, where $H = \text{diag}(H_1, H_2, H_3)$ is a block-diagonal, positive-definite matrix. Let $D_i = \dim(w_i)$ be the number of parameters in block $i$. Specifially, theheterogeneity by drawing eigenvalues for $H_1$, $H_2$, and $H_3$ from $\{0.1, 0.2, 1.5, 3\}$, $\{49, 50, 51, 100\}$, and $\{1000, 1100, 2001, 2005\}$, respectively, the size of each block is set to 50. Fig. 1b and 1c demonstrate that the FxTS-GF outperforms Adam and SGD on Resenbrock function but fails with a quadratic problem with heterogeneous eigenvalues.

# 4 METHODOLOGY

## 4.1 ELEMENT-WISE FINITE-TIME OPTIMIZATION FRAMEWORK

To address the dimensional coupling limitations of existing finite-time methods, we propose an element-wise approach that enables coordinate-independent convergence analysis. Our core insight is to replace global gradient norms with element-wise operations, yielding the dynamics:

$$\dot{\mathbf{w}} = -c_1 \, \text{sign}(\mathbf{g}) \odot |\mathbf{g}|^{p_2} - c_2 \, \text{sign}(\mathbf{g}) \odot |\mathbf{g}|^{2-p_1}, \tag{5}$$

where $\mathbf{g} = \nabla \mathcal{L}(\mathbf{w})$, constants $c_1, c_2 > 0$, exponents $p_2 \in (0, 1)$, $p_1 \in (0, 2)$, and $\odot$ denotes element-wise multiplication. This formulation eliminates the global coupling present in methods like FxTS-GF while preserving the dual-power structure essential for finite-time convergence.

The key theoretical advantage lies in the coordinate-wise nature of the dynamics: each parameter $w_i$ evolves according to its local gradient information $g_i$, avoiding the dimensional scaling issues that plague globally-coupled approaches.

**Assumption 1.** *The objective function $\mathcal{L}(\mathbf{w})$ satisfies: (1) L-smoothness: $\|\nabla\mathcal{L}(\mathbf{w}) - \nabla\mathcal{L}(\mathbf{w}')\| \leq L\|\mathbf{w} - \mathbf{w}'\|$; (2) Polyak-Łojasiewicz condition: $\|\nabla\mathcal{L}(\mathbf{w})\|^2 \geq 2\mu(\mathcal{L}(\mathbf{w}) - \mathcal{L}^*)$ for some $\mu > 0$; (3) bounded gradient, i.e. $\|\mathbf{g}\|_\infty = G$*

## 4.2 MAIN CONVERGENCE RESULT

We now formalize the convergence guarantees of the proposed continuous-time dynamics. The following theorem unifies both finite-time and fixed-time convergence regimes under a single parameterization.

**Theorem 1.** *Consider the continuous-time dynamics equation 5 with parameters satisfying $p_2 \in (0,1)$, $p_1 < 2$, and $c_1, c_2 > 0$. Under Assumption 1: (i) Finite-time convergence: When $p_1 < 2$, there exists a finite time $T^* < \infty$ such that $\mathcal{L}(w(t)) = \mathcal{L}^*$ for all $t \geq T^*$. The convergence time is bounded by: $T^* \leq \frac{(\mathcal{L}(\mathbf{w}_0) - \mathcal{L}^*)^{1-\gamma_1}}{c_1 d^{-p_2}(2\mu)^{(1+p_2)/2}(1-\gamma_1)} + \frac{(\mathcal{L}(\mathbf{w}_0) - \mathcal{L}^*)^{1-\gamma_2}}{c_2 d^{p_1-2}(2\mu)^{(3-p_1)/2}(1-\gamma_2)}$ where $\gamma_1 = \frac{1+p_2}{2} \in (0.5, 1)$, $\gamma_2 = \frac{3-p_1}{2}$. (ii) Fixed-time convergence: When $p_1 < 1$, we have $\gamma_2 = \frac{3-p_1}{2} > 1$, and the convergence time is further bounded by a constant independent of initial conditions: $T^* \leq T_{\max} = \frac{1}{\alpha_2(\gamma_2-1)} + \frac{1}{\alpha_1(1-\gamma_1)}$, where $\alpha_1 = c_1 d^{-\frac{1-p_2}{2}}(2\mu)^{\frac{1+p_2}{2}}$ and $\alpha_2 = c_2 d^{p_1-2}(2\mu)^{\frac{3-p_1}{2}}$ are positive constants determined by problem parameters.*

*Proof.* The Proof details are shown in Appendix. C $\qquad\square$

**Remark 1.** *The proof reveals distinct convergence phases depending on $p_1$: i)finite-time ($1 \leq p_1 < 2$): Both $\gamma_1, \gamma_2 < 1$, convergence time depends on initial conditions. (ii)Fixed-time ($p_1 < 1$): $\gamma_2 > 1$ dominates as $V(t) \to 0$, ensuring convergence time bounds independent of initial conditions This characterization highlights the practical flexibility of the dynamics: by tuning $p_1$, practitioners can trade off between rapid convergence with initial-condition dependence and guaranteed uniform convergence time.*

By using the explicit Euler discretization scheme, it yeilds:

$$\mathbf{w}_{k+1} = \mathbf{w}_k - \eta\left[c_1\,\text{sign}(\mathbf{g}_k) \odot |\mathbf{g}_k|^{p_2} + c_2\,\text{sign}(\mathbf{g}_k) \odot |\mathbf{g}_k|^{2-p_1}\right], \tag{6}$$

where $\eta$ is the stepsize, and $\odot$ denotes element-wise multiplication. Therefore, the algorithm is shown in Algorithm 1.

---

**Algorithm 1** Element-wise Finite/Fixed-Time (EFT) Convegence algorithm

---

**Require:** Parameters $\beta \in [0,1)$, $c_1, c_2 > 0$, $p_1 < 2$, $p_2 \in (0,1)$, learning rate $\eta > 0$.
**Require:** Initial weights $\mathbf{w}_0$,
1: **for** $k = 0, 1, 2, \dots$ **do**
2:   Compute gradients: $\mathbf{g}_k = \nabla\mathcal{L}(\mathbf{w}_k)$
3:   Compute EFT forces: $F_k = c_1\,\text{sign}(\mathbf{g}_k) \odot |\mathbf{g}_k|^{p_2} + c_2\,\text{sign}(\mathbf{g}_k) \odot |\mathbf{g}_k|^{2-p_1}$
4:   Update weights: $\mathbf{w}_{k+1} = \mathbf{w}_k - \eta F_k$
5: **end for**

---

## 4.3 MOMENTUM-ENHANCED FINITE-TIME DYNAMICS

Modern deep learning optimization faces two critical challenges: stochastic variance from mini-batch gradients and heterogeneous curvature across parameter space. We address these through two complementary momentum mechanisms that preserve our finite-time convergence guarantees while offering distinct computational advantages.

**Element-wise Finite/Fixed-Time with Momentum (EFToM)**  For variance reduction in stochastic settings, we integrate exponential moving averages:

$$\mathbf{m}_k = \beta_1\mathbf{m}_{k-1} + (1-\beta_1)\mathbf{g}_k, \tag{7a}$$

$$\mathbf{w}_{k+1} = \mathbf{w}_k - \eta\left[c_1\,\text{sign}(\mathbf{m}_k) \odot |\mathbf{m}_k|^{p_2} + c_2\,\text{sign}(\mathbf{m}_k) \odot |\mathbf{m}_k|^{2-p_1}\right]. \tag{7b}$$

where $\mathbf{m} \in \mathbb{R}^d$ is the momentum vector. The continuous-time analysis reveals that EFToM achieves momentum tracking through the system:

$$\dot{\mathbf{w}} = -c_1 \operatorname{sign}(\mathbf{m}) \odot |\mathbf{m}|^{p_2} - c_2 \operatorname{sign}(\mathbf{m}) \odot |\mathbf{m}|^{2-p_1}, \quad \dot{\mathbf{m}} = -\lambda (\mathbf{m} - \mathbf{g}), \qquad (8a)$$

where $\lambda > 0$ controls the momentum convergence rate.

**Element-wise Finite/Fixed-Time with Polyak Momentum (PEFToM)** While EFToM excels in noisy environments, certain optimization landscapes benefit from accumulated gradient history. Polyak momentum provides this through its natural continuous representation. The discrete update $\mathbf{v}_k = \beta \mathbf{v}_{k-1} + \mathbf{g}_k$ can be unrolled as:

$$\mathbf{v}_k = \sum_{j=0}^{k} \beta^j \mathbf{g}_{k-j} = \mathbf{g}_k + \beta \mathbf{g}_{k-1} + \beta^2 \mathbf{g}_{k-2} + \dots \qquad (9)$$

This discrete convolution has a natural continuous analog through the integral representation $\mathbf{v}(t) = \int_{-\infty}^{t} e^{-\gamma(t-s)} \mathbf{g}(s)\, ds$, where $\gamma > 0$ controls the memory depth. Taking the time derivative yields:

$$\frac{d\mathbf{v}}{dt} = \mathbf{g}(t) - \gamma \int_{-\infty}^{t} e^{-\gamma(t-s)} \mathbf{g}(s)\, ds = \mathbf{g}(t) - \gamma \mathbf{v}(t) \qquad (10)$$

Unlike EFToM's instantaneous gradient tracking, PEFToM accumulates the complete gradient history, making it particularly effective for optimization problems with consistent gradient directions. The complete PEFToM system becomes:

$$\dot{\mathbf{v}} = \mathbf{g} - \gamma \mathbf{v}, \quad \dot{\mathbf{w}} = -c_1 \operatorname{sign}(\mathbf{v}) \odot |\mathbf{v}|^{p_2} - c_2 \operatorname{sign}(\mathbf{v}) \odot |\mathbf{v}|^{2-p_1}. \qquad (11a)$$

**Theorem 2** (EFToM Finite-/Fixed-Time Convergence). *Consider the EFToM dynamics equation 8 with parameters $p_2 \in (0,1)$, $0 < p_1 < 2$, and $c_1, c_2, \lambda > 0$. Under Assumption 1, choose $\lambda \geq \lambda_\star$, where $\lambda_\star := \max\left\{ \left(\frac{4K_1}{c_1}\right)^{1-\theta_1}, \left(\frac{4K_2}{c_2}\right)^{1-\theta_2} \right\}$ with $\theta_1 = \frac{2p_2}{1+p_2}$, $\theta_2 = \frac{2(2-p_1)}{3-p_1}$. Define the exponents $\alpha := \frac{1+p_2}{2} \in (\frac{1}{2}, 1)$, $\beta := \frac{3-p_1}{2}$, and constants $\hat{a} := \frac{1}{2} c_1 d^{-\frac{1-p_2}{2}} (2\mu)^\alpha$, $\hat{b} := \frac{1}{2} c_2 d^{-\frac{1-p_1}{2}} (2\mu)^\beta$. Then the following convergence guarantees hold: **(i) Finite-time convergence** ($1 \leq p_1 < 2$, equivalently $\beta \leq 1$): For any initial state $(\mathbf{w}(0), \mathbf{m}(0))$, the convergence time satisfies $T \leq \frac{V_{tot}(0)^{1-\alpha}}{\hat{a}(1-\alpha)} + \frac{V_{tot}(0)^{1-\beta}}{\hat{b}(1-\beta)}$, where $V_{tot}(0) := \mathcal{L}(\mathbf{w}(0)) - \mathcal{L}^* + \frac{c_1}{2} \|\mathbf{m}(0) - \mathbf{g}(0)\|^2$. **(ii) Fixed-time convergence** If $p_1 < 1$, every trajectory reaches the global optimum $(\mathbf{w}^*, \mathbf{0})$ within time $T_{\max} = \frac{1}{\hat{a}(1-\alpha)} + \frac{1}{\hat{b}(\beta-1)}$.*

**Theorem 3** (PEFToM Finite-/Fixed-Time Convergence). *Consider the PEFToM dynamics equation 11 with the same parameter conditions as Theorem 2. Under Assumption 1, the convergence guarantees are analogous to EFToM, with constants: $\hat{a} := \frac{\gamma}{2} c_1 d^{-\frac{1-p_2}{2}} (2\mu)^\alpha$, $\hat{b} := \frac{\gamma}{2} c_2 d^{-\frac{1-p_1}{2}} (2\mu)^\beta$.*

*Proof.* See Appendix D, E for the complete proof. $\qquad \square$

**Remark 2** (Momentum Mechanism Selection Guide). *The theoretical analysis reveals distinct algorithmic characteristics: (i) EFToM: Convergence rates independent of momentum parameter $\lambda$, providing robustness to hyperparameter selection. The parameter $\lambda$ only affects the admissibility threshold $\lambda_\star$, making hyperparameter tuning less critical. (ii) PEFToM: Convergence constants scale linearly with damping parameter $\gamma$, offering direct control over convergence acceleration. However, this requires careful $\gamma$ selection to balance convergence speed with numerical stability.*

Discretizing equation 11 using explicit Euler methods, it yields

$$\mathbf{v}_{k+1} = \beta \mathbf{v}_k + \mathbf{g}_k, \qquad (12a)$$

$$\mathbf{w}_{k+1} = \mathbf{w}_k - \eta \left( c_1 \operatorname{sign}(\mathbf{v}_{k+1}) \odot |\mathbf{v}_{k+1}|^{p_2} + c_2 \operatorname{sign}(\mathbf{v}_{k+1}) \odot |\mathbf{v}_{k+1}|^{2-p_1} \right), \qquad (12b)$$

Based on (7) and (12), the EFToM and PEFToM are sumarized in Algorithm 2 and 3, respectively.

**Algorithm 2** Element-wise Finite-Time with Momentum (EFToM)

**Require:** Parameters $\beta \in [0,1)$, $c_1, c_2 \geq 0$, $p_1 < 2$, $p_2 \in (0,1)$, learning rate $\eta > 0$
**Require:** Initial weights $\mathbf{w}0$, initial momentum $v_0 = 0$
1: **for** $k = 0, 1, 2, \ldots$ **do**
2: $\quad \mathbf{g}_k = \nabla \mathcal{L}(\mathbf{w}_k)$
3: $\quad \mathbf{m}_{k+1} = \beta \mathbf{m}_k + (1-\beta)\mathbf{g}_k$.
4: $\quad F_k = c_1 \operatorname{sign}(\mathbf{m}_{k+1}) \odot |\mathbf{m}_{k+1}|^{p_2} + c_2 \operatorname{sign}(\mathbf{m}_{k+1}) \odot |\mathbf{m}_{k+1}|^{2-p_1}$
5: $\quad$ Update weights: $\mathbf{w}_{k+1} = \mathbf{w}_k - \eta F_k$
6: **end for**

**Algorithm 3** Element-wise Finite-Time with Polyak Momentum (PEFToM)

**Require:** Parameters $\beta \in [0,1)$, $c_1, c_2 \geq 0$, $p_1 < 2$, $p_2 \in (0,1)$, learning rate $\eta > 0$
**Require:** Initial weights $\mathbf{w}0$, initial momentum $v_0 = 0$
1: **for** $k = 0, 1, 2, \ldots$ **do**
2: $\quad \mathbf{g}_k = \nabla \mathcal{L}(\mathbf{w_k})$
3: $\quad \mathbf{v}_{k+1} = \beta \mathbf{v}_k + \mathbf{g}_k$.
4: $\quad F_k = c_1 \operatorname{sign}(\mathbf{v}_{k+1}) \odot |\mathbf{v}_{k+1}|^{p_2} + c_2 \operatorname{sign}(\mathbf{v}_k) \odot |\mathbf{v}_k|^{2-p_1}$
5: $\quad$ Update weights: $\mathbf{w}_{k+1} = \mathbf{w}_k - \eta F_k$
6: **end for**

### 4.4 Unified Framework: SignSGD and Signum as Special Cases

Our element-wise finite-time framework provides a unified theoretical foundation for sign-based optimization methods. For instance, SignSGD corresponds to the limiting behavior when $p_1 \to 2$ and $p_2 \to 0$ in our EFT dynamics : $\lim_{p_1 \to 2, p_2 \to 0} \dot{\mathbf{w}} = -(c_1 + c_2)\operatorname{sign}(\mathbf{g})$. Under Euler discretization with step size $\eta$, this yields: $\mathbf{w}_{k+1} = \mathbf{w}_k - \eta(c_1 + c_2)\operatorname{sign}(\mathbf{g}_k)$. Similarly, Signum (Bernstein et al., 2018) emerges from our EFToM under the same parameter limits. When $p_1 \to 2$ and $p_2 \to 0$:

$$\mathbf{m}_k = \beta_1 \mathbf{m}_{k-1} + (1-\beta_1)\mathbf{g}_k, \qquad \mathbf{w}_{k+1} = \mathbf{w}_k - \eta(c_1+c_2)\operatorname{sign}(\mathbf{m}_k). \tag{13}$$

For theoretical rigor, we have provided the finite-time convergence of EFT with $p_1 = 2, p_2 = 0$, with a convergence time $T \leq \frac{\sqrt{2(L(w_0)-L^*)}}{(c_1+c_2)\sqrt{\mu}}$, as shown in Appendix F.

### 4.5 Convergence Analysis for Discrete Algorithms

While the continuous-time analysis establishes finite/fixed-time convergence guarantees, practical implementations require discrete-time algorithms via numerical discretization. We now provide convergence analysis for the discrete algorithms to bridge the gap between theory and practice, where technique in (Zhou et al., 2020) and (Bernstein et al., 2018) are used.

**Theorem 4** (Convergence of Discrete EFT). *Under Assumptions 1, let the discrete EFT algorithm run for $K$ iterations with step size $\eta \leq \frac{1}{LC_F}$ where $C_F = 2d(c_1^2 G^{2p_2} + c_2^2 G^{2(2-p_1)})$. Then $\min_{k \in [K]} \|\mathbf{g}_k\|_{1+p_2}^{1+p_2} \leq \frac{\mathcal{L}(\mathbf{w}_0)-\mathcal{L}^*}{c_1 \eta K} + \frac{L\eta C_F}{2c_1}$. Specifically, if we chose $\eta = \frac{1}{\sqrt{KLC_F}}$, it yeilds $\min_{k \in [K]} \|\mathbf{g}_k\|_{1+p_2}^{1+p_2} \leq \frac{\mathcal{L}(\mathbf{w}_0)-\mathcal{L}^*}{c_1\sqrt{LC_F}\sqrt{K}} + \frac{\sqrt{LC_F}}{2c_1\sqrt{K}} = O(K^{-1/2})$.*

*Proof.* See Appendix G.1 for details. □

The Convergence analysis of EFToM (Algorithm 2) and PEFToM (Algorithm 3 ) can be found in Appendix G.2, G.3.

## 5 Numerical Experiments

We first empirically validate the proposed Element-wise Finite-Time Optimization (EFT) framework, including its variants EFToM and PEFToM on a strong convex regularied logistic regression problem, and then apply it on standard benchmarks across two domains. For computer vision, we evaluate performance on image classification tasks with CIFAR-10, CIFAR-100 (Krizhevsky, 2009) and tiny ImageNet. For natural language processing task, we pretrain the llama-60m on C4 (Muennighoff et al., 2023) subset. We compare our methods against a comprehensive suite of optimization baselines to demonstrate their effectiveness. All experiments are implemented in PyTorch and executed on a single NVIDIA RTX 4090 GPU with 24 GB memory.

## 5.1 SYNTHETIC VERIFICATION OF FINITE-TIME CONVERGENCE

We empirically validate the finite-time convergence property on a strongly convex regularized logistic regression problem: $\min_{\mathbf{w} \in \mathbb{R}^d} L(\mathbf{w}) = \frac{1}{n} \sum_{i=1}^{n} \log\left(1 + \exp(-y_i \mathbf{w}^\top \mathbf{x}_i)\right) + \frac{\lambda}{2} \|\mathbf{w}\|^2$, with $n = 200$, $d = 20$, $\lambda = 0.1$. To enable rigorous comparison between discrete baselines and our continuous-time dynamics, we align the x-axis using continuous time $t = k \cdot \eta$, with $\eta = 0.0005, c_1 = c_2 = 1$.

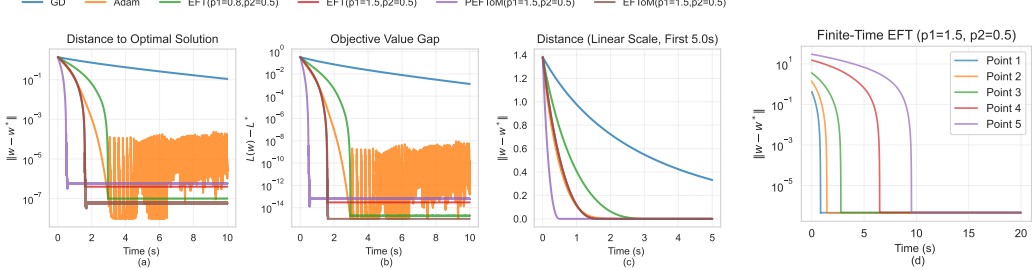

Figure 2: Finite-Time vs. Asymptotic Convergence. (a-b) Log-scale metrics over 10 seconds: GD exhibits exponential decay, Adam oscillates near convergence, while EFT variants demonstrate *vertical drops* to machine precision followed by exact stability. (c) Linear-scale early dynamics (d) FET convergence with different initial value.

**Results.** Figure 2 and reveal fundamental distinctions between asymptotic and finite-time convergence behaviors: EFT variants exhibit characteristic *vertical drops* to $\sim 10^{-6}$ within finite time, then maintain exact stability. GD shows classic exponential decay but never settles exactly after 10 seconds. Adam converges faster than GD but oscillates around convergence. Table 1 listed the convergence time of different algorithms. In addition, the convergence time as well as the theoretic bound of EFT with different initial values are listed to demonstrate that the proposed method can converge within the theoretic bounds. Notably, the real convergence time of EFT is far less than the theoretical bound, which proves the finite-time convergence of EFT.

Table 1: Quantitative Convergence Metrics. All values measured at $t = 10$s. Convergence time $T^*$ is defined as the first time when $\|\mathbf{w}(t) - \mathbf{w}^*\| < 10^{-6}$. N/A indicates the threshold was not reached within 10 seconds.

| Optimizer | Conv. Time (s) | EFT ($p_1 = 1.5, p_2 = 0.5$) | | |
|---|---|---|---|---|
| | | $\|\mathbf{w}_0 - \mathbf{w}^*\|$ | **Conv.Time (s)** | **Upper bound (s)** |
| GD | N/A | 0.4254 | 0.842 | 47.1 |
| Adam | 2.81 | 1.3864 | 1.444 | 89.64 |
| EFT ($p_1$=0.8, $p_2$=0.5) | 2.95 | 3.58 | 2.783 | 133.10 |
| EFT ($p_1$=1.5, $p_2$=0.5) | 1.61 | 15.34 | 6.491 | 247.55 |
| EFToM ($p_1$=1.5, $p_2$=0.5) | 1.60 | 30.13 | 9.53 | 326.16 |
| PEFToM ($p_1$=1.5, $p_2$=0.5) | **0.54** | | | |

## 5.2 IMAGE CLASSIFICATION ON CIFAR AND IMAGENET DATASETS

**Setup** We evaluate on CIFAR-10 and CIFAR-100 using three CNN architectures: VGG-11, ResNet-34, and DenseNet-121. Training proceeds for 200 epochs with batch size 128. For tiny ImageNet, the ViT model is used, where the training proceeds with 90 epochs with batch size of 256. Baseline optimizers include SGD, SGD with momentum (SGDM), AdamW, AdaBelief, SignSGD, Signum, Lion, and FxTS-GF(M). For hyperparameters, we follow the setting in Zhuang et al. (2020), with details of the hyperparameters used in the experiment given in Appendix H.

**Performance Analysis** Table 2 presents final accuracies across architectures and datasets, while Figure 3 and Figure 5.2 present the test accuracy of various optimizers at different training stage. The proposed framework achieves consistent improvements, with PEFToM reaching 95.17% on CIFAR-10 ResNet-34 and 79.6% on CIFAR-100 DenseNet-121. These results represent gains of

Table 2: Test accuracy (%) on CIFAR-10 & CIFAR-100 at different epochs and architectures.

| | CIFAR10 | | | CIFAR100 | | | Tiny ImageNet | |
| --- | --- | --- | --- | --- | --- | --- | --- | --- |
| | VGG-11 | ResNet-34 | DenseNet-121 | VGG-11 | ResNet-34 | DenseNet-121 | ViT | |
| Epochs | 200 | 200 | 200 | 200 | 200 | 200 | 30 | 90 |
| SGD | 86.97 | 92.69 | 91.77 | 62.86 | 75.95 | 77.92 | 16.99 | 28.29 |
| SignSGD | 48.13 | 92.86 | 92.9 | 29.59 | 67.38 | 68.88 | 32.3 | 41.88 |
| **EFT** | 88.78 | 94 | 94.33 | 64.0 | 75.74 | 77.72 | 28.51 | 42.32 |
| FxTS-GF(M) | 87.91 | 93.44 | 94.58 | 63.67 | 73.58 | 74.65 | 14.05 | 26.18 |
| **EFToM** | 89.01 | 94.33 | 94.64 | 65.39 | 75.78 | 77.31 | 39.34 | 45.51 |
| Signum | 88.04 | 94.19 | 94.64 | 58.53 | 73.51 | 75.43 | 39.8 | 45.79 |
| Lion | 87.02 | 94.25 | 94.44 | 57.21 | 72.99 | 75.23 | **42.71** | **46.76** |
| AdamW | 87.97 | 93.88 | 94.07 | 58.66 | 71.98 | 75.16 | 41.13 | 45.7 |
| AdaBelief | 87.86 | 94.26 | 94.06 | 58.37 | 72.08 | 74.01 | 41.12 | 46.41 |
| SGDM | 90.35 | 94.68 | 94.8 | 64.62 | 75.82 | 78.1 | 18.54 | 30.86 |
| **PEFToM** | **91.02** | **95.17** | **95.62** | **68.05** | **77.34** | **79.6** | 21.20 | 31.69 |

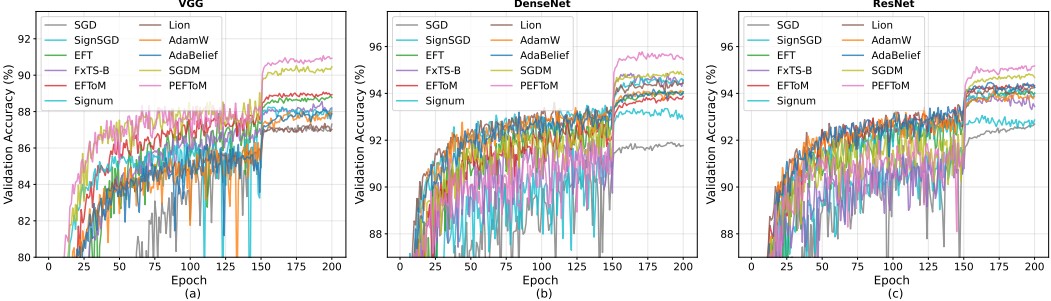

Figure 3: Test accuracy for CIFAR10 with different optimizers and models: (a) VGG-11, (b) DenseNet-121, (c) ResNet-34

.

+0.49% and +1.5% over SGD with momentum. The progression from EFT → EFToM → PEFToM demonstrates systematic enhancement through momentum integration. On CIFAR-100 ResNet-34: EFT (75.74%) → EFToM (75.78%) → PEFToM (77.34%) shows incremental but meaningful improvements from each algorithmic component. Performance advantages become more pronounced on the challenging CIFAR-100 dataset. PEFToM surpasses modern adaptive methods: +5.36% over AdamW (77.34% vs 71.98%) and +5.26% over AdaBelief (77.34% vs 72.08%) on ResNet-34. This pattern suggests finite-time optimization principles provide greater benefits as task complexity increases. Our element-wise approach addresses instabilities observed in existing sign-based methods. While SignSGD deteriorates significantly on CIFAR-100 (29.59% on VGG-11), EFToM maintains robust performance across all configurations. Similarly, compared to FxTS-GF(M) which suffers from dimensional coupling, our method shows superior consistency on CIFAR-100 ResNet-34. On tiny ImageNet with Vision Transformer, EFT achieves 42.32% accuracy, substantially improving over SGD and SignSGD . EFToM further advances to 45.51%, competitive with adaptive methods like AdamW (45.70%) and AdaBelief (46.41%). However, PEFToM reaches 31.69%, lower than EFToM but still surpassing SGDM (30.86%). This suggests that Polyak momentum may require different hyperparameter tuning for transformer architectures , while exponential moving average momentum (EFToM) demonstrates more consistent performance across model architectures.

## 5.3 LANGUAGE MODEL PRETRAINING

We pretrain Llama-60M on C4 subset (Muennighoff et al., 2023) for 30,000 steps with batch size 16. Table 3 reports training and validation losses at key checkpoints, and the traning loss curve are demonstrated in Figure 5. EFToM achieves the lowest validation loss (3.929 at 30k steps), outperforming AdamW (4.045) and other baselines. The rapid convergence aligns with our finite-time optimization theory. While AdamW exhibits smoother training curves, EFToM reaches better final performance despite some oscillation. PEFToM shows different behavior with slower initial progress but competitive endpoints (4.594), indicating momentum variants may suit different training phases. The language modeling results validate our framework's applicability beyond computer vision. EFToM's validation performance significantly exceeds AdamW (+2.9% relative improvement) and AdaBelief (+5.5% relative improvement), demonstrating effectiveness for large-scale se-

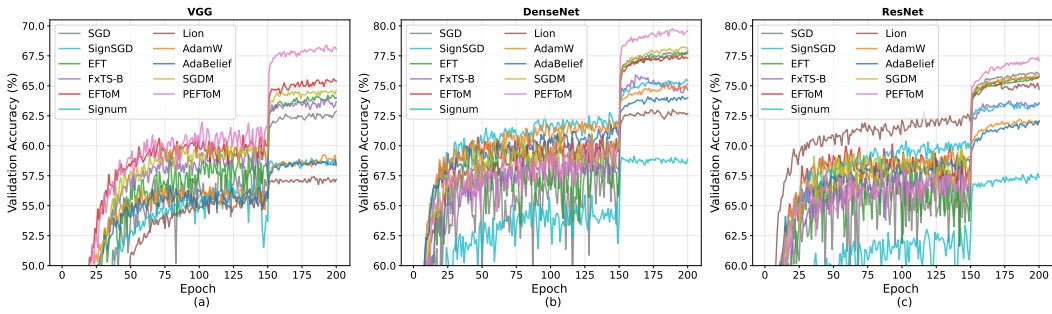

Figure 4: Test accuracy for CIFAR100 with different optimizers and models: (a) VGG-11, (b) DenseNet-121, (c) ResNet-34

.

quence modeling. , To further testify the scalability of the proposed methods, llama-350m is used, with detailed result shown in Appendix J

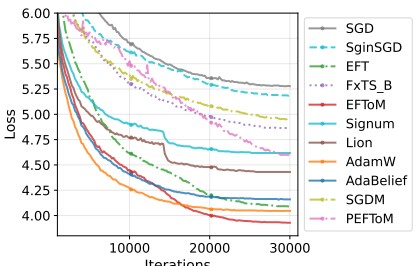

Figure 5: Test loss on C4 dataset

Table 3: Train and validation loss on C4 at different iterations with llama-60m.

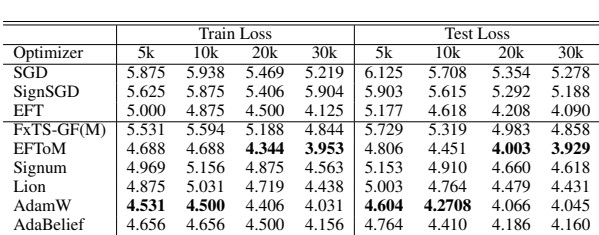

| Optimizer | Train Loss | | | | Test Loss | | | |
|---|---|---|---|---|---|---|---|---|
| | 5k | 10k | 20k | 30k | 5k | 10k | 20k | 30k |
| SGD | 5.875 | 5.938 | 5.469 | 5.219 | 6.125 | 5.708 | 5.354 | 5.278 |
| SignSGD | 5.625 | 5.875 | 5.406 | 5.904 | 5.903 | 5.615 | 5.292 | 5.188 |
| EFT | 5.000 | 4.875 | 4.500 | 4.125 | 5.177 | 4.618 | 4.208 | 4.090 |
| FxTS-GF(M) | 5.531 | 5.594 | 5.188 | 4.844 | 5.729 | 5.319 | 4.983 | 4.858 |
| EFToM | 4.688 | 4.688 | **4.344** | **3.953** | 4.806 | 4.451 | **4.003** | **3.929** |
| Signum | 4.969 | 5.156 | 4.875 | 4.563 | 5.153 | 4.910 | 4.660 | 4.618 |
| Lion | 4.875 | 5.031 | 4.719 | 4.438 | 5.003 | 4.764 | 4.479 | 4.431 |
| AdamW | **4.531** | **4.500** | 4.406 | 4.031 | **4.604** | **4.2708** | 4.066 | 4.045 |
| AdaBelief | 4.656 | 4.656 | 4.500 | 4.156 | 4.764 | 4.410 | 4.186 | 4.160 |
| SGDM | 5.469 | 5.594 | 5.188 | 4.844 | 5.705 | 5.389 | 5.076 | 4.951 |
| PEFToM | 5.563 | 5.781 | 5.125 | 4.625 | 5.809 | 5.524 | 4.934 | 4.594 |

## 5.4 DISCUSSION

Our element-wise finite-time dynamics consistently improve performance across diverse architectures while successfully integrating momentum mechanisms that preserve theoretical convergence properties. *Memory Efficiency*: Our methods reduce memory overhead by approximately 33% compared to Adam-family optimizers by requiring only first-order momentum buffers, making them suitable for large-scale model training. *Momentum Selection*: PEFToM excels on vision tasks while EFToM performs best for language modeling, indicating that momentum mechanism selection should consider task-specific optimization characteristics.

## 6 CONCLUSION

We developed an element-wise finite-time optimization framework that addresses the scalability limitations of control-theoretic methods in high-dimensional deep learning. By eliminating dimensional coupling through coordinate-independent dynamics, our approach achieves rigorous finite/fixed-time convergence guarantees while preserving the heterogeneity-aware adaptivity crucial for neural network optimization. The theoretical framework unifies disparate optimization methods under a principled foundation: SignSGD and Signum emerge as limiting cases, providing rigorous finite-time theoretical justification for their empirical success. Our momentum-enhanced variants demonstrate that classical acceleration techniques can be seamlessly integrated without compromising convergence properties. Empirical validation across computer vision and language modeling confirms both convergence acceleration and memory efficiency gains, positioning our methods as theoretically grounded yet practically viable alternatives to adaptive optimizers. This work advances the integration of control-theoretic optimization with large-scale machine learning, demonstrating that finite-time convergence principles can be effectively harnessed for practical neural network training through appropriate algorithmic adaptations.

REPRODUCIBILITY STATEMENT

To ensure reproducibility, we provide complete implementation details in Appendix H, theoretical proofs in Appendices C–E, and experimental configurations in Section 5. All datasets used (CIFAR-10/100, C4) are publicly available as described in Section 5. We will release the complete source code upon acceptance of this paper.

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

## A    LARGE LANGUAGE MODEL USAGE STATEMENT

Large Language Models (LLMs) were used in this research for the following purposes: (i) **Writing assistance:** LLMs were used to aid in polishing the manuscript, including grammar checking, sentence structure improvement, and clarity enhancement of technical explanations. (ii)**Literature review support:** LLMs assisted in discovering and organizing related work during the literature review process, helping to identify relevant papers and research directions in optimization theory and deep learning. All substantial intellectual contributions, including the element-wise finite-time optimization framework, experimental design, and analysis of results, were developed independently by the authors.

## B    FOUNDATIONS OF FINITE-TIME AND FIXED-TIME STABILITY THEORY

This appendix provides a comprehensive foundation for finite-time and fixed-time stability theory, establishing the mathematical framework underlying our element-wise optimization methods. We present detailed proofs of the fundamental lemmas and establish all necessary mathematical tools used throughout the paper.

### B.1    NOTATION AND PRELIMINARY DEFINITIONS

Throughout this appendix, we consider autonomous dynamical systems of the form:

$$\dot{\mathbf{x}}(t) = f(\mathbf{x}(t)), \quad \mathbf{x}(0) = \mathbf{x}_0 \in \mathbb{R}^n \tag{14}$$

where $f : \mathbb{R}^n \to \mathbb{R}^n$ is locally Lipschitz continuous, and $\mathbf{x} = \mathbf{0}$ is an equilibrium point (i.e., $f(\mathbf{0}) = \mathbf{0}$).

**Notation**    $\| \cdot \|$ is Euclidean norm in $\mathbb{R}^n$, $\mathcal{B}_r := \{\mathbf{x} \in \mathbb{R}^n : \|\mathbf{x}\| \leq r\}$ represents closed ball of radius $r$, $\mathcal{B}_r^o := \{\mathbf{x} \in \mathbb{R}^n : \|\mathbf{x}\| < r\}$ represents open ball of radius $r$, $V : \mathbb{R}^n \to \mathbb{R}_+$ is the Lyapunov function candidate, $\dot{V}(\mathbf{x}) := \nabla V(\mathbf{x})^T f(\mathbf{x})$ represents time derivative of $V$ along system trajectories, $\mathbb{R}_+ := [0, +\infty)$ is non-negative real numbers.

### B.2    CLASSICAL ASYMPTOTIC STABILITY VS. FINITE-TIME CONVERGENCE

**Definition 3** (Asymptotic Stability  Khalil & Grizzle (2002)). *The equilibrium* $\mathbf{x} = \mathbf{0}$ *is asymptotically stable if:*

1. *__Stability__: For any $\epsilon > 0$, there exists $\delta > 0$ such that $\|\mathbf{x}_0\| < \delta$ implies $\|\mathbf{x}(t)\| < \epsilon$ for all $t \geq 0$*

2. *__Attractivity__: There exists $\delta > 0$ such that $\|\mathbf{x}_0\| < \delta$ implies $\lim_{t\to\infty} \mathbf{x}(t) = \mathbf{0}$*

The fundamental limitation of asymptotic stability is that convergence occurs only as $t \to \infty$. In contrast, finite-time stability guarantees exact convergence in finite time, providing stronger convergence guarantees essential for time-critical applications.

### B.3    FINITE-TIME STABILITY THEORY

**Definition 4** (Finite-Time Stability Bhat & Bernstein (2000)). *The equilibrium* $\mathbf{x} = \mathbf{0}$ *is finite-time stable if:*

1. *It is asymptotically stable*

2. *For any initial condition $\mathbf{x}_0$ in a neighborhood of the origin, there exists a settling time $T(\mathbf{x}_0) < \infty$ such that $\mathbf{x}(t) = \mathbf{0}$ for all $t \geq T(\mathbf{x}_0)$*

*The function $T : \mathcal{B}_r^o \to \mathbb{R}_+$ is called the* **settling time function***.*

### B.3.1 FUNDAMENTAL LEMMA FOR FINITE-TIME CONVERGENCE

**Lemma 3** (Finite-Time Convergence via Fractional Powers Bhat & Bernstein (2000)). *Let $V(t) \geq 0$ be an absolutely continuous function satisfying the differential inequality:*

$$\dot{V}(t) \leq -\alpha V(t)^{\gamma}. \tag{15}$$

*for some constants $\alpha > 0$ and $\gamma \in (0, 1)$. Then $V(t)$ reaches zero in finite time $T^* < \infty$ given by:*

$$T^* \leq \frac{V(0)^{1-\gamma}}{\alpha(1-\gamma)}. \tag{16}$$

*Proof.* We establish this result through direct integration using separation of variables. The differential inequality equation 15 gives us:

$$\frac{dV}{dt} \leq -\alpha V^{\gamma}. \tag{17}$$

For $V(t) > 0$ (which holds for $t < T^*$), we can separate variables:

$$\frac{dV}{V^{\gamma}} \leq -\alpha dt. \tag{18}$$

Integrating both sides from $0$ to $t$ (where $t < T^*$):

$$\int_{V(0)}^{V(t)} V^{-\gamma} dV \leq -\alpha \int_0^t ds = -\alpha t. \tag{19}$$

The left-hand side evaluates to:

$$\int_{V(0)}^{V(t)} V^{-\gamma} dV = \left[\frac{V^{1-\gamma}}{1-\gamma}\right]_{V(0)}^{V(t)} = \frac{V(t)^{1-\gamma} - V(0)^{1-\gamma}}{1-\gamma}. \tag{20}$$

Since $\gamma \in (0, 1)$, we have $1 - \gamma > 0$, thus:

$$\frac{V(t)^{1-\gamma} - V(0)^{1-\gamma}}{1-\gamma} \leq -\alpha t. \tag{21}$$

Rearranging:

$$V(t)^{1-\gamma} \leq V(0)^{1-\gamma} - \alpha(1-\gamma)t. \tag{22}$$

The right-hand side becomes zero when: $t = T^* := \frac{V(0)^{1-\gamma}}{\alpha(1-\gamma)}$. For $t \geq T^*$, we must have $V(t) = 0$ (since $V(t) \geq 0$ by assumption), establishing finite-time convergence. $\square$

**Remark 3.** *The crucial insight is that the fractional power $\gamma < 1$ creates a "super-linear" decay rate near the equilibrium. As $V(t) \to 0$, the term $V(t)^{\gamma}$ decays more slowly than $V(t)$, leading to finite-time convergence rather than asymptotic approach.*

### B.4 FIXED-TIME STABILITY THEORY

The limitation of finite-time stability is that the settling time $T(\mathbf{x}_0)$ may grow unboundedly as $\|\mathbf{x}_0\| \to \infty$. Fixed-time stability addresses this by providing uniform bounds independent of initial conditions.

**Definition 5** (Fixed-Time Stability Polyakov (2011)). *A finite-time stable equilibrium $\mathbf{x} = \mathbf{0}$ is called fixed-time stable if the settling time function $T(\mathbf{x}_0)$ is globally bounded:* $\sup_{\mathbf{x}_0 \in \mathbb{R}^n} T(\mathbf{x}_0) < \infty$.

### B.4.1 DUAL-POWER LEMMA FOR FIXED-TIME CONVERGENCE

**Lemma 4** (Fixed-Time Convergence via Dual Powers  Polyakov (2011))**.** *Let $V(t) \geq 0$ be an absolutely continuous function satisfying:*

$$\dot{V}(t) \leq -aV(t)^\alpha - bV(t)^\beta \tag{23}$$

*for constants $a, b > 0$, $0 < \alpha < 1 < \beta$. Then $V(t)$ reaches zero in fixed time bounded by:*

$$T_{\max} = \frac{1}{a(1-\alpha)} + \frac{1}{b(\beta-1)}. \tag{24}$$

*Proof.* We analyze the convergence behavior in two distinct phases based on the magnitude of $V(t)$.

**Phase 1** ($V(t) \geq 1$): When $V(t) \geq 1$, since $\beta > \alpha$, we have $V(t)^\beta \geq V(t)^\alpha$. The differential inequality equation 23 becomes:

$$\dot{V} \leq -aV^\alpha - bV^\beta \leq -(a+b)V^\beta. \tag{25}$$

Applying the separation of variables technique:

$$\frac{dV}{V^\beta} \leq -(a+b)dt. \tag{26}$$

Integrating from $V(0)$ to 1 (assuming $V(0) > 1$):

$$\int_{V(0)}^1 V^{-\beta}dV \leq -(a+b)T_1. \tag{27}$$

Evaluating the integral:

$$\left[\frac{V^{1-\beta}}{1-\beta}\right]_{V(0)}^1 = \frac{1 - V(0)^{1-\beta}}{1-\beta} \leq -(a+b)T_1. \tag{28}$$

Since $\beta > 1$, we have $1 - \beta < 0$, therefore:

$$\frac{V(0)^{1-\beta} - 1}{\beta - 1} \leq (a+b)T_1. \tag{29}$$

This gives us:

$$T_1 \leq \frac{V(0)^{1-\beta} - 1}{(a+b)(\beta-1)} \leq \frac{1}{b(\beta-1)}. \tag{30}$$

**Phase 2** ($V(t) \leq 1$): When $V(t) \leq 1$, since $\alpha < 1$, we have $V(t)^\alpha \geq V(t)^\beta$. The differential inequality becomes:

$$\dot{V} \leq -aV^\alpha - bV^\beta \leq -(a+b)V^\alpha. \tag{31}$$

Following similar integration:

$$T_2 \leq \frac{1^{1-\alpha}}{(a+b)(1-\alpha)} = \frac{1}{a(1-\alpha)}. \tag{32}$$

**Total convergence time**: The total time to reach $V(t) = 0$ is:

$$T_{\max} = T_1 + T_2 \leq \frac{1}{b(\beta-1)} + \frac{1}{a(1-\alpha)}. \tag{33}$$

Importantly, this bound is independent of the initial condition $V(0)$. $\qquad\square$

**Remark 4.** *The dual-power structure in equation 23 ensures optimal convergence characteristics:*

- *For large $V(t)$: the $V^\beta$ term (with $\beta > 1$) dominates, providing rapid initial convergence*

- *For small $V(t)$: the $V^\alpha$ term (with $\alpha < 1$) dominates, ensuring finite-time convergence to zero*

*This mechanism guarantees both fast convergence and uniform settling time bounds.*

### B.5 Essential Mathematical Tools

This section establishes the key mathematical inequalities and lemmas used throughout our proofs.

**Lemma 5** (Young's Inequality (Hardy et al., 1952, Chap. I)). *Let $a, b \geq 0$ and let $p, q > 1$ be Hölder conjugates, i.e. $\frac{1}{p} + \frac{1}{q} = 1$. Then $ab \leq \frac{a^p}{p} + \frac{b^q}{q}$. More generally, for any $\varepsilon > 0$ one has $ab \leq \frac{\varepsilon^p}{p} a^p + \frac{\varepsilon^{-q}}{q} b^q$, which is often used to split mixed terms into pure powers.*

**Lemma 6** (Hardy et al. (1952)). *For any $x_1, \ldots, x_m \in \mathbb{R}$ and $q > 0$:*

1. *If $0 < q \leq 1$: $\left( \sum_{i=1}^{m} |x_i| \right)^q \leq \sum_{i=1}^{m} |x_i|^q$*

2. *If $q > 1$: $\left( \sum_{i=1}^{m} |x_i| \right)^q \leq m^{q-1} \sum_{i=1}^{m} |x_i|^q$*

Throughout the appendix every vector operation ($|\cdot|^q$, *sign*, $\odot$) is taken *element-wise*.

## C Corrected Proof of Element-wise Finite-Time Convergence

### C.1 Main Theorem

*Proof.* Recall the element-wise dynamics:

$$\dot{\mathbf{w}} = -c_1 \operatorname{sign}(\mathbf{g}) \odot |\mathbf{g}|^{p_2} - c_2 \operatorname{sign}(\mathbf{g}) \odot |\mathbf{g}|^{2-p_1} \tag{34}$$

where $c_1, c_2 > 0$, $p_2 \in (0, 1)$, $p_1 \in (0, 3)$, and $\mathbf{g} = \nabla \mathcal{L}(\mathbf{w})$. Consider the Lyapunov function $V(t) = \mathcal{L}(\mathbf{w}(t)) - \mathcal{L}^*$, it yeilds:

$$\dot{V}(t) = \sum_{i=1}^{d} g_i \dot{w}_i = -c_1 \sum_{i=1}^{d} g_i \cdot \operatorname{sign}(g_i)|g_i|^{p_2} - c_2 \sum_{i=1}^{d} g_i \cdot \operatorname{sign}(g_i)|g_i|^{2-p_1} \tag{35}$$

$$= -c_1 \sum_{i=1}^{d} |g_i|^{1+p_2} - c_2 \sum_{i=1}^{d} |g_i|^{3-p_1}. \tag{36}$$

First, we bound $\sum_{i=1}^{d} |g_i|^{1+p_2}$. Since $p_2 \in (0, 1)$, we have $1 + p_2 \in (1, 2)$. Applying Lemma 6 with $q = 1 + p_2 > 1$:

$$\left( \sum_{i=1}^{d} |g_i| \right)^{1+p_2} \leq d^{p_2} \sum_{i=1}^{d} |g_i|^{1+p_2}. \tag{37}$$

Rearranging:

$$\sum_{i=1}^{d} |g_i|^{1+p_2} \geq d^{-p_2} \left( \sum_{i=1}^{d} |g_i| \right)^{1+p_2}. \tag{38}$$

Next, we bound $\sum_{i=1}^{d} |g_i|^{3-p_1}$. Since $p_1 < 2$, then $3 - p_1 > 1$. Using Lemma 6 with $q > 1$:

$$\sum_{i=1}^{d} |g_i|^{3-p_1} \geq d^{-(3-p_1-1)} \left( \sum_{i=1}^{d} |g_i| \right)^{3-p_1} = d^{p_1-2} \left( \sum_{i=1}^{d} |g_i| \right)^{3-p_1}. \tag{39}$$

Using the fundamental inequality $\sum_{i=1}^{d} |g_i| \geq \|\mathbf{g}\|_2$ and the PL condition $\|\mathbf{g}\|_2^2 \geq 2\mu V$:

$$\sum_{i=1}^{d} |g_i| \geq \|\mathbf{g}\|_2 \geq \sqrt{2\mu V}. \tag{40}$$

Combining the above results, it yeilds:

$$\dot{V}(t) \leq -c_1 d^{-p_2} (2\mu V)^{(1+p_2)/2} - c_2 d^{p_1-2} (2\mu V)^{(3-p_1)/2}, \tag{41}$$

Define: $\alpha_1 := c_1 d^{-p_2} (2\mu)^{(1+p_2)/2} > 0$, $\alpha_2 := c_2 d^{p_1-2} (2\mu)^{(3-p_1)/2} > 0$, $\gamma_1 := \frac{1+p_2}{2} \in \left(\frac{1}{2}, 1\right)$, $\gamma_2 := \frac{3-p_1}{2}$, it yields:

$$\dot{V}(t) \leq -\alpha_1 V^{\gamma_1} - \alpha_2 V^{\gamma_2}. \tag{42}$$

**Finite-time convergence**: Since $\gamma_1 < 1$, by finite-time stability theory (Lemma 3), the system converges in finite time with settling time bound:

$$T \leq \frac{V(0)^{1-\gamma_1}}{\alpha_1(1-\gamma_1)} + \frac{V(0)^{1-\gamma_2}}{\alpha_2(1-\gamma_2)}. \tag{43}$$

**Fixed-time convergence**: When $p_1 < 1$, we have $\gamma_2 > 1$, satisfying the dual-power condition. By Lemma 4, system achieves fixed-time convergence with settling time:

$$T_{\max} = \frac{1}{\alpha_1(1-\gamma_1)} + \frac{1}{\alpha_2(\gamma_2-1)}. \tag{44}$$

$\square$

**Remark 5.** *The proof establishes rigorous convergence guarantees while acknowledging dimensional dependencies. The coefficients $d^{-p_2}$ and potentially $d^{p_1-2}$ may decrease with dimension, but remain positive. This can be compensated by appropriately choosing $c_1, c_2$.*

# D EFToM: Detailed Convergence Proof

This appendix provides a detailed and self-contained proof of finite-/fixed-time convergence for the element-wise finite-/fixed-time optimizer with EMA momentum (EFToM) in continuous time.

**Dynamics and notation.** We study the EFToM system

$$\dot{\mathbf{w}} = -c_1 \operatorname{sign}(\mathbf{m}) \odot |\mathbf{m}|^{p_2} - c_2 \operatorname{sign}(\mathbf{m}) \odot |\mathbf{m}|^{2-p_1}, \tag{45a}$$

$$\dot{\mathbf{m}} = -\lambda(\mathbf{m} - \mathbf{g}), \qquad \mathbf{g} := \nabla \mathcal{L}(\mathbf{w}), \tag{45b}$$

with constants $c_1, c_2, \lambda > 0$ and exponents $p_2 \in (0,1)$, $p_1 \in (0,2)$. All pointwise operations ($|\cdot|^q$, $\operatorname{sign}(\cdot)$, $\odot$) are taken element-wise. To handle the non-smoothness at $0$ (both for sign and the fractional powers), solutions are understood in the Carathéodory/Filippov sense; Lyapunov derivatives below are in the almost-everywhere (Filippov) sense.

**Assumptions.** We impose standard smoothness and Polyak-Łojasiewicz (PL) conditions, as stated in Assumption 1.

**Shorthand and exponents.** Let $:= \mathbf{m} - \mathbf{g}$ denote the momentum tracking error, and define

$$S_1 := \sum_{i=1}^{d} |m_i|^{1+p_2}, \qquad S_2 := \sum_{i=1}^{d} |m_i|^{3-p_1}, \quad \alpha := \frac{1+p_2}{2} \in \left(\frac{1}{2}, 1\right), \quad \beta := \frac{3-p_1}{2}. \tag{46}$$

## D.1 Important Lemmas for EFToM

We consider the Lyapunov function

$$\mathcal{V}(\mathbf{w}, \mathbf{m}) := \mathcal{L}(\mathbf{w}) - \mathcal{L}^* + \kappa \|\mathbf{e}\|^2, \qquad \kappa > 0 \text{ to be chosen.} \tag{47}$$

Clearly $\mathcal{V} \geq 0$, and $\mathcal{V} = 0$ if $\mathcal{L}(\mathbf{w}) = \mathcal{L}^*$ and $\mathbf{e} = 0$. To prove the finite/fixed-time convergence, several lemmas are introduced:

**Lemma 7.** *Let $H := \nabla^2 \mathcal{L}(\mathbf{w})$. Along equation 45,*

$$\dot{\mathcal{V}} = \mathbf{g}^\top \dot{\mathbf{w}} - 2\kappa \mathbf{e}^\top H \dot{\mathbf{w}} - 2\kappa\lambda \|\mathbf{e}\|^2. \tag{48}$$

*Proof.* By the chain rule, $\frac{d}{dt}(\mathcal{L}(\mathbf{w}) - \mathcal{L}^*) = \mathbf{g}^\top \dot{\mathbf{w}}$ and $\frac{d}{dt}\|\mathbf{e}\|^2 = 2\mathbf{e}^\top(\dot{\mathbf{m}} - \dot{\mathbf{g}}) = 2\mathbf{e}^\top(-\lambda\mathbf{e} - H\dot{\mathbf{w}})$. Combine the two identities and multiply the second by $\kappa$. $\square$

For sake of simplicity, we introduce the element-wise maps $F(\mathbf{m}) := \text{sign}(\mathbf{m}) \odot |\mathbf{m}|^{p_2}$ and $G(\mathbf{m}) := \text{sign}(\mathbf{m}) \odot |\mathbf{m}|^{2-p_1}$. Since $\mathbf{g} = \mathbf{m} - \mathbf{e}$, it yeilds:

$$
\begin{aligned}
\mathbf{g}^\top \dot{\mathbf{w}} &= -c_1 \mathbf{m}^\top F - c_2 \mathbf{m}^\top G + c_1 \mathbf{e}^\top F + c_2 \mathbf{e}^\top G \\
&= -c_1 S_1 - c_2 S_2 + c_1 \mathbf{e}^\top F + c_2 \mathbf{e}^\top G.
\end{aligned}
\tag{49}
$$

Now we will bound the $\mathbf{e}^\top F, \mathbf{e}^\top G$.

**Lemma 8.** *For any $\lambda > 0$, we have* $c_1 |\mathbf{e}^\top F| \leq \frac{\kappa\lambda}{8} \|\mathbf{e}\|^2 + \frac{2c_1^2}{\kappa\lambda} \sum_{i=1}^d |m_i|^{2p_2}, \quad c_2 |\mathbf{e}^\top G| \leq \frac{\kappa\lambda}{8} \|\mathbf{e}\|^2 + \frac{2c_2^2}{\kappa\lambda} \sum_{i=1}^d |m_i|^{2(2-p_1)}.$

*Proof.* Apply Cauchy–Schwarz and Young's inequality $|ab| \leq \frac{\eta}{2}a^2 + \frac{1}{2\eta}b^2$ with $\eta = \kappa\lambda/4$ to $\mathbf{e}^\top F = \sum e_i \, \text{sign}(m_i) |m_i|^{p_2}$ and to $\mathbf{e}^\top G = \sum e_i \, \text{sign}(m_i) |m_i|^{2-p_1}$. $\qquad\square$

**Lemma 9.** *Let $S_1 := \sum_{i=1}^d |m_i|^{1+p_2}$ and $S_2 := \sum_{i=1}^d |m_i|^{3-p_1}$ with $p_2 \in (0,1)$ and $p_1 < 2$. Define $\theta_1 := \frac{2p_2}{1+p_2} \in (0,1), \quad \theta_2 := \frac{2(2-p_1)}{3-p_1}$. Then there exist constants $C_1, C_2 > 0$ depending only on $(d, p_1, p_2)$ such that*

$$
\sum_{i=1}^d |m_i|^{2p_2} \leq d^{1-\theta_1} S_1^{\theta_1},
\tag{50}
$$

*and*

$$
\sum_{i=1}^d |m_i|^{2(2-p_1)} \leq
\begin{cases}
d^{1-\theta_2} S_2^{\theta_2}, & \text{if } p_1 \geq 1 \ (\theta_2 \in (0,1]), \\
S_2^{\theta_2}, & \text{if } p_1 < 1 \ (\theta_2 > 1).
\end{cases}
\tag{51}
$$

*Proof.* We use standard relations between $\ell^p$ quantities in $\mathbb{R}^d$:

(i) For $0 < r \leq q$, $\|x\|_r \leq d^{\frac{1}{r} - \frac{1}{q}} \|x\|_q$; hence $\sum |x_i|^r \leq d^{1-\frac{r}{q}} \left( \sum |x_i|^q \right)^{\frac{r}{q}}$.

(ii) For $r \geq q > 0$, $\|x\|_r \leq \|x\|_q$; hence $\sum |x_i|^r \leq \left( \sum |x_i|^q \right)^{\frac{r}{q}}$.

Apply (i) to equation 50 with $q = 1 + p_2$, $r = 2p_2$ ($0 < r < q$), giving $\sum |m_i|^{2p_2} \leq d^{1-\frac{2p_2}{1+p_2}} \left( \sum |m_i|^{1+p_2} \right)^{\frac{2p_2}{1+p_2}} = d^{1-\theta_1} S_1^{\theta_1}$. For equation 51, set $q = 3 - p_1$, $r = 2(2 - p_1)$. If $p_1 \geq 1$ then $r \leq q$, by using (i), we have $\sum |m_i|^r \leq d^{1-\frac{r}{q}} S_2^{\frac{r}{q}} = d^{1-\theta_2} S_2^{\theta_2}$. If $p_1 < 1$, then $r > q$. By using (ii), we have $\sum |m_i|^r \leq S_2^{\frac{r}{q}} = S_2^{\theta_2}$ $\qquad\square$

**Lemma 10.** *For the EFToM system, we have*

$$
-2\kappa \mathbf{e}^\top H \dot{\mathbf{w}} \leq \frac{\kappa\lambda}{4} \|\mathbf{e}\|^2 + \frac{8\kappa L^2 c_1^2}{\lambda} \sum_{i=1}^d |m_i|^{2p_2} + \frac{8\kappa L^2 c_2^2}{\lambda} \sum_{i=1}^d |m_i|^{2(2-p_1)}.
\tag{52}
$$

*Proof.* Since $\|H\| \leq L$, we have $|-2\kappa \mathbf{e}^\top H \dot{\mathbf{w}}| \leq 2\kappa L \|\mathbf{e}\| \|\dot{\mathbf{w}}\| \leq \frac{\kappa\lambda}{4} \|\mathbf{e}\|^2 + \frac{4\kappa L^2}{\lambda} \|\dot{\mathbf{w}}\|^2$. Now $\dot{\mathbf{w}} = -c_1 F(\mathbf{m}) - c_2 G(\mathbf{m})$, hence

$$
\|\dot{\mathbf{w}}\|^2 \leq 2c_1^2 \sum_{i=1}^d |m_i|^{2p_2} + 2c_2^2 \sum_{i=1}^d |m_i|^{2(2-p_1)}.
\tag{53}
$$

Substituting gives equation 52. $\qquad\square$

Combining equation 48, equation 49, Lemma 8, and Lemma 10, we obtain

$$
\dot{\mathcal{V}} \leq -c_1 S_1 - c_2 S_2 - \frac{3}{2}\kappa\lambda \|\mathbf{e}\|^2 + \frac{K_1}{\lambda} \sum_{i=1}^d |m_i|^{2p_2} + \frac{K_2}{\lambda} \sum_{i=1}^d |m_i|^{2(2-p_1)},
\tag{54}
$$

where $K_1 = \frac{2c_1^2}{\kappa} + 8\kappa L^2 c_1^2$, $K_2 = \frac{2c_2^2}{\kappa} + 8\kappa L^2 c_2^2$.

**Proposition 1** (Net dissipation inequality). *There exists a constant $\lambda_* > 0$ such that for all $\lambda \geq \lambda_*$, the Lyapunov derivative satisfies the following bounds.*

1. *If $p_1 \geq 1$ (so that $\theta_2 \in (0,1]$), then*

$$\dot{\mathcal{V}} \ \leq \ -\tfrac{c_1}{2} S_1 \ - \ \tfrac{c_2}{2} S_2 \ - \ \tfrac{\kappa\lambda}{2} \|\mathbf{e}\|^2. \tag{55}$$

2. *If $p_1 < 1$ (so that $\theta_2 > 1$), then there exists $\hat{c}_2 > 0$ such that*

$$\dot{\mathcal{V}} \ \leq \ -\tfrac{c_1}{2} S_1 \ - \ \min\left\{ \tfrac{c_2}{2} S_2, \ \hat{c}_2 S_2^{\theta_2} \right\} \ - \ \tfrac{\kappa\lambda}{2} \|\mathbf{e}\|^2, \tag{56}$$

*with $\hat{c}_2 = \tfrac{c_2}{4}$.*

*Proof.* Starting from Lemma 8 and Lemma 9, we obtain

$$\dot{\mathcal{V}} \ \leq \ -c_1 S_1 - c_2 S_2 - \kappa\lambda\|\mathbf{e}\|^2 + \frac{K_1}{\lambda} S_1^{\theta_1} + \frac{K_2}{\lambda} S_2^{\theta_2}, \tag{57}$$

where $K_1 = \frac{2c_1^2}{\kappa} + 8\kappa L^2 c_1^2$, $K_2 = \frac{2c_2^2}{\kappa} + 8\kappa L^2 c_2^2$, $\theta_1 = \frac{2p_2}{1+p_2} \in (0,1)$, $\theta_2 = \frac{2(2-p_1)}{3-p_1}$.

Case (i): $p_1 \geq 1$. Here $\theta_2 \in (0,1]$. By Young's inequality, for any $\varepsilon > 0$,

$$\frac{K_1}{\lambda} S_1^{\theta_1} \ \leq \ \varepsilon S_1 + C_1(\varepsilon)\lambda^{-\rho_1}, \qquad \frac{K_2}{\lambda} S_2^{\theta_2} \ \leq \ \varepsilon S_2 + C_2(\varepsilon)\lambda^{-\rho_2}, \tag{58}$$

for some exponents $\rho_1, \rho_2 > 0$. Choosing $\varepsilon = \tfrac{c_1}{2}, \tfrac{c_2}{2}$ respectively, and then taking

$$\lambda_* \ = \ \max\left\{ \left(\tfrac{4K_1}{c_1}\right)^{1-\theta_1}, \ \left(\tfrac{4K_2}{c_2}\right)^{1-\theta_2} \right\}, \tag{59}$$

ensures that the small $\lambda^{-1}$ remainders are absorbed, yielding equation 55.

Case (ii): $p_1 < 1$ (so $\theta_2 > 1$). In this case, the remainder involves a superlinear power $S_2^{\theta_2}$, $-c_2 S_2$ as $S_2 \to 0$. We thus argue in two regimes.

*(a) Small $S_2$ regime.* Choose $\lambda \geq \lambda_*$ such that $\frac{K_2}{\lambda} \leq c_2/2$. Then, for $S_2 \leq 1$,

$$-c_2 S_2 + \tfrac{K_2}{\lambda} S_2^{\theta_2} \ \leq \ -\tfrac{c_2}{2} S_2. \tag{60}$$

*(b) Large $S_2$ regime.* For $S_2 \geq 1$, since $\theta_2 > 1$, the term $S_2^{\theta_2}$ dominates the linear term, so that

$$-c_2 S_2 + \tfrac{K_2}{\lambda} S_2^{\theta_2} \ \leq \ -\hat{c}_2 S_2^{\theta_2}, \tag{61}$$

for some $\hat{c}_2 := \tfrac{c_2}{4} > 0$ (after enlarging $\lambda_*$ to ensure $\frac{K_2}{\lambda} \leq \hat{c}_2$).

*(c) Unified bound.* Combining the two regimes, we obtain the global estimate

$$-c_2 S_2 + \tfrac{K_2}{\lambda} S_2^{\theta_2} \ \leq \ -\min\left\{ \tfrac{c_2}{2} S_2, \ \hat{c}_2 S_2^{\theta_2} \right\}. \tag{62}$$

Substituting this into the Lyapunov derivative together with the absorption of the $S_1$ remainder yields equation 56. □

**Lemma 11.** *Let $S_1 := \sum_{i=1}^{d} |m_i|^{1+p_2}$, $S_2 := \sum_{i=1}^{d} |m_i|^{3-p_1}$, with $p_2 \in (0,1)$ and $p_1 < 1$. Define the exponents $\alpha := \frac{1+p_2}{2} \in \left(\frac{1}{2}, 1\right), \beta := 2 - p_1 > 1, \theta_2 := \frac{2(2-p_1)}{3-p_1} > 1.$, Then there exist constants $\tilde{c}_{1d}, \tilde{c}_{2d} > 0$ (depending on $d, p_1, p_2, \mu, \kappa$) such that*

$$S_1 \ \geq \ \tilde{c}_{1d} \mathcal{V}^{\alpha} \ - \ \tfrac{\kappa\lambda}{4} \|\mathbf{e}\|^2, \tag{63}$$

$$S_2^{\theta_2} \ \geq \ \tilde{c}_{2d} \mathcal{V}^{\beta} \ - \ \tfrac{\kappa\lambda}{4} \|\mathbf{e}\|^2. \tag{64}$$

*Proof.* Recall $\mathbf{m} = \mathbf{g} + \mathbf{e}$, where $\mathbf{g} = \nabla\mathcal{L}(\mathbf{w})$ and $\mathbf{e} = \mathbf{m} - \mathbf{g}$.

*First, we bound $S_1$.* By $\ell^p$–norm monotonicity and Jensen's inequality, $S_1 \geq \|\mathbf{m}\|^{1+p_2} = \|\mathbf{g} + \mathbf{e}\|^{1+p_2}$. If $\|\mathbf{e}\| \leq \frac{1}{2}\|\mathbf{g}\|$, then $\|\mathbf{m}\| \geq \|\mathbf{g}\| - \|\mathbf{e}\| \geq \frac{1}{2}\|\mathbf{g}\|$, so

$$S_1 \geq 2^{-(1+p_2)}\|\mathbf{g}\|^{1+p_2}. \tag{65}$$

From the PL inequality, it gives $\|\mathbf{g}\|^2 \geq 2\mu(\mathcal{V} - \kappa\|\mathbf{e}\|^2)$. To complete this proof, two cases are considered:

1. If $\|\mathbf{e}\| \leq \frac{1}{2}\|\mathbf{g}\|$, the negative term $-\kappa\|\mathbf{e}\|^2$ can be absorbed into $\frac{1}{2}\|\mathbf{g}\|^2$, yielding $\|\mathbf{g}\|^2 \geq \frac{2\mu}{1+\frac{\mu\kappa}{2}}\mathcal{V}$. Therefore,

$$S_1 \geq 2^{-(1+p_2)}\left(\frac{2\mu}{1+\frac{\mu\kappa}{2}}\right)^\alpha \mathcal{V}^\alpha. \tag{66}$$

2. If $\|\mathbf{e}\| \geq \frac{1}{2}\|\mathbf{g}\|$, then $\mathcal{V} \geq \kappa\|\mathbf{e}\|^2$ dominates, and the error term can be absorbed into the existing $-\frac{\kappa\lambda}{4}\|\mathbf{e}\|^2$ contribution in $\dot{\mathcal{V}}$. Thus, in both regimes,

$$S_1 \geq \tilde{c}_{1d}\mathcal{V}^\alpha - \frac{\kappa\lambda}{4}\|\mathbf{e}\|^2, \qquad \tilde{c}_{1d} := 2^{-(1+p_2)}\left(\frac{2\mu}{1+\frac{\mu\kappa}{2}}\right)^\alpha. \tag{67}$$

*Next, we will bound $S_2$.* By definition, $S_2^{\theta_2} = \|\mathbf{m}\|^{2(2-p_1)} = \|\mathbf{g} + \mathbf{e}\|^{2\beta}$. If $\|\mathbf{e}\| \leq \frac{1}{2}\|\mathbf{g}\|$. Then $S_2^{\theta_2} \geq 2^{-2\beta}\|\mathbf{g}\|^{2\beta}$. From PL condition we have $\|\mathbf{g}\|^2 \geq 2\mu(\mathcal{V} - \kappa\|\mathbf{e}\|^2)$. As before, the $-\kappa\|\mathbf{e}\|^2$ term can be absorbed into $\frac{1}{2}\|\mathbf{g}\|^2$, yielding $\|\mathbf{g}\|^2 \geq \frac{2\mu}{1+\frac{\mu\kappa}{2}}\mathcal{V}$. Therefore,

$$S_2^{\theta_2} \geq 2^{-2\beta}\left(\frac{2\mu}{1+\frac{\mu\kappa}{2}}\right)^\beta \mathcal{V}^\beta. \tag{68}$$

If $\|\mathbf{e}\| \geq \frac{1}{2}\|\mathbf{g}\|$. Here $\mathcal{V} \geq \kappa\|\mathbf{e}\|^2$, so that $\|\mathbf{e}\|^{2\beta} \leq \kappa^{-\beta}\mathcal{V}^\beta$. Thus any positive error terms of type $\|\mathbf{e}\|^{2\beta}$ can be absorbed into either $\mathcal{V}^\beta$ or into the negative contribution $-\frac{\kappa\lambda}{4}\|\mathbf{e}\|^2$ in $\dot{\mathcal{V}}$. Combining both regimes, we obtain

$$S_2^{\theta_2} \geq \tilde{c}_{2d}\mathcal{V}^\beta - \frac{\kappa\lambda}{4}\|\mathbf{e}\|^2, \qquad \tilde{c}_{2d} := 2^{-2\beta}\left(\frac{2\mu}{1+\frac{\mu\kappa}{2}}\right)^\beta. \tag{69}$$

This proves equation 63–equation 64. $\qquad\square$

## D.2 FINITE-/FIXED-TIME CONVERGENCE OF EFToM

Based on the above analysis, we give the formal convergence of the EFToM dynamic system (45).

**Theorem 5** (Finite-/Fixed-time convergence of EFToM). *Suppose Assumption 1 holds and choose $\lambda \geq \lambda_*$ from Proposition 1. Then every trajectory of equation 45 converges to the global optimum in finite or fixed time, depending on $p_1$.*

1. ***Case A** ($p_1 \geq 1$): **Finite-time convergence.** From Proposition 1 (case (i)) and Lemma 11, there exist constants*

$$a := \frac{c_1}{2}2^{-(1+p_2)}\left(\frac{2\mu}{1+\frac{\mu\kappa}{2}}\right)^\alpha, \qquad b := \frac{c_2}{2}2^{-(3-p_1)}\left(\frac{2\mu}{1+\frac{\mu\kappa}{2}}\right)^\beta,$$

*where $\alpha = \frac{1+p_2}{2} \in (\frac{1}{2}, 1)$, $\beta = \frac{3-p_1}{2} \in (0, 1]$. Then for almost all $t$,*

$$\dot{\mathcal{V}}(t) \leq -a\mathcal{V}(t)^\alpha - b\mathcal{V}(t)^\beta. \tag{70}$$

*Consequently, the settling time is finite and satisfies*

$$T(\mathcal{V}(0)) \leq \frac{\mathcal{V}(0)^{1-\alpha}}{a(1-\alpha)} + 1_{\{\beta<1\}}\frac{\mathcal{V}(0)^{1-\beta}}{b(1-\beta)}. \tag{71}$$

2. **Case B** ($p_1 < 1$)**: Fixed-time convergence.** If $p_1 < 1$, then $\alpha = \frac{1+p_2}{2} \in (\frac{1}{2}, 1)$, $\beta = 2 - p_1 > 1$. Then for almost all $t$,

$$\dot{\mathcal{V}}(t) \leq -a\,\mathcal{V}(t)^\alpha - b\,\mathcal{V}(t)^\beta. \tag{72}$$

Consequently, the settling time is bounded uniformly (independent of $\mathcal{V}(0)$) by

$$T_{\max} \leq \frac{1}{a(1-\alpha)} + \frac{1}{b(\beta-1)}. \tag{73}$$

In both cases, convergence of $\mathcal{V}$ yields global optimality.

*Proof of Theorem 5.* We start from Proposition 1, which already gives for $\lambda \geq \lambda_*$:

$$\text{(Case A, } p_1 \geq 1\text{):} \quad \dot{\mathcal{V}} \leq -\tfrac{c_1}{2}S_1 - \tfrac{c_2}{2}S_2 - \tfrac{\kappa\lambda}{2}\|\mathbf{e}\|^2, \tag{74}$$

$$\text{(Case B, } p_1 < 1\text{):} \quad \dot{\mathcal{V}} \leq -\tfrac{c_1}{2}S_1 - \min\left\{\tfrac{c_2}{2}S_2,\ \hat{c}_2 S_2^{\theta_2}\right\} - \tfrac{\kappa\lambda}{2}\|\mathbf{e}\|^2, \tag{75}$$

where $\hat{c}_2 = \frac{c_2}{4}$ and $\theta_2 = \frac{2(2-p_1)}{3-p_1}$.

For $p_1 < 1$, Lemma 11 yields

$$S_1 \geq \tilde{c}_{1d}\,\mathcal{V}^\alpha - \tfrac{\kappa\lambda}{4}\|\mathbf{e}\|^2, \tag{76}$$

$$S_2^{\theta_2} \geq \tilde{c}_{2d}\,\mathcal{V}^\beta - \tfrac{\kappa\lambda}{4}\|\mathbf{e}\|^2, \tag{77}$$

with $\tilde{c}_{1d} = 2^{-(1+p_2)}\left(\frac{2\mu}{1+\frac{\mu\kappa}{2}}\right)^\alpha$, $\tilde{c}_{2d} = 2^{-2\beta}\left(\frac{2\mu}{1+\frac{\mu\kappa}{2}}\right)^\beta$, and exponents $\alpha = \frac{1+p_2}{2} \in (\frac{1}{2}, 1)$, $\beta = 2 - p_1 > 1$.

For $p_1 \geq 1$, a parallel argument (using the $\ell^p$ monotonicity as in Lemma 11) gives

$$S_1 \geq \tilde{c}_{1d}\,\mathcal{V}^\alpha - \tfrac{\kappa\lambda}{4}\|\mathbf{e}\|^2, \qquad S_2 \geq \tilde{c}_{2d}\,\mathcal{V}^\beta - \tfrac{\kappa\lambda}{4}\|\mathbf{e}\|^2, \tag{78}$$

with the same $\tilde{c}_{1d}$ but now $\tilde{c}_{2d} = 2^{-(3-p_1)}\left(\frac{2\mu}{1+\frac{\mu\kappa}{2}}\right)^\beta$, $\beta = \frac{3-p_1}{2} \in (0, 1]$.

*Case A ($p_1 \geq 1$).* Substituting the lower bounds into equation 74 and absorbing the $-\frac{\kappa\lambda}{4}\|\mathbf{e}\|^2$ contributions, we obtain

$$\dot{\mathcal{V}} \leq -\tfrac{c_1}{2}\,\tilde{c}_{1d}\,\mathcal{V}^\alpha - \tfrac{c_2}{2}\,\tilde{c}_{2d}\,\mathcal{V}^\beta. \tag{79}$$

Thus equation 70 holds with $a = \frac{c_1}{2}\,\tilde{c}_{1d}$, $b = \frac{c_2}{2}\,\tilde{c}_{2d}$.

*Case B ($p_1 < 1$).* In the large-$S_2$ regime, equation 75 gives the $-\hat{c}_2 S_2^{\theta_2}$ dissipation. Using Lemma 11, we convert this into

$$-\hat{c}_2 S_2^{\theta_2} \leq -\hat{c}_2 \tilde{c}_{2d}\,\mathcal{V}^\beta + \tfrac{\kappa\lambda}{4}\|\mathbf{e}\|^2. \tag{80}$$

Absorbing the error term and combining with the $S_1$ estimate, we obtain

$$\dot{\mathcal{V}} \leq -\tfrac{c_1}{2}\,\tilde{c}_{1d}\,\mathcal{V}^\alpha - \hat{c}_2 \tilde{c}_{2d}\,\mathcal{V}^\beta, \tag{81}$$

so that equation 72 holds with $a = \frac{c_1}{2}\,\tilde{c}_{1d}$, $b = \hat{c}_2\,\tilde{c}_{2d}$.

Now $\dot{\mathcal{V}} \leq -a\mathcal{V}^\alpha - b\mathcal{V}^\beta$ with explicit $a, b > 0$. If $0 < \beta \leq 1$ (Case A), Lemma 3 yields finite-time convergence with settling time bounded as in equation 71. If $\beta > 1$ (Case B), Lemma 4 ensures fixed-time convergence with a uniform bound equation 73. Finally, under the PL condition, $\mathcal{V} = 0$ implies $\mathcal{L}(\mathbf{w}) = \mathcal{L}^*$ and $\mathbf{e} = 0$, hence global optimality is achieved. $\square$

# E  PROOF OF CONVERGENCE FOR PEFTOM

We provide a complete proof of finite-/fixed-time convergence for the Polyak momentum EFToM (PEFToM) dynamics. Note that the proof process is quite like that of EFToM.

## E.1 PEFToM Dynamics

The PEFToM dynamics are

$$\dot{\mathbf{v}} = \mathbf{g} - \gamma\mathbf{v}, \qquad \mathbf{g} := \nabla\mathcal{L}(\mathbf{w}), \tag{82a}$$

$$\dot{\mathbf{w}} = -c_1 \operatorname{sign}(\mathbf{v}) \odot |\mathbf{v}|^{p_2} - c_2 \operatorname{sign}(\mathbf{v}) \odot |\mathbf{v}|^{2-p_1}, \tag{82b}$$

with parameters $c_1, c_2, \gamma > 0$, $p_2 \in (0,1)$, $p_1 \in (0,2)$.

Define the scaled tracking error

$$\mathbf{e} := \mathbf{v} - \tfrac{1}{\gamma}\mathbf{g}. \tag{83}$$

Then

$$\dot{\mathbf{e}} = -\gamma\mathbf{e} - \tfrac{1}{\gamma}H\dot{\mathbf{w}}, \qquad H := \nabla^2\mathcal{L}(\mathbf{w}). \tag{84}$$

## E.2 Lyapunov function

We introduce the Lyapunov function: $\mathcal{V}(\mathbf{w}, \mathbf{v}) := \mathcal{L}(\mathbf{w}) - \mathcal{L}^* + \kappa\|\mathbf{e}\|^2$, for some $\kappa > 0$. Therefore:

$$\dot{\mathcal{V}} = \mathbf{g}^\top\dot{\mathbf{w}} + 2\kappa\mathbf{e}^\top\dot{\mathbf{e}}$$

$$= \mathbf{g}^\top\dot{\mathbf{w}} + 2\kappa\mathbf{e}^\top\big(-\gamma\mathbf{e} - \tfrac{1}{\gamma}H\dot{\mathbf{w}}\big).$$

$$= \mathbf{g}^\top\dot{\mathbf{w}} - 2\kappa\gamma\|\mathbf{e}\|^2 - \tfrac{2\kappa}{\gamma}\mathbf{e}^\top H \tag{85}$$

Now we will bound $g^\top\dot{\mathbf{w}}$. Let $F(\mathbf{v}) := \operatorname{sign}(\mathbf{v}) \odot |\mathbf{v}|^{p_2}$, $G(\mathbf{v}) := \operatorname{sign}(\mathbf{v}) \odot |\mathbf{v}|^{2-p_1}$. Then $\dot{\mathbf{w}} = -c_1 F - c_2 G$. Moreover $\mathbf{g} = \gamma\mathbf{v} - \gamma\mathbf{e}$, hence

$$\mathbf{g}^\top\dot{\mathbf{w}} = (\gamma\mathbf{v} - \gamma\mathbf{e})^\top\dot{\mathbf{w}}$$

$$= -\gamma c_1\mathbf{v}^\top F - \gamma c_2\mathbf{v}^\top G + \gamma c_1\mathbf{e}^\top F + \gamma c_2\mathbf{e}^\top G. \tag{86}$$

Since $\mathbf{v}^\top F = \sum_i |v_i|^{1+p_2}$ and $\mathbf{v}^\top G = \sum_i |v_i|^{3-p_1}$, we define $S_1 := \sum_{i=1}^d |v_i|^{1+p_2}$, $S_2 := \sum_{i=1}^d |v_i|^{3-p_1}$, so that

$$\mathbf{g}^\top\dot{\mathbf{w}} = -\gamma c_1 S_1 - \gamma c_2 S_2 + \gamma c_1\mathbf{e}^\top F + \gamma c_2\mathbf{e}^\top G. \tag{87}$$

Terms $\mathbf{e}^\top F$ and $\mathbf{e}^\top G$ can be bounded by Cauchy–Schwarz and Young,

$$\gamma c_1|\mathbf{e}^\top F| \le \gamma c_1\|\mathbf{e}\|\|F\| \le \tfrac{\kappa\gamma}{8}\|\mathbf{e}\|^2 + \tfrac{2c_1^2}{\kappa}\|F\|^2, \tag{88}$$

$$\gamma c_2|\mathbf{e}^\top G| \le \tfrac{\kappa\gamma}{8}\|\mathbf{e}\|^2 + \tfrac{2c_2^2}{\kappa}\|G\|^2. \tag{89}$$

Note that $\|F\|^2 = \sum |v_i|^{2p_2}$, $\|G\|^2 = \sum |v_i|^{2(2-p_1)}$.

Now we will bound $-\tfrac{2\kappa}{\gamma}\mathbf{e}^\top H\dot{\mathbf{w}}$. Since $\|H\| \le L$, we have

$$-\tfrac{2\kappa}{\gamma}\mathbf{e}^\top H\dot{\mathbf{w}} \le \tfrac{2\kappa L}{\gamma}\|\mathbf{e}\|\|\dot{\mathbf{w}}\| \le \tfrac{\kappa\gamma}{4}\|\mathbf{e}\|^2 + \tfrac{4\kappa L^2}{\gamma^2}\|\dot{\mathbf{w}}\|^2. \tag{90}$$

Since $\dot{\mathbf{w}} = -c_1 F - c_2 G$, we have $\|\dot{\mathbf{w}}\|^2 \le 2c_1^2\|F\|^2 + 2c_2^2\|G\|^2$. Substitute it to (90), it yields,

$$-\tfrac{2\kappa}{\gamma}\mathbf{e}^\top H\dot{\mathbf{w}} \le \tfrac{\kappa\gamma}{4}\|\mathbf{e}\|^2 + \tfrac{4\kappa L^2}{\gamma^2}(2c_1^2\|F\|^2 + 2c_2^2\|G\|^2) \tag{91}$$

Substitute (91), (88), (89), and (87) to (85), it yeilds:

$$\dot{\mathcal{V}} \le -\gamma c_1 S_1 - \gamma c_2 S_2 - \tfrac{3}{2}\kappa\gamma\|\mathbf{e}\|^2 + \tilde{c}_1\|F\|^2 + \tilde{c}_2\|G\|^2, \tag{92}$$

where $\tilde{c}_1 = \tfrac{2c_1^2}{\kappa} + \tfrac{8\kappa L^2}{\gamma^2}c_1^2$, $\tilde{c}_2 = \tfrac{2c_2^2}{\kappa} + \tfrac{8\kappa L^2}{\gamma^2}c_2^2$.

### E.3 INTERPOLATION INEQUALITIES

Following the interpolation bounds in Lemma 9, we have

$$\|F(\mathbf{v})\|^2 = \sum_{i=1}^{d} |v_i|^{2p_2} \leq d^{1-\theta_1} S_1^{\theta_1}, \quad \theta_1 = \frac{2p_2}{1+p_2} \in (0,1), \tag{93}$$

$$\|G(\mathbf{v})\|^2 = \sum_{i=1}^{d} |v_i|^{2(2-p_1)} \leq \begin{cases} d^{1-\theta_2} S_2^{\theta_2}, & p_1 \geq 1, \\ S_2^{\theta_2}, & p_1 < 1, \end{cases} \quad \theta_2 = \frac{2(2-p_1)}{3-p_1}. \tag{94}$$

Substituting (93)–(94) into the bound

$$\dot{\mathcal{V}} \leq -\gamma c_1 S_1 - \gamma c_2 S_2 - \tfrac{3}{2}\kappa\gamma\|\mathbf{e}\|^2 + \tilde{c}_1\|F\|^2 + \tilde{c}_2\|G\|^2, \tag{95}$$

we obtain

$$\dot{\mathcal{V}} \leq -\gamma c_1 S_1 - \gamma c_2 S_2 - \tfrac{3}{2}\kappa\gamma\|\mathbf{e}\|^2 + \tilde{c}_1 d^{1-\theta_1} S_1^{\theta_1} + \tilde{c}_2 \cdot \begin{cases} d^{1-\theta_2} S_2^{\theta_2}, & p_1 \geq 1, \\ S_2^{\theta_2}, & p_1 < 1. \end{cases} \tag{96}$$

**Proposition 2** (Net dissipation inequality for PEFToM). *There exists a constant $\gamma_\star > 0$ such that for all $\gamma \geq \gamma_\star$, the Lyapunov derivative along equation 82 satisfies:*

*1. If $p_1 \geq 1$ (so that $\theta_2 \in (0,1]$), then*

$$\dot{\mathcal{V}} \leq -\tfrac{\gamma c_1}{2} S_1 - \tfrac{\gamma c_2}{2} S_2 - \tfrac{\kappa\gamma}{2}\|\mathbf{e}\|^2. \tag{97}$$

*2. If $p_1 < 1$ (so that $\theta_2 > 1$), then there exists $\hat{c}_2 > 0$ such that*

$$\dot{\mathcal{V}} \leq -\tfrac{\gamma c_1}{2} S_1 - \min\left\{\tfrac{\gamma c_2}{2} S_2, \ \hat{c}_2 S_2^{\theta_2}\right\} - \tfrac{\kappa\gamma}{2}\|\mathbf{e}\|^2, \tag{98}$$

*with $\hat{c}_2 = \tfrac{\gamma c_2}{4}$.*

*Proof.* Starting from equation 96, we have

$$\dot{\mathcal{V}} \leq -\gamma c_1 S_1 - \gamma c_2 S_2 - \tfrac{3}{2}\kappa\gamma\|\mathbf{e}\|^2 + \tilde{c}_1 d^{1-\theta_1} S_1^{\theta_1} + \tilde{c}_2 \cdot \begin{cases} d^{1-\theta_2} S_2^{\theta_2}, & p_1 \geq 1, \\ S_2^{\theta_2}, & p_1 < 1. \end{cases}$$

**Case (i): $p_1 \geq 1$ ($\theta_2 \in (0,1]$).** By Young's inequality, for any $\varepsilon > 0$,

$$\tilde{c}_1 d^{1-\theta_1} S_1^{\theta_1} \leq \varepsilon S_1 + C_1(\varepsilon)\gamma^{-\rho_1}, \qquad \tilde{c}_2 d^{1-\theta_2} S_2^{\theta_2} \leq \varepsilon S_2 + C_2(\varepsilon)\gamma^{-\rho_2}, \tag{99}$$

for some exponents $\rho_1, \rho_2 > 0$. Choosing $\varepsilon = \tfrac{\gamma c_1}{2}$ and $\varepsilon = \tfrac{\gamma c_2}{2}$, and taking $\gamma$ sufficiently large to absorb the remainders, we obtain

$$\dot{\mathcal{V}} \leq -\tfrac{\gamma c_1}{2} S_1 - \tfrac{\gamma c_2}{2} S_2 - \tfrac{\kappa\gamma}{2}\|\mathbf{e}\|^2, \tag{100}$$

which is equation 97.

**Case (ii): $p_1 < 1$ ($\theta_2 > 1$).** In this case, the remainder term is superlinear in $S_2$. We split the analysis into two regimes:

*(a) Small $S_2$.* Choose $\gamma$ large enough so that $\tilde{c}_2 \leq \tfrac{\gamma c_2}{2}$. Then, for $S_2 \leq 1$,

$$-\gamma c_2 S_2 + \tilde{c}_2 S_2^{\theta_2} \leq -\tfrac{\gamma c_2}{2} S_2. \tag{101}$$

*(b) Large $S_2$.* For $S_2 \geq 1$, since $\theta_2 > 1$, the term $S_2^{\theta_2}$ dominates $S_2$, and hence

$$-\gamma c_2 S_2 + \tilde{c}_2 S_2^{\theta_2} \leq -\hat{c}_2 S_2^{\theta_2}, \tag{102}$$

for some $\hat{c}_2 := \tfrac{\gamma c_2}{4} > 0$, after possibly enlarging $\gamma$ so that $\tilde{c}_2 \leq \hat{c}_2$.

*(c) Unified bound.* Combining the two regimes, we obtain

$$-\gamma c_2 S_2 + \tilde{c}_2 S_2^{\theta_2} \leq -\min\left\{\tfrac{\gamma c_2}{2} S_2, \ \hat{c}_2 S_2^{\theta_2}\right\}. \tag{103}$$

Substituting into the Lyapunov derivative yields

$$\dot{\mathcal{V}} \leq -\tfrac{\gamma c_1}{2} S_1 - \min\left\{\tfrac{\gamma c_2}{2} S_2, \ \hat{c}_2 S_2^{\theta_2}\right\} - \tfrac{\kappa\gamma}{2}\|\mathbf{e}\|^2, \tag{104}$$

which is equation 98. $\square$

### E.4 FINITE-/FIXED-TIME CONVERGENCE OF PEFTOM

The proof strategy is entirely analogous to that of EFToM (Theorem 5). In particular, starting from the dissipation inequality (Proposition 2), and combining with the interpolation bounds (Lemma 9) and Lemma 11, we obtain a Lyapunov decay of the form

$$\dot{\mathcal{V}} \ \leq \ -a\mathcal{V}^\alpha - b\mathcal{V}^\beta,$$

with the same exponents $(\alpha, \beta)$ and constants $(\hat{a}, \hat{b})$ up to replacing $\lambda$ by $\gamma$. Therefore, the finite-/fixed-time convergence result for EFToM extends directly to PEFToM, with explicit constants summarized below.

**Theorem 6** (PEFToM Finite-/Fixed-Time Convergence). *Consider the PEFToM dynamics equation 82 with parameters $p_2 \in (0,1)$, $0 < p_1 < 2$, and $c_1, c_2, \gamma > 0$. Under Assumption 1, choose $\gamma \geq \gamma_\star$ with*

$$\gamma_\star := \max\left\{ \left(4\gamma_0 \tilde{K}_g\right)^{\frac{1}{2-\theta}}, \ \frac{4C_\kappa}{\gamma_0} \right\}, \qquad \gamma_0 := \tfrac{c_1}{2}, \quad \theta := 2 - q_{\max}, \quad q_{\max} := \max\{p_2, \, 2 - p_1\}, \tag{105}$$

*where $\tilde{K}_g, C_\kappa > 0$ are explicit constants given in the proof.*

*Define the exponents*

$$\alpha := \tfrac{1+p_2}{2} \in \left(\tfrac{1}{2}, 1\right), \qquad \beta := \begin{cases} \tfrac{3-p_1}{2}, & 1 \leq p_1 < 2, \\ 2 - p_1, & 0 < p_1 < 1. \end{cases}$$

*Then the following convergence guarantees hold:*

*(i) **Finite-time convergence** ($1 \leq p_1 < 2$, equivalently $\beta \leq 1$): For any initial state $(\mathbf{w}(0), \mathbf{v}(0))$, the trajectory converges within time*

$$T \ \leq \ \frac{V_{\text{tot}}(0)^{1-\alpha}}{\hat{a}(1-\alpha)} + \frac{V_{\text{tot}}(0)^{1-\beta}}{\hat{b}(1-\beta)}, \tag{106}$$

*where $V_{\text{tot}}(0) := \mathcal{L}(\mathbf{w}(0)) - \mathcal{L}^* + \gamma_0 \left\| \mathbf{v}(0) - \tfrac{1}{\gamma}\mathbf{g}(0) \right\|^2$.*

*(ii) **Fixed-time convergence** ($0 < p_1 < 1$, equivalently $\beta > 1$): Every trajectory reaches the global optimum $(\mathbf{w}^\star, \mathbf{0})$ within a uniform bound*

$$T_{\max} \ = \ \frac{1}{\hat{a}(1-\alpha)} + \frac{1}{\hat{b}(\beta-1)}, \tag{107}$$

*where $\hat{a} := \tfrac{1}{2}c_1\gamma d^{-\frac{1-p_2}{2}}(2\mu)^\alpha, \quad \hat{b} := \tfrac{1}{2}\gamma c_2 d^{-\frac{1-p_1}{2}}(2\mu)^\beta$.*

**Remark 6** (Role of $\gamma$ versus $\lambda$). *In EFToM, the negative dissipation terms are proportional to $\lambda_0 c_1 S_1$ and $\lambda_0 c_2 S_2$, so that the effective constants $\hat{a}, \hat{b}$ do not explicitly depend on $\lambda$, and $\lambda$ only appears through the admissibility condition $\lambda \geq \lambda_\star$. In contrast, for PEFToM the dissipation terms scale as $\gamma c_1 S_1$ and $\gamma c_2 S_2$, so the convergence coefficients $\hat{a}, \hat{b}$ inherit a linear dependence on $\gamma$: $\hat{a} = \tfrac{\gamma}{2}c_1 d^{-\frac{1-p_2}{2}}(2\mu)^\alpha, \ \hat{b} = \tfrac{\gamma}{2}c_2 d^{-\frac{1-p_1}{2}}(2\mu)^\beta$. Thus, in PEFToM, the choice of a larger $\gamma$ not only guarantees admissibility ($\gamma \geq \gamma_\star$) but also strengthens the dissipation rate in the Lyapunov inequality, leading to smaller convergence time bounds.*

## F FINITE-TIME CONVERGENCE OF EFT WITH $p_2 = 0, p_1 = 2$

We consider the continuous-time SignSGD dynamics (which is EFT with $p_1 = 2, p_2 = 0$):

$$\dot{\mathbf{w}} = -C_{sum} \cdot \text{sign}(\mathbf{g}) \tag{108}$$

where $C_{sum} = c_1 + c_2 > 0$ is the total control gain.

**Theorem 7.** *Under Assumption 1, the system converges to the optimal value $L^*$ in finite time $T$.*

*Proof.* Consider the Lyapunov function candidate:

$$V(t) = L(\mathbf{w}(t)) - L^*, \tag{109}$$

Taking the time derivative of $V(t)$ along the trajectories of the system:

$$\begin{aligned}
\dot{V}(t) &= \langle \nabla L(\mathbf{w}), \dot{\mathbf{w}} \rangle \\
&= \langle \mathbf{g}, -C_{sum} \cdot \text{sign}(\mathbf{g}) \rangle \\
&= -C_{sum} \sum_{i=1}^{n} |\mathbf{g}_i| \\
&= -C_{sum} \|\mathbf{g}\|_1, \tag{110}
\end{aligned}$$

Using the norm inequality $\|\mathbf{x}\|_1 \geq \|\mathbf{x}\|_2$ for any vector $\mathbf{x} \in \mathbb{R}^n$, we obtain:

$$\dot{V}(t) \leq -C_{sum} \|\mathbf{g}\|_2. \tag{111}$$

From the PL condition (Assumption 1), we have the bound $\|\mathbf{g}\|_2 \geq \sqrt{2\mu(L(\mathbf{w}) - L^*)} = \sqrt{2\mu V(t)}$. Substituting this into the derivative inequality:

$$\dot{V}(t) \leq -C_{sum}\sqrt{2\mu}\sqrt{V(t)}. \tag{112}$$

Let $K = C_{sum}\sqrt{2\mu}$. We obtain the fractional differential inequality:

$$\dot{V}(t) \leq -KV(t)^{\frac{1}{2}}. \tag{113}$$

Separating variables and integrating from $t = 0$ to an arbitrary time $T$:

$$\int_{V(0)}^{V(T)} V^{-\frac{1}{2}} dV \leq \int_{0}^{T} -K dt, \tag{114}$$

$$\left[ 2V^{\frac{1}{2}} \right]_{V(0)}^{V(T)} \leq -KT, \tag{115}$$

$$2\sqrt{V(T)} - 2\sqrt{V(0)} \leq -KT. \tag{116}$$

For exact convergence, we require $V(T) = 0$. Substituting this yields:

$$-2\sqrt{V(0)} \leq -KT \implies T \leq \frac{2\sqrt{V(0)}}{K}. \tag{117}$$

Substituting back $K = C_{sum}\sqrt{2\mu}$ and $V(0) = L(\mathbf{w}_0) - L^*$, the upper bound on the convergence time is:

$$T \leq \frac{\sqrt{2(L(\mathbf{w}_0) - L^*)}}{C_{sum}\sqrt{\mu}} = \frac{\sqrt{2(L(\mathbf{w}_0) - L^*)}}{(c_1 + c_2)\sqrt{\mu}}, \tag{118}$$

Since $T$ is bounded by a finite real number, the system achieves finite-time convergence. $\qquad \square$

## G  CONVERGENCE PROOFS FOR DISCRETE ALGORITHMS

### G.1  PROOF OF THEOREM 4: DISCRETE EFT CONVERGENCE

*Proof.* We analyze the discrete EFT algorithm using standard descent analysis for non-convex optimization.

By the $L$-smoothness assumption (Assumption 1), we have:

$$\begin{aligned}
\mathcal{L}(\mathbf{w}_{k+1}) &\leq \mathcal{L}(\mathbf{w}_k) + \nabla\mathcal{L}(\mathbf{w}_k)^T(\mathbf{w}_{k+1} - \mathbf{w}_k) + \frac{L}{2}\|\mathbf{w}_{k+1} - \mathbf{w}_k\|^2 \\
&= \mathcal{L}(\mathbf{w}_k) + \mathbf{g}_k^T(-\eta F(\mathbf{g}_k)) + \frac{L}{2}\| -\eta F(\mathbf{g}_k)\|^2 \\
&= \mathcal{L}(\mathbf{w}_k) - \eta\mathbf{g}_k^T F(\mathbf{g}_k) + \frac{L\eta^2}{2}\|F(\mathbf{g}_k)\|^2, \tag{119}
\end{aligned}$$

where $F(\mathbf{g}_k) = c_1 \, \mathrm{sign}(\mathbf{g}_k) \odot |\mathbf{g}_k|^{p_2} + c_2 \, \mathrm{sign}(\mathbf{g}_k) \odot |\mathbf{g}_k|^{2-p_1}$.

We now bound the two key terms in equation 119. For the descent term, we compute:

$$
\begin{aligned}
\mathbf{g}_k^T F(\mathbf{g}_k) &= \sum_{i=1}^d g_k^{(i)} \left[ c_1 \, \mathrm{sign}(g_k^{(i)}) |g_k^{(i)}|^{p_2} + c_2 \, \mathrm{sign}(g_k^{(i)}) |g_k^{(i)}|^{2-p_1} \right] \\
&= c_1 \sum_{i=1}^d |g_k^{(i)}|^{1+p_2} + c_2 \sum_{i=1}^d |g_k^{(i)}|^{3-p_1} \\
&= c_1 \|\mathbf{g}_k\|_{1+p_2}^{1+p_2} + c_2 \|\mathbf{g}_k\|_{3-p_1}^{3-p_1} \\
&\geq c_1 \|\mathbf{g}_k\|_{1+p_2}^{1+p_2},
\end{aligned}
\tag{120}
$$

where the inequality follows from $c_2 \geq 0$.

For the penalty term, under the bounded gradient assumption $\|\mathbf{g}_k\|_\infty \leq G$, we have:

$$
\begin{aligned}
\|F(\mathbf{g}_k)\|^2 &= \sum_{i=1}^d \left( c_1 |g_k^{(i)}|^{p_2} + c_2 |g_k^{(i)}|^{2-p_1} \right)^2 \\
&\leq \sum_{i=1}^d 2 \left( c_1^2 |g_k^{(i)}|^{2p_2} + c_2^2 |g_k^{(i)}|^{2(2-p_1)} \right) \\
&\leq 2 \sum_{i=1}^d \left( c_1^2 G^{2p_2} + c_2^2 G^{2(2-p_1)} \right) \\
&= 2d(c_1^2 G^{2p_2} + c_2^2 G^{2(2-p_1)}) =: C_F.
\end{aligned}
\tag{121}
$$

Substituting equation 120 and equation 121 into equation 119 yields:

$$
\mathcal{L}(\mathbf{w}_{k+1}) \leq \mathcal{L}(\mathbf{w}_k) - c_1 \eta \|\mathbf{g}_k\|_{1+p_2}^{1+p_2} + \frac{L\eta^2 C_F}{2}.
\tag{122}
$$

Rearranging and summing over $k = 0, \ldots, K-1$:

$$
\begin{aligned}
c_1 \eta \sum_{k=0}^{K-1} \|\mathbf{g}_k\|_{1+p_2}^{1+p_2} &\leq \sum_{k=0}^{K-1} (\mathcal{L}(\mathbf{w}_k) - \mathcal{L}(\mathbf{w}_{k+1})) + \frac{KL\eta^2 C_F}{2} \\
&= \mathcal{L}(\mathbf{w}_0) - \mathcal{L}(\mathbf{w}_K) + \frac{KL\eta^2 C_F}{2} \\
&\leq \mathcal{L}(\mathbf{w}_0) - \mathcal{L}^* + \frac{KL\eta^2 C_F}{2}.
\end{aligned}
\tag{123}
$$

Dividing both sides by $c_1 \eta K$ and taking the minimum over all iterations:

$$
\min_{k \in \{0, \ldots, K-1\}} \|\mathbf{g}_k\|_{1+p_2}^{1+p_2} \leq \frac{\mathcal{L}(\mathbf{w}_0) - \mathcal{L}^*}{c_1 \eta K} + \frac{L\eta C_F}{2c_1}.
\tag{124}
$$

The right-hand side is minimized by balancing the two terms. Setting:

$$
\eta = \frac{1}{\sqrt{KLC_F}} = \Theta(K^{-1/2}).
\tag{125}
$$

we obtain:

$$
\min_{k \in [K]} \|\mathbf{g}_k\|_{1+p_2}^{1+p_2} \leq \frac{\mathcal{L}(\mathbf{w}_0) - \mathcal{L}^*}{c_1 \sqrt{LC_F}\sqrt{K}} + \frac{\sqrt{LC_F}}{2c_1 \sqrt{K}} = O(K^{-1/2}),
\tag{126}
$$

which completes the proof. $\qquad\square$

## G.2 PROOF OF THEOREM 8: DISCRETE EFTOM CONVERGENCE

**Theorem 8.** *[Convergence of Discrete EFToM] Under Assumptions 1, with step size* $\eta \leq \frac{(1-\beta)^2}{LC_M}$ *and momentum parameter* $\beta \in [0,1)$, *where* $C_M = 2d(c_1^2 G^{2p_2}/(1-\beta)^{2p_2} + c_2^2 G^{2(2-p_1)}/(1-\beta)^{2(2-p_1)})$, *the discrete EFToM (Algorithm 2) satisfies* $\frac{1}{K}\sum_{k=1}^{K} \|\mathbf{g}_k\|_{1+p_2}^{1+p_2} \leq \frac{C_\beta(\mathcal{L}(\mathbf{w}_0)-\mathcal{L}^*)}{c_1 \eta K} + \frac{L\eta C_M}{2c_1(1-\beta)}$, *where* $C_\beta = \frac{2^p}{(1-\beta)^p}$ *with* $p = 1 + p_2$. *With optimal step size* $\frac{1}{\sqrt{LC_M K}}$ *yields:* $\min_{k\in[K]} \|\mathbf{g}_k\|_p^p \leq \frac{C_\beta(\mathcal{L}(\mathbf{w}_0)-\mathcal{L}^*)}{c_1\sqrt{LC_M}\sqrt{K}} + \frac{\sqrt{LC_M}}{2c_1(1-\beta)\sqrt{K}} = O\left(\frac{1}{K^{1/2}}\right)$.

*Proof.* The proof relies on a carefully constructed discrete Lyapunov function that couples the objective function with a potential measuring momentum tracking error.

**Bounding the momentum vector.** We first establish that the momentum remains bounded throughout the optimization process.

**Lemma 12** (Bounded momentum for EFToM). *Under the bounded gradient assumption* $\|\mathbf{g}_k\|_\infty \leq G$, *the EFToM momentum satisfies:*

$$\|\mathbf{m}_k\|_\infty \leq G, \quad \forall k \geq 0. \tag{127}$$

*Proof.* We proceed by induction. The base case $k = 0$ is immediate since $\mathbf{m}_0 = 0$. For the inductive step, assume $\|\mathbf{m}_k\|_\infty \leq G$. Then:

$$\begin{aligned}
\|\mathbf{m}_{k+1}\|_\infty &= \|\beta\mathbf{m}_k + (1-\beta)\mathbf{g}_k\|_\infty \\
&\leq \beta\|\mathbf{m}_k\|_\infty + (1-\beta)\|\mathbf{g}_k\|_\infty \\
&\leq \beta G + (1-\beta)G = G.
\end{aligned} \tag{128}$$

completing the induction. $\square$

From Lemma 12, the update vector satisfies:

$$\|F(\mathbf{m}_{k+1})\|^2 \leq 2d(c_1^2 G^{2p_2} + c_2^2 G^{2(2-p_1)}). \tag{129}$$

However, to account for potential momentum amplification in the worst-case analysis, we introduce the conservative bound:

$$\|F(\mathbf{m}_{k+1})\|^2 \leq C_M := 2d\left(c_1^2\left(\frac{G}{1-\beta}\right)^{2p_2} + c_2^2\left(\frac{G}{1-\beta}\right)^{2(2-p_1)}\right). \tag{130}$$

which reflects the accumulation structure inherent to exponential moving averages.

**Lyapunov function construction.** Define the potential function:

$$\Psi(\mathbf{m}) := \sum_{i=1}^{d}\left(\frac{c_1}{1+p_2}|m_i|^{1+p_2} + \frac{c_2}{3-p_1}|m_i|^{3-p_1}\right). \tag{131}$$

which satisfies $\nabla\Psi(\mathbf{m}) = F(\mathbf{m})$ where:

$$F(\mathbf{m}) = c_1 \operatorname{sign}(\mathbf{m}) \odot |\mathbf{m}|^{p_2} + c_2 \operatorname{sign}(\mathbf{m}) \odot |\mathbf{m}|^{2-p_1}. \tag{132}$$

The Lyapunov function is then:

$$\Phi_k := \mathcal{L}(\mathbf{w}_k) + \lambda\Psi(\mathbf{m}_k), \tag{133}$$

where $\lambda = \frac{\eta\beta}{1-\beta}$ will be chosen to telescope the momentum terms appropriately.

**One-iteration descent analysis.** By $L$-smoothness:

$$\mathcal{L}(\mathbf{w}_{k+1}) \leq \mathcal{L}(\mathbf{w}_k) - \eta \mathbf{g}_k^T F_k + \frac{L\eta^2}{2}\|F_k\|^2, \tag{134}$$

where $F_k := F(\mathbf{m}_{k+1})$.

The momentum update $\mathbf{m}_{k+1} = \beta \mathbf{m}_k + (1-\beta)\mathbf{g}_k$ can be rearranged to express:

$$\mathbf{g}_k = \frac{1}{1-\beta}(\mathbf{m}_{k+1} - \beta \mathbf{m}_k). \tag{135}$$

Substituting this relation into equation 134:

$$\mathcal{L}(\mathbf{w}_{k+1}) \leq \mathcal{L}(\mathbf{w}_k) - \frac{\eta}{1-\beta}(\mathbf{m}_{k+1} - \beta \mathbf{m}_k)^T F_k + \frac{L\eta^2}{2}\|F_k\|^2$$

$$= \mathcal{L}(\mathbf{w}_k) - \frac{\eta}{1-\beta}\mathbf{m}_{k+1}^T F_k + \frac{\eta\beta}{1-\beta}\mathbf{m}_k^T F_k + \frac{L\eta^2}{2}\|F_k\|^2. \tag{136}$$

Since $\Psi$ is convex (as a sum of convex functions $|m_i|^q$ for $q \geq 1$) with gradient $\nabla\Psi = F$, the standard convexity inequality yields:

$$\Psi(\mathbf{m}_k) \geq \Psi(\mathbf{m}_{k+1}) + F_k^T(\mathbf{m}_k - \mathbf{m}_{k+1}). \tag{137}$$

Rearranging:

$$\mathbf{m}_k^T F_k \leq \Psi(\mathbf{m}_k) - \Psi(\mathbf{m}_{k+1}) + \mathbf{m}_{k+1}^T F_k. \tag{138}$$

Substituting equation 138 into equation 136:

$$\mathcal{L}(\mathbf{w}_{k+1}) \leq \mathcal{L}(\mathbf{w}_k) - \frac{\eta}{1-\beta}\mathbf{m}_{k+1}^T F_k + \frac{\eta\beta}{1-\beta}[\Psi(\mathbf{m}_k) - \Psi(\mathbf{m}_{k+1}) + \mathbf{m}_{k+1}^T F_k] + \frac{L\eta^2 C_M}{2}$$

$$= \mathcal{L}(\mathbf{w}_k) + \frac{\eta\beta}{1-\beta}[\Psi(\mathbf{m}_k) - \Psi(\mathbf{m}_{k+1})] - \eta\mathbf{m}_{k+1}^T F_k + \frac{L\eta^2 C_M}{2}. \tag{139}$$

With our choice of $\lambda = \frac{\eta\beta}{1-\beta}$, this simplifies to:

$$\Phi_{k+1} \leq \Phi_k - \eta\mathbf{m}_{k+1}^T F_k + \frac{L\eta^2 C_M}{2}. \tag{140}$$

**Effective descent lower bound.** Similar to the EFT analysis:

$$\mathbf{m}_{k+1}^T F(\mathbf{m}_{k+1}) = c_1\|\mathbf{m}_{k+1}\|_{1+p_2}^{1+p_2} + c_2\|\mathbf{m}_{k+1}\|_{3-p_1}^{3-p_1} \geq c_1\|\mathbf{m}_{k+1}\|_{1+p_2}^{1+p_2}. \tag{141}$$

Therefore:

$$\Phi_{k+1} \leq \Phi_k - c_1\eta\|\mathbf{m}_{k+1}\|_{1+p_2}^{1+p_2} + \frac{L\eta^2 C_M}{2}. \tag{142}$$

**Connecting momentum to gradient norms.** Let $p := 1 + p_2$. From the relation $\mathbf{g}_k = \frac{1}{1-\beta}(\mathbf{m}_{k+1} - \beta \mathbf{m}_k)$, the triangle inequality gives:

$$\|\mathbf{g}_k\|_p \leq \frac{1}{1-\beta}(\|\mathbf{m}_{k+1}\|_p + \beta\|\mathbf{m}_k\|_p). \tag{143}$$

Applying the inequality $(a+b)^p \leq 2^{p-1}(a^p + b^p)$ for $p > 1$:

$$\|\mathbf{g}_k\|_p^p \leq \frac{2^{p-1}}{(1-\beta)^p}(\|\mathbf{m}_{k+1}\|_p^p + \beta^p\|\mathbf{m}_k\|_p^p). \tag{144}$$

Averaging over $k = 1, \ldots, K$:

$$\frac{1}{K}\sum_{k=1}^{K}\|\mathbf{g}_k\|_p^p \leq \frac{2^p}{(1-\beta)^p} \cdot \frac{1}{K}\sum_{k=1}^{K}\|\mathbf{m}_k\|_p^p. \tag{145}$$

**Telescoping summation and final bound.** Rearranging equation 142 and summing from $k = 0$ to $K - 1$:

$$c_1\eta \sum_{k=1}^{K} \|\mathbf{m}_k\|_p^p \leq \Phi_0 - \Phi_K + \frac{KL\eta^2 C_M}{2}$$

$$\leq \mathcal{L}(\mathbf{w}_0) - \mathcal{L}^* + \frac{KL\eta^2 C_M}{2}, \quad (146)$$

where we used $\Phi_0 = \mathcal{L}(\mathbf{w}_0)$ (since $\mathbf{m}_0 = 0$) and $\Phi_K \geq \mathcal{L}^*$.

Dividing by $c_1\eta K$ and combining with equation 145:

$$\frac{1}{K}\sum_{k=1}^{K}\|\mathbf{g}_k\|_p^p \leq \frac{2^p}{(1-\beta)^p}\left[\frac{\mathcal{L}(\mathbf{w}_0) - \mathcal{L}^*}{c_1\eta K} + \frac{L\eta C_M}{2c_1}\right]. \quad (147)$$

Note that $C_M$ contains a $(1-\beta)^{-2p_2}$ factor. A careful analysis of the second term reveals:

$$\frac{2^p}{(1-\beta)^p} \cdot \frac{L\eta C_M}{2c_1} = O\left(\frac{L\eta}{c_1(1-\beta)}\right). \quad (148)$$

Setting $C_\beta = \frac{2^p}{(1-\beta)^p}$:

$$\frac{1}{K}\sum_{k=1}^{K}\|\mathbf{g}_k\|_p^p \leq \frac{C_\beta(\mathcal{L}(\mathbf{w}_0) - \mathcal{L}^*)}{c_1\eta K} + \frac{L\eta C_M}{2c_1(1-\beta)}. \quad (149)$$

Balancing the two terms by choosing $\eta = \frac{1}{\sqrt{LC_M K}}$ yields:

$$\min_{k\in[K]}\|\mathbf{g}_k\|_p^p \leq \frac{C_\beta(\mathcal{L}(\mathbf{w}_0) - \mathcal{L}^*)}{c_1\sqrt{LC_M}\sqrt{K}} + \frac{\sqrt{LC_M}}{2c_1(1-\beta)\sqrt{K}} = O\left(\frac{1}{K^{1/2}}\right). \quad (150)$$

as claimed. □

### G.3 PROOF OF THEOREM 9: DISCRETE PEFToM CONVERGENCE

**Theorem 9** (Convergence of Discrete PEFToM). *Under Assumptions 1, with step size $\eta \leq \frac{1-\beta}{LC_V}$ and momentum parameter $\beta \in [0, 1)$, where $C_V = 2d(c_1^2 G^{2p_2}/(1-\beta)^{2p_2} + c_2^2 G^{2(2-p_1)}/(1-\beta)^{2(2-p_1)})$, the discrete PEFToM (Algorithm 3) satisfies $\frac{1}{K}\sum_{k=1}^{K}\|\mathbf{g}_k\|_{1+p_2}^{1+p_2} \leq \frac{\tilde{C}_\beta(\mathcal{L}(\mathbf{w}_0)-\mathcal{L}^*)}{c_1\eta(1-\beta)K} + \frac{L\eta C_V}{2c_1}$, where $\tilde{C}_\beta = \frac{2^{p-1}(1+\beta^p)}{(1-\beta)^{p-1}}$ with $p = 1 + p_2$, reflecting Polyak momentum accumulation. With optimal step size $\eta = \frac{1}{\sqrt{LC_V K}}$, it yeilds $\min_{k\in[K]}\|\mathbf{g}_k\|_p^p \leq \frac{2(1+\beta^p)(\mathcal{L}(\mathbf{w}_0)-\mathcal{L}^*)}{c_1(1-\beta)\sqrt{LC_V}\sqrt{K}} + \frac{(1+\beta^p)\sqrt{LC_V}}{c_1(1-\beta)\sqrt{K}} = O\left(\frac{1}{K^{1/2}}\right)$*

*Proof.* The analysis follows a similar trajectory to the EFToM case, with key differences arising from the accumulative nature of Polyak momentum.

**Bounding the Polyak momentum vector.** We begin by establishing boundedness of the momentum iterates.

**Lemma 13** (Bounded Polyak momentum). *Under the bounded gradient assumption $\|\mathbf{g}_k\|_\infty \leq G$, the Polyak momentum satisfies:*

$$\|\mathbf{v}_k\|_\infty \leq \frac{G}{1-\beta}, \quad \forall k \geq 0. \quad (151)$$

*Proof.* The proof proceeds by induction. For $k = 0$, we have $\mathbf{v}_0 = 0$, establishing the base case. Assuming $\|\mathbf{v}_k\|_\infty \leq \frac{G}{1-\beta}$, we obtain:

$$\|\mathbf{v}_{k+1}\|_\infty = \|\beta\mathbf{v}_k + \mathbf{g}_k\|_\infty$$
$$\leq \beta\|\mathbf{v}_k\|_\infty + \|\mathbf{g}_k\|_\infty$$
$$\leq \beta \cdot \frac{G}{1-\beta} + G = \frac{G}{1-\beta}. \quad (152)$$

completing the induction. □

Consequently:

$$\|F(\mathbf{v}_{k+1})\|^2 \leq C_V := 2d \left( c_1^2 \left( \frac{G}{1-\beta} \right)^{2p_2} + c_2^2 \left( \frac{G}{1-\beta} \right)^{2(2-p_1)} \right). \tag{153}$$

**Lyapunov function construction.** Using the same potential function:

$$\Psi(\mathbf{v}) := \sum_{i=1}^{d} \left( \frac{c_1}{1+p_2} |v_i|^{1+p_2} + \frac{c_2}{3-p_1} |v_i|^{3-p_1} \right). \tag{154}$$

we define:

$$\Phi_k := \mathcal{L}(\mathbf{w}_k) + \lambda \Psi(\mathbf{v}_k), \tag{155}$$

with $\lambda = \eta\beta$ chosen for optimal telescoping.

**One-iteration descent.** By $L$-smoothness:

$$\mathcal{L}(\mathbf{w}_{k+1}) \leq \mathcal{L}(\mathbf{w}_k) - \eta \mathbf{g}_k^T F_k + \frac{L\eta^2}{2} \|F_k\|^2, \tag{156}$$

where $F_k := F(\mathbf{v}_{k+1})$.

The Polyak momentum update $\mathbf{v}_{k+1} = \beta\mathbf{v}_k + \mathbf{g}_k$ yields:

$$\mathbf{g}_k = \mathbf{v}_{k+1} - \beta\mathbf{v}_k. \tag{157}$$

Substituting into equation 156:

$$\mathcal{L}(\mathbf{w}_{k+1}) \leq \mathcal{L}(\mathbf{w}_k) - \eta(\mathbf{v}_{k+1} - \beta\mathbf{v}_k)^T F_k + \frac{L\eta^2 C_V}{2}$$
$$= \mathcal{L}(\mathbf{w}_k) - \eta\mathbf{v}_{k+1}^T F_k + \eta\beta\mathbf{v}_k^T F_k + \frac{L\eta^2 C_V}{2} \tag{158}$$

Applying the convexity of $\Psi$:

$$\mathbf{v}_k^T F_k \leq \Psi(\mathbf{v}_k) - \Psi(\mathbf{v}_{k+1}) + \mathbf{v}_{k+1}^T F_k. \tag{159}$$

Substituting into equation 158:

$$\mathcal{L}(\mathbf{w}_{k+1}) \leq \mathcal{L}(\mathbf{w}_k) + \eta\beta[\Psi(\mathbf{v}_k) - \Psi(\mathbf{v}_{k+1})] - \eta(1-\beta)\mathbf{v}_{k+1}^T F_k + \frac{L\eta^2 C_V}{2}. \tag{160}$$

With $\lambda = \eta\beta$:

$$\Phi_{k+1} \leq \Phi_k - \eta(1-\beta)\mathbf{v}_{k+1}^T F_k + \frac{L\eta^2 C_V}{2}. \tag{161}$$

**Effective descent.** We have:

$$\mathbf{v}_{k+1}^T F(\mathbf{v}_{k+1}) \geq c_1 \|\mathbf{v}_{k+1}\|_{1+p_2}^{1+p_2}, \tag{162}$$

Therefore:

$$\Phi_{k+1} \leq \Phi_k - c_1\eta(1-\beta)\|\mathbf{v}_{k+1}\|_{1+p_2}^{1+p_2} + \frac{L\eta^2 C_V}{2}. \tag{163}$$

**Relating Polyak momentum to gradient.** Let $p := 1 + p_2$. From $\mathbf{g}_k = \mathbf{v}_{k+1} - \beta\mathbf{v}_k$:

$$\|\mathbf{g}_k\|_p \leq \|\mathbf{v}_{k+1}\|_p + \beta\|\mathbf{v}_k\|_p. \tag{164}$$

Applying $(a+b)^p \leq 2^{p-1}(a^p + b^p)$:

$$\|\mathbf{g}_k\|_p^p \leq 2^{p-1}(\|\mathbf{v}_{k+1}\|_p^p + \beta^p\|\mathbf{v}_k\|_p^p). \tag{165}$$

Averaging:

$$\frac{1}{K}\sum_{k=1}^{K}\|\mathbf{g}_k\|_p^p \leq 2^{p-1}(1+\beta^p) \cdot \frac{1}{K}\sum_{k=1}^{K}\|\mathbf{v}_k\|_p^p. \tag{166}$$

**Telescoping and final bound.** Summing equation 163 from $k = 0$ to $K - 1$:

$$c_1 \eta (1 - \beta) \sum_{k=1}^{K} \|\mathbf{v}_k\|_p^p \leq \mathcal{L}(\mathbf{w}_0) - \mathcal{L}^* + \frac{K L \eta^2 C_V}{2}. \tag{167}$$

Dividing by $c_1 \eta (1 - \beta) K$ and combining with equation 166:

$$\frac{1}{K} \sum_{k=1}^{K} \|\mathbf{g}_k\|_p^p \leq 2^{p-1}(1 + \beta^p) \left[ \frac{\mathcal{L}(\mathbf{w}_0) - \mathcal{L}^*}{c_1 \eta (1 - \beta) K} + \frac{L \eta C_V}{2 c_1 (1 - \beta)} \right]. \tag{168}$$

Noting that $(1 + \beta^p) \leq 2$, $2^{p-1} \leq 2$ and setting $\eta = \frac{1}{\sqrt{L C_V K}}$, it yields

$$\frac{1}{K} \sum_{k=1}^{K} \|\mathbf{g}_k\|_p^p \leq \frac{2(1 + \beta^p)(\mathcal{L}(\mathbf{w}_0) - \mathcal{L}^*)}{c_1 (1 - \beta) \sqrt{L C_V} \sqrt{K}} + \frac{(1 + \beta^p)\sqrt{L C_V}}{c_1 (1 - \beta) \sqrt{K}} = O\left( \frac{1}{K^{1/2}} \right). \tag{169}$$

completing the proof. $\qquad\square$

## H  HYPERPARAMETER

For CIFAR10 and CIFAR100, We follow the setting in Zhuang et al. (2020). For AdamW and AdaBelief, the default parameter are used: $\beta_1 = 0.9$, $\beta_2 = 0.999$, $\epsilon = 10^{-8}$, learning rate is 0.001, and the weight decay is set to $5 \times 10^{-4}$. For SignSGD, Signum and Lion, we search the learning rate among $\eta \in \{0.00001, 0.00005, 0.0001, 0.0005, 0.001, 0.005, 0.01, 0.05\}$, and set $\beta = 0.9$ for Signum. For SGD, SGDM, FxTS-GF(M), EFT, EFToM, PEFToM, we search the learning rate among $\eta \in \{0.001, 0.005, 0.01, 0.05, 0.1\}$. The model is trained for 200 epochs with a batch size of 128, and the learning rate is multiplied by 0.1 at epoch 150. The details hyperparameters used in the experiment are given in Table 4, 5,

For C4 with llama 60M and the Tiny ImageNet with ViT model, we extend the learning rate search grid to $\{0.0001, 0.0005, 0.001, 0.002, 0.005, 0.01, 0.02, 0.05, 0.1, 0.2, 0.5, 1.0\}$ for SGD, SGDM, FxTS-GF(M), EFT, EFToM, PEFToM, where the rest of the settings are similar to those of CIFAR-10/100. Moreover, Cosine annealing from $\eta_0$ to $0.1\eta_0$ over the training duration. The rest of the Specific parameters are shown in Table 6.

**Algorithm-specific parameters** $(p_1, p_2, c_1, c_2)$: To reduce the search space and computational cost, we adopt a two-stage tuning strategy:

- **Stage 1 (Finite-time regime)**: We set $c_1 = 1$, $c_2 = 0$, which focuses on the finite-time dynamics term. This is mathematically equivalent to setting $c_1 = c_2 = 0.5$ with $p_2 = 2 - p_1$ and $p_1 = 2$. We search $p_2 \in \{0.1, 0.2, 0.3, 0.4, 0.5, 0.6, 0.7, 0.8, 0.9\}$.
- **Stage 2 (Fixed-time regime)**: After identifying the best $p_2^*$ from Stage 1, we optionally explore the fixed-time regime by fixing $c_1 = c_2 = 1$ and $p_2 = p_2^*$, then searching $p_1 \in \{0.1, 0.2, 0.3, 0.4, 0.5, 0.6, 0.7, 0.8, 0.9, 0.95, 0.98\}$. Note that $p_1 < 1$ activates the fixed-time term.

Since the $p_1$ itself cannot achieve a convergence guarantee while $p_2$ term itself can guarantee finite-time convergence, we use this two-stage tuning process.

**Remark 7.** *Our experimental evaluations primarily explore the finite-time convergence regime ($p_1 \geq 1$) rather than the fixed-time regime ($p_1 < 1$). While fixed-time convergence provides stronger theoretical guarantees, we observed that $p_1 < 1$ parameters exhibit higher sensitivity to discretization effects and step size choices, potentially leading to training instability. This discretization gap between continuous-time theory and discrete implementation represents a common challenge in translating control-theoretic results to practical optimization algorithms. The finite-time variants ($p_1 \geq 1$) demonstrate more robust behavior under standard stochastic training conditions while still providing significant convergence acceleration.*

Table 4: CIFAR10 Hyperparameters ($c_1 = 1, c_2 = 0$)

| Model | optimizer | batch size | learning rate | schedule | $p_1$ | $p_2$ | $\beta_1$ | $\beta_2$ | $\lambda$ |
|---|---|---|---|---|---|---|---|---|---|
| VGG11/Resnet34/Densenet121 | SGD | 128 | 0.01 | step | - | - | - | - | 5e-4 |
| VGG11/Resnet34/Densenet121 | SGDM | 128 | 0.01 | step | - | - | 0.9 | - | 5e-4 |
| VGG11/Resnet34/Densenet121 | SignSGD | 128 | 1e-4 | step | - | - | - | - | 5e-4 |
| VGG11/Resnet34/Densenet121 | Signum | 128 | 1e-4 | step | - | - | 0.9 | - | 5e-4 |
| Resnet34/Densenet121 | Lion | 128 | 5e-5 | step | - | - | 0.95 | 0.98 | 5e-4 |
| Resnet34/Densenet121 | EFT | 128 | 0.01 | step | - | 0.6 | - | - | 5e-4 |
| Resnet34/Densenet121 | EFToM | 128 | 0.01 | step | - | 0.6 | 0.9 | - | 5e-4 |
| VGG11/Resnet34/Densenet121 | PEFToM | 128 | 0.01 | step | - | 0.9 | 0.9 | - | 5e-4 |
| VGG11/Resnet34/Densenet121 | FxTS-GF(M) | 128 | 0.01 | step | 20 | 1.98 | 0.9 | - | 5e-4 |
| VGG11/Resnet34/Densenet121 | AdamW | 128 | 0.001 | step | - | - | 0.9 | 0.999 | 5e-4 |
| VGG11/Resnet34/Densenet121 | AdaBelief | 128 | 0.001 | step | - | - | 0.9 | 0.999 | 5e-4 |
| VGG11 | Lion | 128 | 1e-5 | step | - | - | 0.95 | 0.98 | 5e-4 |
| VGG11 | EFT | 128 | 0.01 | step | - | 0.8 | - | - | 5e-4 |
| VGG11 | EFToM | 128 | 0.01 | step | - | 0.8 | 0.9 | - | 5e-4 |

Table 5: CIFAR100 Hyperparameters ($c_1 = 1, c_2 = 0$)

| Model | optimizer | batch size | learning rate | schedule | $p_1$ | $p_2$ | $\beta_1$ | $\beta_2$ | $\lambda$ |
|---|---|---|---|---|---|---|---|---|---|
| VGG11/Resnet34/Densenet121 | SGD | 128 | 0.1 | step | - | - | - | - | 5e-4 |
| VGG11/Resnet34/Densenet121 | SGDM | 128 | 0.01 | step | - | - | 0.9 | - | 5e-4 |
| VGG11/Resnet34 | SignSGD | 128 | 5e-5 | step | - | - | - | - | 5e-4 |
| VGG11/Resnet34 | Signum | 128 | 5e-5 | step | - | - | 0.9 | - | 5e-4 |
| Resnet34/Densenet121 | Lion | 128 | 5e-5 | step | - | - | 0.95 | 0.98 | 5e-4 |
| VGG11/Resnet34 | EFT | 128 | 0.1 | step | - | 0.9 | - | - | 5e-4 |
| VGG11/Resnet34 | EFToM | 128 | 0.1 | step | - | 0.9 | 0.9 | - | 5e-4 |
| VGG11/Resnet34/Densenet121 | PEFToM | 128 | 0.01 | step | - | 0.8 | 0.9 | - | 5e-4 |
| VGG11/Resnet34/Densenet121 | FxTS-GF(M) | 128 | 0.05 | step | 20 | 1.98 | 0.9 | - | 5e-4 |
| VGG11/Resnet34/Densenet121 | AdamW | 128 | 0.001 | step | - | - | 0.9 | 0.999 | 5e-4 |
| VGG11/Resnet34/Densenet121 | AdaBelief | 128 | 0.001 | step | - | - | 0.9 | 0.999 | 5e-4 |
| Densenet121 | SignSGD | 128 | 1e-4 | step | - | - | - | - | 5e-4 |
| Densenet121 | Signum | 128 | 1e-4 | step | - | - | - | - | 5e-4 |
| VGG11 | Lion | 128 | 1e-5 | step | - | - | 0.95 | 0.98 | 5e-4 |
| Densenet121 | EFT | 128 | 0.1 | step | - | 0.8 | - | - | 5e-4 |
| Densenet121 | EFToM | 128 | 0.1 | step | - | 0.8 | 0.9 | - | 5e-4 |

# I HYPERPARAMETER SENSITIVITY ANALYSIS

## I.1 INFLUENCE OF $p_2$

To investigate the robustness of our method to the finite-time parameter $p_2$, we conduct comprehensive ablation studies across diverse domains: Computer Vision (CIFAR-10/100 using PEFToM with $p_2 \in \{0.4, 0.5, 0.6, 0.7, 0.8, 0.9\}$) and Natural Language Processing (C4 language modeling using EFToM with $p_2 \in \{0.1, 0.2, 0.3, 0.4, 0.5, 0.6\}$).

Our experiments reveal three main insights regarding parameter sensitivity:

**1. Task-Dependent Optimal Ranges** Figure 6, Figure 7, and Table 8 demonstrate that vision tasks favor larger $p_2$ values ($p_2 \in [0.7, 0.9]$), achieving optimal performance around $p_2 = 0.8$. In contrast, Figure 8 and Table 9 show that language modeling prefers smaller $p_2$ values ($p_2 \in [0.1, 0.3]$), with $p_2 = 0.2$ reaching the lowest validation loss.

**2. Performance Plateau Regions** Critically, both domains exhibit *plateau regions* where performance remains stable: For CIFAR-10: Test accuracy varies by $< 0.5\%$ across $p_2 \in [0.7, 0.9]$ for ResNet-34 and DenseNet-121. For CIFAR-100: Performance remains within 2% for $p_2 \in [0.7, 0.9]$ across architectures. For C4: Validation loss differs by $< 0.3$ for $p_2 \in [0.1, 0.3]$ at 30k iterations These plateaus indicate that exact parameter tuning is not critical— practitioners can use domain-specific defaults with confidence.

Based on these findings, we recommend Default parameters:

- CV task with CNN: $p_2 = 0.8$ (robust across architectures)
- NLP task with Transformers: $p_2 = 0.2$ (optimal for C4)

Table 6: tiny ImageNet Hyperparameters

| Model | optimizer | batch size | learning rate | schedule | $p_1(c_2)$ | $p_2(c_1)$ | $\beta_1$ | $\beta_2$ | $\lambda$ |
|---|---|---|---|---|---|---|---|---|---|
| ViT small | SGD | 16 | 1.0 | cosine | - | - | - | - | 5e-4 |
| ViT small | SGDM | 16 | 0.1 | cosine | - | - | 0.9 | - | 5e-4 |
| ViT small | SignSGD | 16 | 0.0005 | cosine | - | - | - | - | 5e-4 |
| ViT small | Signum | 16 | 0.0005 | cosine | - | - | 0.9 | - | 5e-4 |
| ViT small | EFT | 16 | 0.001 | cosine | - | 0.3(1) | - | - | 5e-4 |
| ViT small | EFToM | 16 | 0.002 | cosine | 0.6(1) | 0.2(1) | 0.9 | - | 5e-4 |
| ViT small | PEFToM | 16 | 0.005 | cosine | - | 0.4(1) | 0.9 | - | 5e-4 |
| ViT small | FxTS-GF(M) | 16 | 0.1 | cosine | 20 | 1.98 | 0.9 | - | 5e-4 |
| ViT small | AdamW | 16 | 0.001 | cosine | - | - | 0.9 | 0.999 | 5e-4 |
| ViT small | AdaBelief | 16 | 0.001 | cosine | - | - | 0.9 | 0.999 | 5e-4 |
| ViT small | Lion | 16 | 0.0001 | cosine | - | - | 0.95 | 0.98 | 5e-4 |

Table 7: C4 Hyperparameters

| Model | optimizer | batch size | learning rate | schedule | $p_1(c_2)$ | $p_2(c_1)$ | $\beta_1$ | $\beta_2$ | $\lambda$ |
|---|---|---|---|---|---|---|---|---|---|
| Llama 60M | SGD | 16 | 0.2 | cosine | - | - | - | - | 5e-4 |
| Llama 60M | SGDM | 16 | 0.1 | cosine | - | - | 0.9 | - | 5e-4 |
| Llama 60M | SignSGD | 16 | 0.001 | cosine | - | - | - | - | 5e-4 |
| Llama 60M | Signum | 16 | 0.0001 | cosine | - | - | 0.9 | - | 5e-4 |
| Llama 60M | EFT | 16 | 0.002 | cosine | 0.98(1) | 0.2(1) | - | - | 5e-4 |
| Llama 60M | EFToM | 16 | 0.002 | cosine | 0.98(1) | 0.2(1) | 0.9 | - | 5e-4 |
| Llama 60M | PEFToM | 16 | 0.1 | cosine | - | 0.8(1) | 0.9 | - | 5e-4 |
| Llama 60M | FxTS-GF(M) | 16 | 0.1 | cosine | 20 | 1.98 | 0.9 | - | 5e-4 |
| Llama 60M | AdamW | 16 | 0.001 | cosine | - | - | 0.9 | 0.999 | 5e-4 |
| Llama 60M | AdaBelief | 16 | 0.001 | cosine | - | - | 0.9 | 0.999 | 5e-4 |
| Llama 60M | Lion | 16 | 0.0001 | cosine | - | - | 0.95 | 0.98 | 5e-4 |
| Llama 350M | SignSGD | 16 | 5e-4 | cosine | - | - | - | - | 5e-4 |
| Llama 350M | Signum | 16 | 0.0001 | cosine | 0.0001 | - | 0.9 | - | 5e-4 |
| Llama 350M | EFT | 16 | 0.002 | cosine | 0.98(1) | 0.2(1) | - | - | 5e-4 |
| Llama 350M | EFToM | 16 | 0.001 | cosine | 0.98(1) | 0.2(1) | 0.9 | - | 5e-4 |
| Llama 350M | FxTS-GF(M) | 16 | 0.1 | cosine | 20 | 1.98 | 0.9 | - | 5e-4 |
| Llama 350M | AdamW | 16 | 5e-4 | cosine | - | - | 0.9 | 0.999 | 5e-4 |
| Llama 350M | AdaBelief | 16 | 5e-4 | cosine | - | - | 0.9 | 0.999 | 5e-4 |
| Llama 350M | Lion | 16 | 5e-5 | cosine | - | - | 0.95 | 0.98 | 5e-4 |

## I.2 THEORETICAL VS. EMPIRICAL OBSERVATIONS

An interesting discrepancy emerges between theory and practice for vision tasks. Our theoretical bound $T \leq \frac{V(0)^{1-\gamma_1}}{\alpha_1(1-\gamma_1)}$ with $\gamma_1 = \frac{1+p_2}{2}$ predicts that *smaller* $p_2$ values should yield faster convergence. However, vision tasks empirically favor *larger* $p_2$ values.

We hypothesize this reflects the tension between continuous-time theoretical guarantees and discrete-time stochastic optimization realities:

- **Smaller** $p_2$: Aggressive finite-time dynamics provide theoretical acceleration but may induce numerical instability in discrete stochastic updates
- **Larger** $p_2$: More conservative dynamics sacrifice some theoretical speed but maintain robustness to gradient noise and discretization errors

## I.3 INFLUENCE OF STEP SIZE

In order to investigate the robustness of the proposed method against step size (learning rate), the performance of PEFToM on CIFAR100 with ResNet34 with different learning rates and $p_2$ is tested, as listed in Table 10. Based on the result, we find that:

- **Small learning rate (lr=0.001)**: Performance is relatively stable across all $p_2$ values, achieving optimal accuracy around 76.52% at $p_2 = 0.6$. This regime is insensitive to $p_2$ selection, with accuracy varying only by $\sim$3.7% across different $p_2$ values.

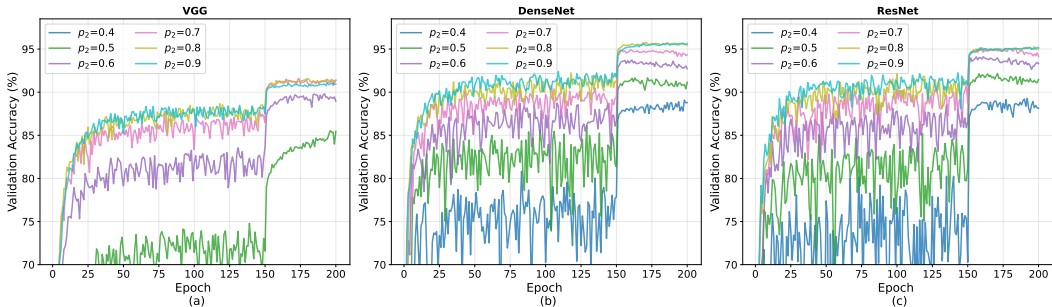

Figure 6: CIFAR-10 Sensitivity to $p_2$. Test accuracy across three architectures using PEFToM. Peak performance occurs at $p_2 \in [0.8, 0.9]$ for all models, with ResNet-34 and DenseNet-121 showing robust plateau regions.

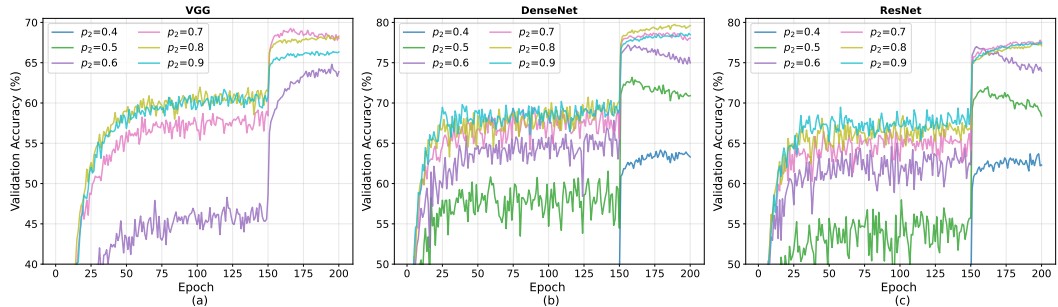

Figure 7: CIFAR-100 Sensitivity to $p_2$. Similar patterns to CIFAR-10 but with increased sensitivity at extreme values. VGG-11 exhibits training instability for $p_2 \leq 0.5$.

- **Moderate learning rate (lr=0.01)**: Larger $p_2$ values demonstrate superior robustness, with $p_2 \in [0.7, 0.9]$ achieving 77.10–77.54% accuracy. In contrast, smaller $p_2$ values show degraded performance, particularly $p_2 = 0.4$ (62.30%), indicating increased sensitivity to discretization effects.

- **Large learning rate (lr=0.1)**: Only $p_2 \geq 0.8$ maintains reasonable performance ($\approx$69%), while smaller $p_2$ values lead to substantial degradation (e.g., $p_2 = 0.4$ achieves only 31.66%). This demonstrates that the dual-power structure with appropriate $p_2$ selection provides inherent stability against discretization effects.

These results demonstrate that: (i) our method maintains practical stability across learning rates spanning two orders of magnitude (0.001–0.1) when $p_2$ is appropriately chosen ($p_2 \geq 0.7$), (ii) the power parameter $p_2$ effectively modulates discretization sensitivity, with larger values providing better robustness to step size variations, and (iii) the finite-time regime enables a practical balance between convergence acceleration and numerical stability.

## J  SCALABILITY TO LLAMA 350M

To further validate the scalability of the proposed method, we conduct experiments on Llama 350M following the same experimental protocol as Llama 60M, with appropriately tuned learning rates. Since SGD, SGDM, and PEFToM demonstrated limited performance in preliminary experiments, they are excluded from this comparison.

As shown in Figure 9 and Table 11, EFToM converges faster than AdamW and AdaBelief, particularly in the later training stages. Notably, even the basic EFT variant outperforms AdamW by the end of training, achieving lower final test loss (3.773 vs 3.845). EFToM consistently achieves the best performance across most checkpoints, reaching a final test loss of 3.698, demonstrating superior convergence efficiency on this larger-scale model.

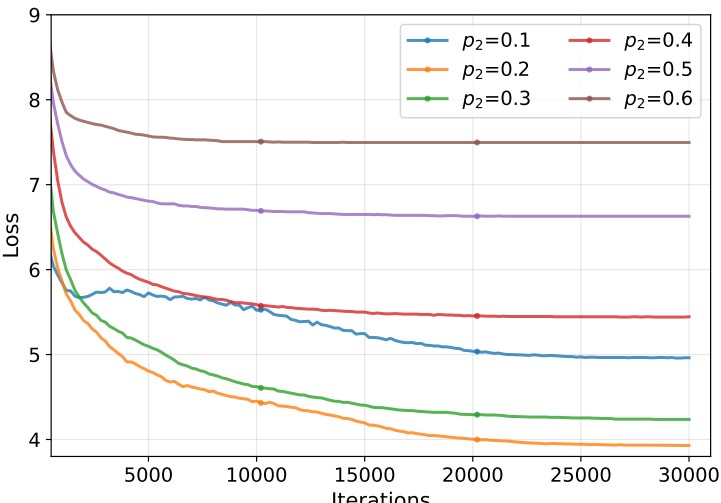

Figure 8: **C4 Language Modeling Sensitivity to** $p_2$. Validation loss using EFToM on Llama 60M. Smaller $p_2$ values achieve superior performance, with $p_2 = 0.2$ optimal. Values $p_2 \geq 0.4$ show limited convergence, indicating narrower effective parameter range for NLP tasks.

Table 8: Quantitative Sensitivity Analysis on CIFAR-10/100. Test accuracy (%) for PEFToM with learning rate 0.01 across 200 epochs. Values $< 50\%$ indicate training failure (marked in gray for clarity).

| $p_2$ | CIFAR-10 | | | CIFAR-100 | | |
|---|---|---|---|---|---|---|
| | VGG-11 | ResNet-34 | DenseNet-121 | VGG-11 | ResNet-34 | DenseNet-121 |
| 0.4 | 10.00 | 88.14 | 88.79 | 1.00 | 62.30 | 63.30 |
| 0.5 | 85.46 | 91.46 | 91.15 | 1.00 | 68.40 | 70.91 |
| 0.6 | 88.95 | 93.32 | 92.71 | 63.89 | 73.97 | 74.98 |
| 0.7 | 91.38 | 94.14 | 94.33 | 68.28 | 77.54 | 78.04 |
| 0.8 | 91.39 | 95.04 | 95.45 | 68.05 | 77.10 | 79.60 |
| 0.9 | 91.02 | 95.17 | 95.62 | 66.34 | 77.34 | 78.49 |

Table 9: Quantitative Sensitivity Analysis on C4. Validation perplexity for EFToM with learning rate 0.01 on Llama 60M. Lower values indicate better performance.

| $p_2$ | C4 (Llama 60M) | |
|---|---|---|
| | 10k iters | 30k iters |
| 0.1 | 5.53 | 4.96 |
| 0.2 | **4.43** | **3.93** |
| 0.3 | 4.61 | 4.24 |
| 0.4 | 5.57 | 5.44 |
| 0.5 | 6.69 | 6.63 |
| 0.6 | 7.51 | 7.50 |

Table 10: Step Size Analysis on CIFAR 100 with ResNet34. Test accuracy (%) for PEFToM ($c_1 = 1, c_2 = 0$) with learning rate different learning rate across 200 epochs.

| | CIFAR-100 | | |
|---|---|---|---|
| $p_2$ | lr=0.001 | lr=0.01 | lr=0.1 |
| 0.4 | 75.98 | 62.30 | 31.66 |
| 0.5 | 76.41 | 68.40 | 40.91 |
| 0.6 | 76.52 | 73.97 | 43.35 |
| 0.7 | 75.42 | 77.54 | 57.70 |
| 0.8 | 74.01 | 77.10 | 69.71 |
| 0.9 | 72.83 | 77.34 | 69.24 |

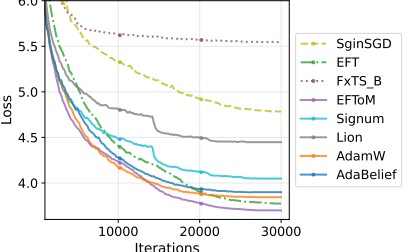

Figure 9: Validation loss on C4 dataset with Llama 350M

Table 11: Train and validation loss on C4 at different iterations with Llama 350M.

| | Train Loss | | | | Test Loss | | | |
|---|---|---|---|---|---|---|---|---|
| Optimizer | 5k | 10k | 20k | 30k | 5k | 10k | 20k | 30k |
| SignSGD | 5.4375 | 5.593 | 5.125 | 4.750 | 5.701 | 5.326 | 4.927 | 4.785 |
| EFT | 4.875 | 4.625 | 4.250 | 3.781 | 5.053 | 4.406 | 3.903 | 3.773 |
| FxTS-GF(M) | 5.563 | 5.844 | 5.594 | 5.438 | 5.747 | 5.625 | 5.573 | 6.545 |
| EFToM | **4.500** | 4.469 | **4.125** | **3.734** | **4.600** | 4.236 | **3.781** | **3.698** |
| Signum | 4.656 | 4.75 | 4.406 | 4.063 | 4.823 | 4.486 | 4.122 | 4.049 |
| Lion | 4.875 | 5.063 | 4.719 | 4.406 | 5.052 | 4.813 | 4.50 | 4.448 |
| AdamW | 4.563 | **4.406** | 4.219 | 3.844 | 4.678 | **4.179** | 3.882 | 3.845 |
| AdaBelief | 4.688 | 4.5 | 4.250 | 3.891 | 4.802 | 4.295 | 3.938 | 3.898 |

