# OpenReview forum: "Scalable Element-wise Finite-Time Optimization for Deep Neural Networks"
_ICLR.cc/2026/Conference — Submitted to ICLR 2026_

### Official Review · Reviewer_MagU · 2025-10-26

**Soundness:** 2
**Presentation:** 3
**Contribution:** 2
**Rating:** 2
**Confidence:** 4

**Summary:**

This paper builds upon the work of Budhraja et al. by introducing an ODE that exhibits finite-time convergence guarantees. Compared to previously studied finite-time gradient flows, the proposed methods enjoys coordinatewise adaptivity and can be seen as a generalization of signSGD. A proof of finite-time and fixed-time convergence is provided in the smooth and PL regime, and the results are shown to hold for variants incorporating exponential moving averages and momentum. In addition, the empirical effectiveness of the proposed optimizer is explored on vision and pretraining tasks.

**Strengths:**

1. The paper is clearly written and easy to follow.
2. The idea of extending existing finite-time gradient flows to be coordinatewise adaptable is well-motivated.
3. The hyperparameter conditions for finite-time and fixed-time convergence are clearly stated in the theorems.

**Weaknesses:**

1. The method itself, equation (2), is not intuitively motivated or explained. For instance, it is not immediately obvious what $p_1$ and $p_2$ are supposed to represent, even though they are of critical importance to the effectiveness of the method.
2. The proposed optimizer introduces a slew of hyperparameters, for which there is little discussion of practical recommendations or tuning suggestions.
3. The theoretical results are given for the continuous-time dynamics, and no discussion is provided on how these results transfer to the discretized versions. The paper lacks the results that are expected of a typical optimizer paper, e.g. a convergence rate on the gradient norm or objective value.
4. No empirical evidence is provided to support the claim that the method achieves finite-time convergence. If the continuous-time results do indeed transfer to the discrete-time setting, then such an empirical result would greatly support the effectiveness of the proposed method.
5. A central claim of the paper is that the proposed method is scalable, but there is little evidence to support this with the provided experiment results. I would suggest providing results on models with 7B+ parameters and evaluating on tasks that are significantly harder than CIFAR.

I did not carefully review the proofs, but unfortunately, the mentioned issues are already significant enough for me to recommend rejection.

**Questions:**

See Weaknesses

---

> ### Author Response · Authors · 2025-11-22
> **Response to reviewer MagU**
>
> We would like to thank the reviewer for the valuable comments and time spent on this paper. All revisions are **blue highlighted** in the revised manuscript.
>
> **comment 1**:  The method itself, equation (2), is not intuitively motivated or explained. For instance, it is not immediately obvious what $p_1$ and $p_2$ are supposed to represent, even though they are of critical importance to the effectiveness of the method.
>
> **Response to comment 1**: We thank the reviewer for this important feedback. We have significantly revised the presentation to address these concerns:
> - **(1)Intuitive Explanation of Equation (2)**: We rewrite equation (2) in an alternative form to clarify its meaning:
> \begin{equation}
> \dot{\mathbf{w}} = -c_1 \, \mathrm{sign}(\mathbf{g}) \odot |\mathbf{g}|^{p_2} - c_2 \, \mathrm{sign}(\mathbf{g}) \odot |\mathbf{g}|^{2-p_1} = - \frac{c_1 \mathbf{g}} {|\mathbf{g}|^{1-p_2}} - \frac{c_2\mathbf{g}}{|\mathbf{g}|^{p_1-1}},
> \end{equation}
> where $p_1 \leq 2$, $p_2 \in (0,1)$. This can be viewed as gradient descent with an element-wise adaptive learning rate of scale $c_1|\mathbf{g}|^{p_2-1} + c_2|\mathbf{g}|^{1-p_1}$.
> - (2) Intuition for the dual-power design:
>     $p_2$ term ($p_2 < 1$): When $|\mathbf{g}|$ is small, $|\mathbf{g}|^{p_2-1}$ becomes large, providing aggressive adaptation in flat regions to escape plateaus and saddle points.
> - (3) $p_1$ term ($p_1 \leq 2$): When $p_1 < 1$ (fixed-time regime): $|\mathbf{g}|^{1-p_1}$ increases with big $|\mathbf{g}|$, enabling faster descent in steep regions (though requiring careful tuning in discrete time due to discretization effects).
>
>  Both terms accelerate convergence but target different gradient regimes. The $p_1$ term controls the aggressiveness in large gradient regions (dominating at early training stage), while  $p_2$ term handles small gradients, aims to achieve faster convergence near the optimal region.
>
> This differs from Adam-style methods by using *fractional powers* rather than square roots, which (theoretically in continuous time) enables finite-time convergence guarantees.
>
>
> **comment 2**: The proposed optimizer introduces a slew of hyperparameters, for which there is little discussion of practical recommendations or tuning suggestions.
>
> **response to comment 2**: we are sorry that the hyperparameter tuning strategy are not well explained. We now provide clear guidance in Appendix H of the revised manuscript. Specifically,
> **Algorithm-specific parameters tuning** ($p_1$, $p_2$, $c_1$, $c_2$) follows a two-stage tuning strategy to reduce the search space and computational cost:
>
>   - **Stage 1** (Finite-time regime)}: We set $c_1 = 1$, $c_2 = 0$, which focuses on the finite-time dynamics term.  We search $p_2 \in \{0.1, 0.2, 0.3, 0.4, 0.5, 0.6, 0.7, 0.8, 0.9\}$.
>
>    - **Stage 2** (Fixed-time regime): After identifying the best $p_2^*$ from Stage 1, we optionally explore the fixed-time regime by fixing $c_1 = c_2 = 1$ and best $p_2 $, then searching $p_1 \in \{0.1, 0.2, 0.3, 0.4, 0.5, 0.6, 0.7, 0.8, 0.9, 0.95, 0.98\}$.
>
> - **Recommended defaults**: Based on extensive experiments across tasks and architectures, we recommend the following strong defaults:
>   - CV tasks with CNNs: $c_1 = 1$, $c_2 = 0$, $p_2 = 0.8$
>   - NLP tasks with Transformers: $c_1 = 1$, $c_2 = 0$, $p_2 = 0.2$
>
> **Note**: The *parameter setting and tuning process of all methods regarding different dataset* are now given in **Appendix H**, please find the details there.
>
>
> **comment 3**: The theoretical results are given for the continuous-time dynamics, and no discussion is provided on how these results transfer to the discretized versions. The paper lacks the results that are expected of a typical optimizer paper, e.g. a convergence rate on the gradient norm or objective value.
>
> **Response to comment3**: In the previous manuscript, we focused on the finite/fixed time convergence, which can only achieved within the contineous-time framework. It is nontrivial to provide such convergence analysis for the EFT and its two momentum variant.
> In order to address your concern, we have added the convergence analysis of its discretized version.  We obtain $O(K^{-1/2})$ convergence rates (measured in gradient norms) for all three variants under standard assumptions (L-smoothness and bounded gradients). **EFT convergence(for example)**: Under coordinate-wise $L$-smoothness and bounded gradient assumption ($|\mathbf{g_i}|\le G$), let the discrete EFT algorithm run for $K$ iterations with step size
> \begin{equation}
> \eta = \frac{1}{\sqrt{KLC_F}}, \quad \text{where } C_F = 2d(c_1^2 G^{2p_2} + c_2^2 G^{2(2-p_1)}).
> \end{equation}
> Then:
> \begin{equation}
> \min_{k \in [K]} ||\mathbf{g_k}||_{1+p_2}^{1+p_2} \leq \frac{\mathcal{L}(\mathbf{w}_0) - \mathcal{L}^*}{c_1\sqrt{LC_F K}} + \frac{\sqrt{LC_F}}{2c_1 \sqrt{K}} .
> \end{equation}
> **please refer to section 4.5 and Appendix G of the revised manuscript for details**

---

> ### Author Response · Authors · 2025-11-22
> **Response to reviewer MagU (comments 4-5)**
>
> **comment 4**: No empirical evidence is provided to support the claim that the method achieves finite-time convergence. If the continuous-time results do indeed transfer to the discrete-time setting, then such an empirical result would greatly support the effectiveness of the proposed method.
>
> **Response to comment 4**: Thank you for your comments.  Since deep learning optimization landscapes are highly non-convex and lack strong convexity, directly demonstrating finite-time convergence is challenging. To address your concern, we provide empirical validation on a strongly convex problem where finite-time convergence can be rigorously verified.  Specifically, We empirically validate the finite-time convergence property on a strongly convex regularized logistic regression problem:
> $\min_{\mathbf{w} \in \mathbb{R}^d} f(\mathbf{w}) = \frac{1}{n} \sum_{i=1}^{n} \log\left(1 + \exp(-y_i \mathbf{w}^\top \mathbf{x}_i)\right) + \frac{\lambda}{2} ||\mathbf{w}||^2,$
> with $n=200$, $d=20$, $\lambda=0.1$.   (**refer to 5.1 of the revised manuscript for more details**)
>
> - We first compared the proposed method with GD and Adam under same initial condition.  The detail convergence curve are presented in Figure 2 ( in the revised manuscript) which reveal fundamental distinctions between asymptotic and finite-time convergence behaviors: EFT variants exhibit  *vertical drops* to $10^{-6}$ within finite time. GD shows classic exponential decay but never settles exactly after 10 seconds. Adam converges faster than GD but oscillates around convergence. Table 1 listed the convergence time of different algorithms.
>
> ### Table 1
> | Optimizers| Conv. Time (s) |
> |---|---|
> | GD| N/A |
> | Adam| 2.81 |
> | EFT (p1=1.5, p2=0.5)  | 1.61 |
> | EFToM (p1=1.5, p2=0.5) | 1.60 |
> | PEFToM (p1=1.5, p2=0.5)  | **0.54** |
>
> - We demonstrate the finite-time convergence characteristic of EFT (the EFToM and PEFToM are similar) with various initial values while keeping the hyperparameters ($p_1=1.5, p_2=0.5$). The convergence time as well as the theoretical bound of EFT with  are listed in Table 2 to demonstrate that the proposed method can converge within the theoretical bounds.
>
> ### Table 2. EFT (p₁ = 1.5, p₂ = 0.5) Metrics
>  | \|$\mathbf{w}_0 - \mathbf{w}^*$\| | Conv. Time (s) | Upper Bound (s) |
> |---|---|---|
>  | 0.4254 | 0.842 | 47.1|
>  | 1.3864 |1.444| 89.64|
>  | 3.58| 2.783| 133.10|
> | 15.34 | 6.491| 247.55|
> | 30.13 | 9.53 | 326.16 |
>
>
>
> **comment 5**: A central claim of the paper is that the proposed method is scalable, but there is little evidence to support this with the provided experiment results. I would suggest providing results on models with 7B+ parameters and evaluating on tasks that are significantly harder than CIFAR.
>
> **Response to comment 5**:  Thank you for this important concern about scalability validation. We would like to clarify our scalability claims and provide additional context.
>
> - **1. Massive Scale-Up from Control-Theoretic Baselines**:
> Our primary scalability contribution is methodological rather than purely empirical. We address a fundamental limitation of control-theoretic finite-time methods that previously prevented their application to deep learning. Previous control-theoretic approaches like FxTS-GF were limited to problems with $ 10^5 $ parameters due to dimensional coupling. Our methods  eliminate dimensional coupling through element-wise dynamics, enabling application to modern neural networks successfully handle neural networks with up to $ 6 \times 10^7$ parameters (Llama 60M), representing a 600$\times$ scale increase.
>
> - **2. Comprehensive Cross-Architecture Validation**:
> We validate our framework across diverse settings that demonstrate its general applicability: including CNNs (VGG-11, ResNet34, DenseNet-121) up to $10^7$ parameters on CIFAR-10/100, and Transformer (Llama 60M ) on C4. In the revised manuscript, the vision transformer are tested on tiny Imagenet. Importantly, our methods consistently outperform or match specialized optimizers
>
> - **3 Addressing the 7B+ Parameter Request**:
> While we acknowledge the value of experiments on 7B+ models, we respectfully note several considerations:
>   - (a)  Our framework is a general-purpose optimizer with rigorous finite-time convergence guarantees, not a specialized LLM optimizer. The comparison baseline should be general methods (Adam, SGD) rather than architecture-specific ones.
>   - (b) We are currently conducting experiments on Llama 350M (3.5×$10^8$ parameters) on the C4 dataset to further validate scalability. However, 7B+ parameter models present substantial computational barriers that extend beyond the review timelines considering the hyperparameter exploration and that multiple optimizers need to be compared.
>
> While 7B+ experiments would provide an additional data point, our multi-scale validation combined with theoretical dimensional independence already substantiates the key scalability claim for a general-purpose optimizer framework.

---

> ### Author Response · Authors · 2025-11-26
> **Response to MagU (experiment on larger models)**
>
> To further validate the scalability of the proposed method, we conduct experiments on
> Llama 350M (5.8× larger than Llama 60M) following the same experimental protocol with
> appropriately tuned learning rates for each optimizer. Since SGD, SGDM, and PEFToM
> demonstrated limited performance in preliminary experiments, they are excluded from this
> comparison to focus on the most competitive baselines.
>
>
> As  detailed in Table 3,
> our proposed methods demonstrate strong performance on this larger-scale model. EFToM
> achieves the best results across most checkpoints, converging faster than both AdamW
> and AdaBelief, particularly in the later training stages (after 10k iterations). The
> final test loss of 3.698 represents a 3.8\% improvement over AdamW (3.845) and 5.1\%
> over AdaBelief (3.898). Notably, even the basic EFT variant, without momentum enhancement,
> outperforms AdamW by the end of training (3.773 vs 3.845), validating the effectiveness
> of our element-wise finite-time dynamics.  **Details are provided in Appendix J of the revised manuscript**
>
> The consistent superiority across different model scales (60M and 350M) strongly supports
> the scalability claims of our framework and suggests promising potential for even larger
> language models.
>
>
>
> ### Table 3. Training and Validation Loss on C4 at Different Iterations (Llama 350M)
> | Optimizer | Train Loss (5k) | Train Loss (10k) | Train Loss (20k) | Train Loss (30k) | Test Loss (5k) | Test Loss (10k) | Test Loss (20k) | Test Loss (30k) |
> |-----------|-----------------|------------------|------------------|------------------|----------------|-----------------|-----------------|-----------------|
> | SignSGD | 5.4375 | 5.593 | 5.125 | 4.750 | 5.701 | 5.326 | 4.927 | 4.785 |
> | EFT | 4.875 | 4.625 | 4.250 | 3.781 | 5.053 | 4.406 | 3.903 | 3.773 |
> | FxTS-GF(M) | 5.563 | 5.844 | 5.594 | 5.438 | 5.747 | 5.625 | 5.573 | 6.545 |
> | **EFToM** | **4.500** | 4.469 | **4.125** | **3.734** | **4.600** | 4.236 | **3.781** | **3.698** |
> | Signum | 4.656 | 4.75 | 4.406 | 4.063 | 4.823 | 4.486 | 4.122 | 4.049 |
> | Lion | 4.875 | 5.063 | 4.719 | 4.406 | 5.052 | 4.813 | 4.50 | 4.448 |
> | AdamW | 4.563 | **4.406** | 4.219 | 3.844 | 4.678 | **4.179** | 3.882 | 3.845 |
> | AdaBelief | 4.688 | 4.5 | 4.250 | 3.891 | 4.802 | 4.295 | 3.938 | 3.898 |

---

### Official Review · Reviewer_JQh7 · 2025-10-29

**Soundness:** 3
**Presentation:** 3
**Contribution:** 2
**Rating:** 4
**Confidence:** 5

**Summary:**

This paper introduces an Element-wise Finite-Time (EFT) optimization framework and its momentum variants for deep neural network training. The core motivation is to adapt control-theoretic finite-time and fixed-time stability concepts of ODEs to large-scale deep learning. The proposed EFT framework replaces this global norm dependency with element-wise operations based on the sign and fractional powers of individual gradient components.

**Strengths:**

Problem Relevance: The paper addresses a practical and important issue: how to apply optimization theories with stronger convergence guarantees (finite/fixed-time) to large-scale, high-dimensional deep learning tasks, overcoming the scalability bottlenecks of existing methods.

Experimental Results: The paper presents preliminary evidence showing that the proposed methods (especially the momentum variants) outperform some baseline optimizers (including SGD, AdamW, and FxTS-GF) on benchmarks like CIFAR image classification and C4 language modeling, indicating its potential.

**Weaknesses:**

Major ones:

1. Borrowed Core Idea: The core theoretical framework using dual-power dynamics (combining terms with exponents less than 1 and greater than 1) to achieve finite/fixed-time convergence is not original to this paper. This concept has already been developed such as Powerball method (https://arxiv.org/pdf/1603.07421), FxTS-GF method (https://arxiv.org/pdf/1808.10474) and others.

2. Discretization Gap: The paper provides continuous-time analysis but implements discrete updates via Euler discretization. The authors acknowledge a "discretization gap" in the appendix, noting that the fixed-time regime ($p_1 < 1$) is sensitive to discretization effects and step size choices in practice, potentially leading to instability. This is a critical issue. Properties relied upon in continuous-time finite-time stability proofs often do not hold under fixed step-size discretization $\eta$. The practical relevance of the continuous-time guarantees is therefore doubtful without a discrete-time analysis or a more thorough empirical investigation of how step size $\eta$ affects stability and convergence. Please refer to https://www.ijcai.org/proceedings/2020/451 for further details, could these results be helpful for analyzing (1)?

(Minor ones)
3. Contribution as Adaptation: The main claimed innovation is making the dynamics element-wise to eliminate dimensional coupling. The principle of element-wise scaling (based on local gradient info rather than global norms) is a key feature of successful adaptive optimizers in deep learning (like AdaGrad, RMSprop, Adam) and is a common technique. Therefore, the core contribution appears to be applying a standard deep learning heuristic (element-wise adaptation) to an existing theoretical framework. The work should be more accurately positioned as an element-wise adaptation of existing fixed-time gradient flows, not an entirely new framework or paradigm.

4. Interpretation of Results: The results presented in the paper that frames SignSGD and Signum as limiting cases of their framework were already known in Zhou et al. IJCAI, 2020  (https://www.ijcai.org/Proceedings/2020/0451.pdf). The connection drawn to SignSGD/Signum as limiting cases ($p_1 \to 2, p_2 \to 0$) 13 is mathematically interesting but its practical significance is unclear. As parameters approach these limits, the theoretical convergence bounds might degrade or become infinite. Does this connection offer new insights into why SignSGD works, or is it just a boundary condition of the mathematical form? The paper claims it provides "theoretical grounding" but doesn't elaborate on the specific insights gained.

**Questions:**

please refer to my specific comments,

---

> ### Author Response · Authors · 2025-11-22
> **Response to Reviewer JQh7**
>
> We would like to thank the reviewer for the valuable comments and time spent on this paper. All revisions are **blue highlighted** in the revised manuscript.
>
> **comment 1**: borrowed core idea
> **Response to comment 1**:   We sincerely thank the reviewer for this insightful comment and for bringing the Powerball method to our attention. This feedback has prompted us to carefully examine the relationship between our work and existing methods.
>
> - **Primary Motivation and Acknowledgment**
> We acknowledge that our work was primarily motivated by FxTS-GF and the goal of applying control-theoretic finite/fixed-time convergence to large-scale deep learning. We do not claim to have invented the dual-power dynamics framework, which originates from control theory. Following the reviewer's feedback, we have carefully reviewed the Powerball method and recognize its application of finite-time concepts in discrete optimization.
>
> - **Key Differences from Existing Work**
>   - Compared to Powerball (2017):  Powerball analyzes finite-time convergence from a discrete-time perspective, proving linear convergence rates for strongly convex functions. Their continuous-time ODE analysis serves primarily as intuition.
>    -  We provide rigorous continuous-time analysis for both finite-time and fixed-time convergence. In addition, we systematically develop two momentum variants with convergence guarantees.
>
>  - Compared to FxTS-GF:
>     - FxTS-GF uses global norm-dependent updates with dimensional coupling
>     - Our contribution: Element-wise reformulation that eliminates dimensional coupling, enabling scalability (demonstrated in Fig. 1c of our manuscript)
>
> - 3. Our Main Contributions: Bridging Theory to Practice
>   - Previous works typically extend to only one momentum variant. We systematically develop two momentum variants (EFToM and PEFToM) and prove both preserve finite/fixed-time convergence. This is non-trivial because momentum introduces additional state variables that complicate Lyapunov analysis. By demonstrating that control-theoretic methods can compete with or outperform standard optimizers on real-world tasks, we hope to encourage the deep learning community to explore and leverage the rich toolkit of control theory.
>
> **comment2**: Discretization Gap
>
> **Response to Comment 2**: We sincerely thank the reviewer for this critical observation and for suggesting the pbSGD reference (IJCAI 2020), which we have now cited in the revised manuscript in section 4.5. We acknowledge that the discretization gap between continuous-time finite-time convergence and discrete-time implementation is an important theoretical challenge.
>
> - **(1) provided discrete time convergence**, following the reviewer's suggestion and inspired by the analysis techniques in pbSGD and SignSGD, we have developed rigorous convergence proofs for the discrete versions of our algorithms. We obtain $O(K^{-1/2})$ convergence rates (measured in gradient norms) for all three variants under standard assumptions (L-smoothness and bounded gradients). **EFT convergence(for example)**: Under coordinate-wise $L$-smoothness and bounded gradient assumption ($|\mathbf{g_i}|\le G$), let the discrete EFT algorithm run for $K$ iterations with step size
> \begin{equation}
> \eta = \frac{1}{\sqrt{KLC_F}}, \quad \text{where } C_F = 2d(c_1^2 G^{2p_2} + c_2^2 G^{2(2-p_1)}).
> \end{equation}
> Then:
> \begin{equation}
> \min_{k \in [K]} ||\mathbf{g_k}||_{1+p_2}^{1+p_2} \leq \frac{\mathcal{L}(\mathbf{w}_0) - \mathcal{L}^*}{c_1\sqrt{LC_F K}} + \frac{\sqrt{LC_F}}{2c_1 \sqrt{K}} .
> \end{equation}
> **please refer to section 4.5 and Appendix G of the revised manuscript for details**
>
> - **(2) sensitivity analysis**. To empirically investigate the discretization effects, we have added experiments studying step size sensitivity in Appendix I. Where we use the PEFToM optimizer with ResNet 34 on CIFAR100. The learning rate are ranging from (0.001, 0.01, 0.1), with the finite-time convergence coefficient $p_2$ selected from $\{0.4, 0.5, 0.6, 0.7, 0.8, 0.9 \}$,
>
> ### Table 1, Sensitivity analysis
> | p₂  | lr=0.001 | lr=0.01 | lr=0.1 |
> |-----|----------|---------|--------|
> | 0.4 | 75.98    | 62.30   | 31.66  |
> | 0.5 | 76.41    | 68.40   | 40.91  |
> | 0.6 | 76.52    | 73.97   | 43.35  |
> | 0.7 | 75.42    | 77.54   | 57.70  |
> | 0.8 | 74.01    | 77.10   | 69.71  |
> | 0.9 | 72.83    | 77.34   | 69.24  |
>
> Results demonstrate strong sensitivity to learning rate selection. Small lr (0.001) yields stable performance across all $p_2$ values. Moderate lr (0.01) achieves optimal accuracy (77.10--77.54\%) with larger $p_2 \geq 0.7$. Large lr (0.1) requires $p_2 \geq 0.8$ to maintain reasonable performance, highlighting the critical interaction between step size and power parameter selection.

---

> > ### Author Response · Authors · 2025-11-22
> > **Response to Reviewer JQh7 (minor)**
> >
> > **comments 3**: Contribution as Adaptation
> >
> > **Response to comment 3**:  We respectfully disagree with the characterization that our contribution is merely "applying a standard heuristic." While element-wise operations are common in adaptive optimizers, our contribution differs fundamentally in both motivation and theoretical implications.
> >
> > - **(1) Distinction from Adaptive Optimizers**
> >    - Adaptive methods (Adam, etc.): Element-wise scaling adapts learning rates based on gradient statistics ($m_t/\sqrt{v_t}$) to handle varying gradient magnitudes
> >     - Our element-wise formulation: Eliminates dimensional coupling inherent in control-theoretic methods to enable finite-time convergence analysis in high dimensions
> >
> > - **(2) Theoretical Non-triviality**
> > The reformulation from global to element-wise dynamics with convergence guarantee is non-trivial:
> > FxTS-GF:  $\dot{\bf{w}}(t) = -c_1 \frac{\nabla \mathcal{L}(\bf{w})}{\|\nabla \mathcal{L}(\bf{w})\|^{\frac{p_1-2}{p_1-1}}} - c_2 \frac{\nabla \mathcal{L}(\bf{w})}{\|\nabla \mathcal{L}(\bf{w})\|^{\frac{p_2-2}{p_2-1}}},$, EFT:  $\dot{\mathbf{w}} = -c_1 \, \mathrm{sign}(\mathbf{g}) \odot |\mathbf{g}|^{p_2} - c_2 \, \mathrm{sign}(\mathbf{g}) \odot |\mathbf{g}|^{2-p_1}.$  This requires New Lyapunov analysis. Extension to moving average momentum and hevay ball moementum variants introducing more challenges.
> >
> >
> > - (3) Consider Adam—it combines momentum with element-wise adaptive scaling, "adapting" existing ideas but recognized as a significant contribution. Similarly, our systematic reformulation with rigorous analysis bridges control theory and deep learning by adapting.
> >
> > - **(4) Revised Positioning**: We propose framing our work in the contribution part as:  *A coordinate-independent reformulation of fixed-time gradient flows that eliminates dimensional coupling through element-wise dual-power dynamics, enabling scalable application to deep neural networks with rigorous convergence guarantees.*
> >
> >
> > **comment 4**: Interpretation of Results.
> >
> > **Response to comments 4**:  We thank the reviewer for this insightful comment and clarify our contributions relative to prior work.
> >
> > - **(1) Distinction from Zhou et al. (IJCAI 2020)**
> > While Zhou et al. frame SignSGD as a limiting case using powerball momentum, there are key differences:
> >
> >     - Zhou et al. (pbSGD): Uses heavy-ball momentum and frames SignSGD as $\gamma \to 0$
> >    - Our work: Uses exponential moving average (EMA) momentum and frames both SignSGD and Signum as limiting cases through dual-power parameters $(p_1, p_2)$ .
> > Importantly, *Signum cannot be derived from pbSGD* because pbSGD uses heavy-ball momentum, whereas Signum is fundamentally based on EMA momentum. Our framework unifies both under a single theoretical lens.
> >
> > - **(2) Practical Significance of the Limiting Cases**
> > The reviewer raises an important question about practical insights. We clarify:
> >   - Convergence behavior at limits: Contrary to the reviewer's concern, our convergence bounds *do not degrade* as $(p_1, p_2) \to (2,0)$.
> > The convergence time of EFT is:
> >
> > \begin{equation}
> > T^\star \leq \frac{d^{p_2}(\mathcal{L}(\mathbf{w_0}) - \mathcal{L}^*)^{1-\gamma_1}}{c_1 (2\mu)^{(1+p_2)/2} (1-\gamma_1)}   + \frac{d^{2-p_1}(\mathcal{L}(\mathbf{w_0}) - \mathcal{L}^\star)^{1-\gamma_2}}{c_2  (2\mu)^{(3-p_1)/2} (1-\gamma_2)}
> > \end{equation}
> >
> > when $(p_1, p_2) \to (2,0)$, it reduce to
> > \begin{equation}
> > T_{sign}^\star\leq \frac{(\mathcal{L}(\mathbf{w_0}) - \mathcal{L}^*)^{1-\gamma_1}}{c_1 (2\mu)^{(1+p_2)/2} (1-\gamma_1)}   + \frac{(\mathcal{L}(\mathbf{w_0}) - \mathcal{L}^\star)^{1-\gamma_2}}{c_2  (2\mu)^{(3-p_1)/2} (1-\gamma_2)}
> > \end{equation}
> >
> >  In fact, the convergence time may become shorter at the limit, i.e. $T_{sign}^* \le T^*$, consistent with the finite-time convergence properties of sliding mode control.
> >
> >    - **(3) rigorous proof**: To make a more rigorous claim, we have provided the finite-time convergence gurantee with $p_1=2, p_2=0$ in Section 4 and Appendix F of the revised manuscript.
> > Specifically， with dynamic $\dot{\mathbf{w}} = -(c_1 + c_2) \text{sign}(\mathbf{g}).$ It's convegence time can be quantified as $        T \le \frac{\sqrt{2(L(w_0) - L^*)}}{(c_1+c_2) \sqrt{\mu}},$ **please refer to section 4.4 and Appendix F** for detail proof.

---

### Official Review · Reviewer_J1XU · 2025-11-01

**Soundness:** 3
**Presentation:** 3
**Contribution:** 2
**Rating:** 4
**Confidence:** 3

**Summary:**

The paper motivates from the continuous-time counterpart of optimization algorithms, and proposes novel optimizers EFTOM and PEFTOM based on this perspective. Theoretical analysis and empirical justification of the algorithms are provided in the together for the algorithms.

**Strengths:**

1. The idea of deriving optimizers from a continuous perspective looks interesting, and the derived algorithms are reasonable since it matches the optimzier derived from the standard steepest descent framework (e.g., SignSGD).
2. The theoretical convergence results seem to be valid, showing finite-time convergence of the algorithm incorporated with momentum.

**Weaknesses:**

1. For the theoretical part, the convergence rates all have heavy dependence on dimensionality, which can be extremely large in practice. This drawback can be avoided by analysis of similar algorithms like SignSGD or Signum.
2. For the experiment part, could the authors give how the hyperparameters are tuned for EFTOM and PEFTOM? It also looks strange that Table 5 indicates the best SGD learning rate is $ 0.2 $ , which is not listed in the listed SGD learning rate search grid. This unclear setting raises questions in fair comparisons with other optimizers.

**Questions:**

1. Could we fix the explicit dependence on dimensionality by employing similar assumptions as the SignSGD paper, i.e., we consider smoothness in a different norm?

---

> ### Author Response · Authors · 2025-11-22
> **Response to Reviewer J1XU**
>
> We would like to thank the reviewer for the valuable comments and time spent on this paper. All revisions are **blue highlighted** in the revised manuscript.
>
> **Comment 1 and question**: For the theoretical part, the convergence rates all have heavy dependence on dimensionality, which can be extremely large in practice. This drawback can be avoided by analysis of similar algorithms like SignSGD or Signum.  Could we fix the explicit dependence on dimensionality by employing similar assumptions as the SignSGD paper, i.e., we consider smoothness in a different norm?
>
> **Response to comment 1 and question**: Thank you for this insightful suggestion regarding dimension dependence. We have carefully reconsidered this issue and revised our discrete-time convergence analysis following the coordinate-wise framework used in SignSGD.
>
> - **(1) Nature of Dimension Dependence in Element-Wise Methods**: Dimension dependence is a fundamental characteristic of element-wise optimization methods that arises from the mathematical structure of the algorithms rather than the analysis technique. We clarify this point below:
> Our algorithms operate element-wise on individual coordinates:
> \begin{equation}
> \mathbf{v}_i \propto |g_i|^{p_2} \cdot \text{sign}(g_i), \quad i = 1, \ldots, d.
> \end{equation}
>
> To establish global convergence guarantees, we must aggregate information from $d$ independent coordinate updates into a single convergence metric. This aggregation inherently involves dimensional factors. Specifically:
> Our element-wise dynamics naturally produce terms like $\sum_{i=1}^d |g_i|^{1+p_2}$. Converting to standard norms (e.g., $\|\mathbf{g}\|_2$ for PL conditions) requires inequalities that introduce dimensional scaling
>  For instance, $||\mathbf{g}||_1 \leq \sqrt{d} ||\mathbf{g}||_2$ and $\left(\sum_i |g_i|^q\right)^{1/q} \leq d^{1/q-1/2} ||\mathbf{g}||_2$.
>
>
>
> - **(2)Demension dependence in SignSGD**
> The coordinate-wise framework in SignSGD provides cleaner analysis, but it is important to note that **dimension dependence persists in this formulation** . SignSGD achieves the convergence rate:
> \begin{equation}
> \mathbb{E}\left[\frac{1}{K} \sum_{k=0}^{K-1} \|\mathbf{g}_k\|_1\right]^2 \leq \frac{1}{K}\left[\sqrt{||\vec{L}||_1}(\mathcal{L}_0-\mathcal{L}^* + \frac{1}{2}) + 2\|\boldsymbol{\sigma}\|_1\right]^2,
> \end{equation}
>
> where $||\vec{L}||= \sum_{i=1}^d L_i$ is the sum of coordinate-wise Lipschitz constants. The $l_1$ norm implicitly includes the demisntion $d$. Therefore, SignSGD itself cannot get rid of the demension parameter when obtain the convergence rate
>
>
> - Following your suggestion, we have adopted the coordinate-wise smoothness framework similar to that used in SignSGD. We have added three new theorems for the discrete-time algorithms EFT, EFToM, and PEFToM.
>
> - **(3) EFT convergence**: Under coordinate-wise $L$-smoothness and bounded gradient assumption ($|\mathbf{g_i}|\le G$), let the discrete EFT algorithm run for $K$ iterations with step size
> \begin{equation}
> \eta = \frac{1}{\sqrt{KLC_F}}, \quad \text{where } C_F = 2d(c_1^2 G^{2p_2} + c_2^2 G^{2(2-p_1)}).
> \end{equation}
> Then:
> \begin{equation}
> \min_{k \in [K]} ||\mathbf{g_k}||_{1+p_2}^{1+p_2} \leq \frac{\mathcal{L}(\mathbf{w}_0) - \mathcal{L}^*}{c_1\sqrt{LC_F K}} + \frac{\sqrt{LC_F}}{2c_1 \sqrt{K}} .
> \end{equation}
> Note that the convegence rate is similart to SignSGD. Tthe convergence of EFToM and PEFToM can be found in **Appendix G** of the revised manuscript.
>
>
> **Comments 2**:  hyperparameters tuning and learning rate clarification.
>
> **Response to comment 2**: Thank you for raising these important questions about experimental fairness and reproducibility. We provide detailed clarifications below and have updated the manuscript accordingly.
>
>
>
> - (1) We apologize for the confusion regarding the SGD learning rate in Table 5. The discrepancy arose because:
>  The hyperparameter search grid described in Appendix H was specific to CIFAR-10/100 experiments. Table 5 reports C4 language modeling results (NLP task with Transformers), which required a different search grid. For C4, we used a fine-grained learning rate grid: $\eta \in \{ 0.0001, 0.0005, 0.001, 0.002, 0.005, 0.01, 0.02, 0.05, 0.1, 0.2, 0.5, 1.0\}.$  We have now clarified this in the revised manuscript with task-specific hyperparameter tables.
>
> - **(2) Complete Hyperparameter Tuning Procedure**
>
>   - Stage 1 (Finite-time regime)}: We set $c_1 = 1$, $c_2 = 0$, which focuses on the finite-time dynamics term.  We search $p_2 \in \{0.1, 0.2, 0.3, 0.4, 0.5, 0.6, 0.7, 0.8, 0.9\}$.
>
>    - Stage 2 (Fixed-time regime): After identifying the best $p_2^*$ from Stage 1, we optionally explore the fixed-time regime by fixing $c_1 = c_2 = 1$ and best $p_2 $, then searching $p_1 \in \{0.1, 0.2, 0.3, 0.4, 0.5, 0.6, 0.7, 0.8, 0.9, 0.95, 0.98\}$.
>
> **Note**: The *parameter setting and tuning process of all methods regarding different dataset* are now given in **Appendix H**, please find the details there.

---

### Official Review · Reviewer_ATA2 · 2025-11-04

**Soundness:** 3
**Presentation:** 3
**Contribution:** 3
**Rating:** 6
**Confidence:** 3

**Summary:**

This work introduces an element-wise optimization framework that overcomes the scalability issues of control-theoretic methods in high-dimensional deep learning. By using coordinate-independent dynamics, it ensures rigorous finite-time convergence while preserving the adaptivity essential for neural network training. This theory unifies various optimizers under one principled foundation, rigorously justifying the empirical success of SignSGD and Signum as special cases.

**Strengths:**

- The work provides valuable new insights by bridging control-theoretic stability principles and large-scale non-convex optimization.

- It is beneficial for understanding optimizer convergence behavior, moving beyond traditional asymptotic rates to finite-time guarantees.

- The proposed PEFTom optimizer demonstrates competitive performance compared to existing state-of-the-art methods.

**Weaknesses:**

- The experimental results do not fully substantiate the central theoretical claim that PEFTom achieves finite-time convergence. More targeted experiments are needed to directly illustrate this property.

- The experimental setup relies on CNN architectures (e.g., ResNet, DenseNet) and datasets (CIFAR-10, CIFAR-100) that are now considered somewhat stale. To convincingly demonstrate the optimizer's effectiveness and scalability, experiments on more modern architectures (e.g., Vision Transformers) and larger, more complex datasets (e.g., ImageNet) are recommended.

**Questions:**

The PEFTom optimizer introduces several new hyperparameters (e.g., $c_1, c_2, p_1, p_2$). Could the authors provide:

* A sensitivity analysis to show how the algorithm's performance is affected by variations in these parameters?

* Practical guidance or heuristics for tuning these hyperparameters effectively?

---

> ### Author Response · Authors · 2025-11-22
> **Response to reviewer ATA2**
>
> We would like to thank the reviewer for the valuable comments and time spent on this paper. All revisions are **blue highlighted** in the revised manuscript.
>
> **Comment 1**: More targeted experiments are needed to directly illustrate finite-time property.
>
> **Response to comment 1**: Thank you for your comments. To address your concern, we provide empirical validation on a strongly convex problem where finite-time convergence can be rigorously verified. Specifically, We empirically validate the finite-time convergence property on a strongly convex regularized logistic regression problem:
> $\min_{\mathbf{w} \in \mathbb{R}^d} f(\mathbf{w}) = \frac{1}{n} \sum_{i=1}^{n} \log\left(1 + \exp(-y_i \mathbf{w}^\top \mathbf{x}_i)\right) + \frac{\lambda}{2} ||\mathbf{w}||^2,$
> with $n=200$, $d=20$, $\lambda=0.1$.   (**refer to 5.1 of the revised manuscript for more details**)
>
> - We first compared the proposed method with GD and Adam under same initial condition.  The detail convergence curve are presented in Figure 2 ( in the revised manuscript) which reveal fundamental distinctions between asymptotic and finite-time convergence behaviors: EFT variants exhibit  *vertical drops* to $10^{-6}$ within finite time. GD shows classic exponential decay but never settles exactly after 10 seconds. Adam converges faster than GD but oscillates around convergence. Table 1 listed the convergence time of different algorithms.
>
> ### Table 1
> | Optimizers| Conv. Time (s) |
> |---|---|
> | GD| N/A |
> | Adam| 2.81 |
> | EFT (p1=1.5, p2=0.5)  | 1.61 |
> | EFToM (p1=1.5, p2=0.5) | 1.60 |
> | PEFToM (p1=1.5, p2=0.5)  | **0.54** |
>
> - We demonstrate the finite-time convergence characteristic of EFT (the EFToM and PEFToM are similar) with various initial values while keeping the hyperparameters ($p_1=1.5, p_2=0.5$). The convergence time as well as the theoretical bound of EFT with  are listed in Table 2 to demonstrate that the proposed method can converge within the theoretical bounds.
>
> ### Table 2. EFT (p₁ = 1.5, p₂ = 0.5) Metrics
>  | \|$\mathbf{w}_0 - \mathbf{w}^*$\| | Conv. Time (s) | Upper Bound (s) |
> |---|---|---|
>  | 0.4254 | 0.842 | 47.1|
>  | 1.3864 |1.444| 89.64|
>  | 3.58| 2.783| 133.10|
> | 15.34 | 6.491| 247.55|
> | 30.13 | 9.53 | 326.16 |
>
> **Comments 2**: experiments on more modern architectures (e.g.,Vision Transformers) and larger, more complex datasets (e.g., ImageNet) are recommended
>
> **Response to comment 2**:  Thank you for the suggestion. We have added a new empirical simulation result using the ViT and the tiny Imagenet, shown in Table 3.  On tiny ImageNet with Vision Transformer, PEFToM reaches 31.69\%, lower than most optimizers but still surpassing SGDM (30.86\%). EFT achieves 42.32\% accuracy, substantially improving over SGD  and outperforming SignSGD. EFToM further advances to 45.51\%, competitive with adaptive methods like AdamW and AdaBelief, which demonstrate its scalability.  (**please refer to section 5.2 in the revised manuscript**)
>
> ### Table 3. Test acc (%) of ViT on tiny Imagenet
> | Method    | 30 epochs | 90 epochs |
> |---|---|---|
> | SGD |16.99| 28.29 |
> | SignSGD| 32.3 | 41.88|
> | EFT | 28.51 | 42.32 |
> | FxTS-GF(M)| 14.05 | 26.18|
> | EFToM| 39.34| 45.51|
> | Signum| 39.8| 45.79|
> | Lion | 42.71 | 46.76|
> | AdamW| 41.13  | 45.7|
> | AdaBelief | 41.12 | 46.41 |
> | SGDM|18.54| 30.86|
> | PEFToM| 21.20 | 31.69|
>
>
> **Response to questions** : Thank you for the questions.
> - **tuning strategy**: To reduce the search space and computational cost, we adopt a two-stage tuning strategy:
>   - Stage 1: We set $c_1 = 1$, $c_2 = 0$, which focuses on the finite-time dynamics term. We search $p_2 \in {0.1, 0.2, 0.3, 0.4, 0.5, 0.6, 0.7, 0.8, 0.9}$.
>   - Stage 2: After identifying the best $p_2$ from Stage 1, we explore the fixed-time regime by fixing $c_1 = c_2 = 1$ and best $p_2 $, then searching $p_1 \in {0.1, 0.2, 0.3, 0.4, 0.5, 0.6, 0.7, 0.8, 0.9, 0.95, 0.98}$.
>   - Since the $p_1$ itself cannot achieve a convergence guarantee while $p_2$ term itself can guarantee finite-time convergence, we use this two-stage tuning process.
>
>  - **sensitivity analysis**: based on the tuning strategy, in most case, $p_2$ is the most influenced parameter. We give the sensitivity of $p_2$ in Table 4 (CIFAR100 for demonstration, **more results can be found in Appendix I** of the revised manuscript)
>
> ### Table 4. Test acc (%) of PEFToM on CIFAR100 (lr=0.01)
> | p₂  | VGG11 | ResNet34 |DenseNet121|
> |---|---|---|---|
> | 0.4 |  *1.00*  | 62.30 | 63.30  |
> | 0.5 | *1.00*  | 68.40 | 70.91 |
> | 0.6 |  63.89 | 73.97 | 74.98 |
> | 0.7 | 68.28 | 77.54 | 78.04  |
> | 0.8 |  68.05 | 77.10 | 79.60 |
> | 0.9 | 66.34  | 77.34 | 78.49  |
>
>   Performance remains within 2\% for
>     $p_2 \in [0.7, 0.9]$ across architectures. In contrast, smaller $p_2$ values show degraded performance. Specifically, for VGG11, training fails when $p_2 <0.5$.

---

> > ### Author Response · Authors · 2025-11-26
> > **Response to reviewer ATA2 (experiment for scability verification)**
> >
> > To further validate the scalability of the proposed method, we conduct experiments on
> > Llama 350M (5.8× larger than Llama 60M) following the same experimental protocol with
> > appropriately tuned learning rates for each optimizer. Since SGD, SGDM, and PEFToM
> > demonstrated limited performance in preliminary experiments, they are excluded from this
> > comparison to focus on the most competitive baselines.
> >
> >
> > As  detailed in Table 3,
> > our proposed methods demonstrate strong performance on this larger-scale model. EFToM
> > achieves the best results across most checkpoints, converging faster than both AdamW
> > and AdaBelief, particularly in the later training stages (after 10k iterations). The
> > final test loss of 3.698 represents a 3.8\% improvement over AdamW (3.845) and 5.1\%
> > over AdaBelief (3.898). Notably, even the basic EFT variant, without momentum enhancement,
> > outperforms AdamW by the end of training (3.773 vs 3.845), validating the effectiveness
> > of our element-wise finite-time dynamics.  **the details are provided in Appendix J of the revised manuscript**
> >
> > The consistent superiority across different model scales (60M and 350M) strongly supports
> > the scalability claims of our framework and suggests promising potential for even larger
> > language models.
> >
> >
> >
> > ### Table 5. Training and Validation Loss on C4 at Different Iterations (Llama 350M)
> > | Optimizer | Train Loss (5k) | Train Loss (10k) | Train Loss (20k) | Train Loss (30k) | Test Loss (5k) | Test Loss (10k) | Test Loss (20k) | Test Loss (30k) |
> > |-----------|-----------------|------------------|------------------|------------------|----------------|-----------------|-----------------|-----------------|
> > | SignSGD | 5.4375 | 5.593 | 5.125 | 4.750 | 5.701 | 5.326 | 4.927 | 4.785 |
> > | EFT | 4.875 | 4.625 | 4.250 | 3.781 | 5.053 | 4.406 | 3.903 | 3.773 |
> > | FxTS-GF(M) | 5.563 | 5.844 | 5.594 | 5.438 | 5.747 | 5.625 | 5.573 | 6.545 |
> > | **EFToM** | **4.500** | 4.469 | **4.125** | **3.734** | **4.600** | 4.236 | **3.781** | **3.698** |
> > | Signum | 4.656 | 4.75 | 4.406 | 4.063 | 4.823 | 4.486 | 4.122 | 4.049 |
> > | Lion | 4.875 | 5.063 | 4.719 | 4.406 | 5.052 | 4.813 | 4.50 | 4.448 |
> > | AdamW | 4.563 | **4.406** | 4.219 | 3.844 | 4.678 | **4.179** | 3.882 | 3.845 |
> > | AdaBelief | 4.688 | 4.5 | 4.250 | 3.891 | 4.802 | 4.295 | 3.938 | 3.898 |

---

### Meta-Review · Area_Chair_18Lv · 2026-01-11

**Summary:**

Across reviews, the main concerns shaping my recommendation are:
- (i) whether the paper’s claimed contribution is sufficiently novel given closely related finite/fixed-time gradient-flow and dual-power methods, and
- (ii) whether the empirical evidence convincingly supports the central claims (finite-time behavior and scalability) in realistic deep learning settings.

While reviewers generally found the paper clear and the direction interesting, the strongest negative assessment emphasized that the dual-power finite/fixed-time idea is largely inherited from prior work and that the paper’s contribution may be primarily an element-wise adaptation, with unclear added practical insight from framing SignSGD/Signum as limiting cases. Another reviewer recommended rejection due to missing discrete-time relevance and weak scalability evidence in the original submission. Although the rebuttal adds substantial material, the remaining issues are chiefly about contribution strength and empirical convincingness.

**Reviewer Concerns:**

### Addressed by the rebuttal:
The authors directly responded to repeated requests for bridging theory and practice by adding discrete-time convergence results (rates measured in gradient norm under standard assumptions, including a coordinate-wise analysis inspired by SignSGD/pbSGD-style arguments). They also added a targeted finite-time demonstration on a strongly convex regularized logistic regression problem, which strengthens the claim that finite-time behavior can be observed in an appropriate setting. For reproducibility, they clarified hyperparameter tuning via a two-stage procedure and provided sensitivity evidence for key parameters. On “stale experiments,” they added results on a Vision Transformer (ViT on tiny ImageNet) and an additional larger-scale language modeling experiment (Llama 350M), partially addressing scalability and modern-architecture coverage.

### Still outstanding / only partially addressed:
The most consequential critique remains. A high-confidence reviewer argued that the core dual-power finite/fixed-time framework is not new (citing related lines such as Powerball/FxTS-GF/pbSGD) and that the “SignSGD/Signum as limiting cases” framing does not clearly translate into new practical insight. The rebuttal acknowledges prior work and reframes contributions, but does not fully resolve whether the incremental step (element-wise reformulation plus momentum variants) reaches the novelty bar.

Empirically, while ViT/tiny ImageNet and Llama 350M broaden coverage, the scalability concern is only partially mitigated: the added finite-time evidence is on a strongly convex toy problem rather than non-convex deep learning training, and the larger-model comparison omits some standard baselines due to weak preliminary performance, which can raise questions about the completeness of comparisons.

**Reviewer Scores:**

- Reviewer ATA2 (6, confidence 3): With added finite-time validation, ViT/tiny ImageNet, and hyperparameter sensitivity/tuning guidance, likely +1 (to 7) or remain 6 with a more positive stance.

- Reviewer J1XU (4, confidence 3): With clarified tuning and corrected task-specific learning-rate grids, plus added coordinate-wise discrete-time analysis, likely flip to positive side (to 6).

 - Reviewer JQh7 (4, confidence 5): The added discrete-time analysis and step-size sensitivity address the discretization-gap concern, but novelty/practical-insight concerns likely remain. Expected change: 0 (stay 4 ).

- Reviewer MagU (2, confidence 4): Improved motivation, added tuning guidance, discrete-time results, and larger-model experiments should increase the score, but remaining scalability/convincingness issues likely keep it below threshold. Expected change: +1 (to 3).

---

### Decision · Program_Chairs · 2026-01-26

Reject